# Functional Scaling Laws in Kernel Regression: Loss Dynamics and Learning Rate Schedules

**Binghui Li**[1,*]  **Fengling Chen**[2,*]  **Zixun Huang**[2,*]  **Lean Wang**[3,*]  **Lei Wu**[1,2,4,†]

[1]Center for Machine Learning Research, Peking University
[2]School of Mathematical Sciences, Peking University
[3]State Key Laboratory of Multimedia Information Processing,
School of Computer Science, Peking University
[4]AI for Science Institute, Beijing

{libinghui, lean}@pku.edu.cn,  flchen_lwycc@stu.pku.edu.cn
alexpku@stu.pku.edu.cn,  leiwu@math.pku.edu.cn

We strongly recommend reading the arXiv version of this paper,
available at https://arxiv.org/abs/2509.19189.

## Abstract

Scaling laws have emerged as a unifying lens for understanding and guiding the training of large language models (LLMs). However, existing studies predominantly focus on the final-step loss, leaving open whether the entire *loss dynamics* obey similar laws and, crucially, how the *learning rate schedule* (LRS) shapes them. We address these gaps in a controlled theoretical setting by analyzing stochastic gradient descent (SGD) on a power-law kernel regression model. The key insight is a novel **intrinsic-time** viewpoint, which captures the training progress more faithfully than iteration count. We then establish a **Functional Scaling Law (FSL)** that captures the full loss trajectory under arbitrary LRSs, with the schedule's influence entering through a simple convolutional functional. We further instantiate the theory for three representative LRSs—constant, exponential decay, and warmup–stable–decay (WSD)—and derive explicit scaling relations in both data- and compute-limited regimes. These comparisons explain key empirical phenomena: (i) higher-capacity models are more data- and compute-efficient; (ii) learning-rate decay improves training efficiency; and (iii) WSD-type schedules outperform pure decay. Finally, experiments on LLMs ranging from 0.1B to 1B parameters demonstrate the practical relevance of FSL as a surrogate model for fitting and predicting loss trajectories in large-scale pre-training.

## 1 Introduction

It is well established that the training of large-scale deep learning models mysteriously follows *scaling laws*, which describe how model performance scales *predictably* with available resources such as compute or data [19]. In particular, the landmark study by Kaplan et al. [25] demonstrated that, in LLM pre-training, the loss $L$ decreases with model size $M$ and dataset size $D$ according to a power-law relation:

$$L(M, D) = L_0 + C_M M^{-\alpha_M} + C_D D^{-\alpha_D}, \tag{1}$$

where $\alpha_M$ and $\alpha_D$ are the scaling exponents, $L_0$ denotes the irreducible loss, and $C_M, C_D$ are some constants. Such empirical relations have proven remarkably robust across scales, architec-

---

[*]Equal contribution.
[†]Corresponding author.

39th Conference on Neural Information Processing Systems (NeurIPS 2025).

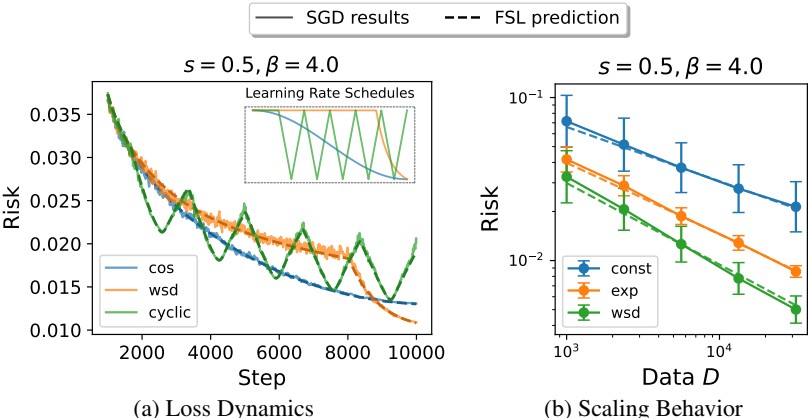

Figure 1: **FSL accurately captures the loss dynamics and scaling behavior of SGD in PLK regression.**
In both subplots, solid lines denote the results of SGD, while dashed lines represent the corresponding FSL
predictions. **(a)** FSL accurately tracks the loss dynamics of SGD, averaged over 1000 runs, for three learning rate
schedules: cosine, WSD-like, and a non-standard cyclic schedule. **(b)** FSL predictions (dashed) are computed
using the analytical forms from Section 5, and compared with the mean of 200 SGD runs (solid).

tures, and training setups [20, 60, 36] and have become foundational principles for guiding LLM
development [18, 24, 1, 5, 55, 27]. In practice, they are now routinely used to design optimal
resource-allocation strategies [20] and to tune key hyperparameters such as learning rates and batch
sizes [36, 29].

Despite their empirical success, the theoretical understanding of scaling laws remains limited. Recent
studies have begun to illuminate the underlying mechanisms [54, 22, 39, 63, 23, 42, 43, 2, 14, 3, 7,
35, 50, 8, 70], yet two important gaps persist:

- **Determinants of scaling efficiency.** Existing studies lack a systematic characterization of
  how key factors—such as model capacity, task difficulty, and hyperparameter choices—govern
  scaling efficiency, as reflected by the exponents $\alpha_M$ and $\alpha_D$. In particular, **learning rate
  schedules (LRSs)** are known to be critical in practice [40, 4, 17], but their precise role in
  shaping scaling behavior remains unclear.

- **Beyond the final-step loss.** The scaling law (1) focuses only on the end-of-training loss [25, 20],
  thus leaving open whether the *full trajectory* follows similar laws. Empirical studies [59, 38]
  suggest this possibility, but the fits there are still crude and lack theoretical grounding.

## 1.1   Our Contribution

In this paper, we take a step toward addressing these gaps in a controlled yet representative theoretical
setting. We study stochastic gradient descent (SGD) training of the **power-law kernel (PLK)**
regression—a widely adopted surrogate for scaling-law analysis [7, 3, 50, 35, 8]. The PLK regression
is characterized by four parameters: the task difficulty $s$, the capacity exponent $\beta$, the model size $M$,
and the label-noise level $\sigma$. To capture the influence of learning-rate schedules (LRSs), we model
SGD via an **intrinsic-time SDE**, in which the concept of **intrinsic time** emerges as a key quantity
enabling a unified characterization of how different LRSs shape the loss dynamics. Building on this
formulation, we establish the **Functional Scaling Law (FSL)**, which provides a unified description
of the entire *loss dynamics*—beyond the traditional final-loss prediction. Concretely, for a general
intrinsic-time LRS $\gamma : [0, \infty) \to [0, \infty)$, and under some conditions, the dynamics of the expected
loss $\mathbb{E}[\mathcal{R}(\boldsymbol{\nu}_t)]$ (where $t$ denotes the intrinsic time) satisfies:

$$\mathbb{E}[\mathcal{R}(\boldsymbol{\nu}_t)] - \underbrace{\frac{\sigma^2}{2}}_{\text{irreducible error}} \approx \underbrace{\frac{1}{M^{s\beta}}}_{\text{approx. error}} + \underbrace{e(t)}_{\text{signal learning}} + \underbrace{\int_0^t \mathcal{K}(t-z)\,[e(z)+\sigma^2]\,\gamma(z)\,\mathrm{d}z}_{\text{noise accumulation}}, \quad (2)$$

where $e(t) = (1+t)^{-s}$ and $\mathcal{K}(t) = (1+t)^{-(2-1/\beta)}$. Each term in FSL admits clear interpretation:
$\frac{\sigma^2}{2}$ denotes the **irreducible error** caused by label noise, $M^{-s\beta}$ represents the **approximation error**,
$e(t)$ characterizes the **signal-learning dynamics** under noiseless (full-batch) gradient descent, and
the final term captures the injection and dissipation of gradient noise, with the LRS $\gamma$ entering through

Table 1: **Learning-rate schedule (LRS) strongly influences scaling efficiency in power-law kernel regression.** Efficiency is determined by two key factors: relative task difficulty $s \in (0, \infty)$ and model capacity $\beta > 1$. We distinguish between an *easy-learning regime* ($s \geqslant 1 - 1/\beta$) and a *hard-learning regime* ($s < 1 - 1/\beta$).

| Learning Rate Schedule (LRS) | Data-Optimal Scaling Laws | | Compute-Optimal Scaling Laws | |
|---|---|---|---|---|
| | Easy | Hard | Easy | Hard |
| Constant | $D^{-\frac{s}{s+1}}$ | | $C^{-\frac{s\beta}{1+s\beta+\beta}}$ | |
| Exponential-decay | $D^{-\frac{s\beta}{1+s\beta}}(\log D)^{\frac{s\beta}{1+s\beta}}$ | $D^{-s}(\log D)^s$ | $C^{-\frac{s\beta}{2+s\beta}}(\log C)^{\frac{s\beta}{2+s\beta}}$ | $C^{-\frac{s\beta}{1+\beta}}(\log C)^{\frac{s\beta}{1+\beta}}$ |
| Warmup-stable-decay (WSD) | $D^{-\frac{s\beta}{1+s\beta}}(\log D)^{\frac{s\beta-s}{1+s\beta}}$ | $D^{-s}$ | $C^{-\frac{s\beta}{2+s\beta}}(\log C)^{\frac{s\beta-s}{2+s\beta}}$ | $C^{-\frac{s\beta}{1+\beta}}$ |

a tractable **convolutional functional**. The function $\mathcal{K}$, a **memory kernel**, which we also refer to as the **forgetting kernel**, quantifies how fast the injected noise dissipates during training.

Building on FSL, we derive concrete scaling laws for the final-step loss under three representative LRSs—constant, exponential decay [15], and warmup–stable–decay (WSD) [69, 21]—in both **data-limited** and **compute-limited regimes**. The results, summarized in Table 1, recover and extend prior analyses [7, 8, 50, 35], and reveal several unifying insights.

- **Scaling efficiency of different schedules.** WSD achieves the best scaling efficiency, followed by exponential decay and then constant schedules. This efficiency hierarchy provides theoretical justification for learning-rate decay and explains empirical success of WSD [69, 21, 58, 36].
- **Role of model capacity.** Higher-capacity models are consistently more efficient in both compute and data, highlighting the necessity of scaling model capacity [25].
- **Data–model trade-off.** Compute-optimal training requires scaling data more than model size, consistent with established heuristics in LLM pre-training [20].
- **Scaling law for peak learning rate.** Optimal scaling requires the peak learning rate (LR) to scale appropriately with the training budget (data or compute), revealing the importance of careful peak LR tuning [6, 29].

Beyond PLK regression, we further apply the FSL ansatz to fit and predict loss trajectories from LLM pre-training experiments with model sizes ranging from 0.1B to 1B parameters, covering both dense and MoE architectures. These results highlight the potential of FSL as a practical surrogate for understanding and guiding LLM pre-training. To better situate our contribution, we provide a detailed comparison with related work in Appendix B.

**Notation.** For any $n \in \mathbb{N}$, let $[n] := \{1, 2, \ldots, n\}$. For a positive semi-definite (PSD) matrix $\mathbf{S}$, denote by $\mu_j(\mathbf{S})$ its $j$-th largest eigenvalue, and define the $\mathbf{S}$-induced norm $\|\mathbf{u}\|_{\mathbf{S}} := \sqrt{\mathbf{u}^\top \mathbf{S} \mathbf{u}}$ for any vector $\mathbf{u}$. We write $\mathbf{A} \preceq \mathbf{B}$ (resp. $\mathbf{A} \succeq \mathbf{B}$) if $\mathbf{B} - \mathbf{A}$ (resp. $\mathbf{A} - \mathbf{B}$) is PSD. Throughout the paper, we use $\asymp$ to denote equivalence up to a constant factor, and $\lesssim$ (resp. $\gtrsim$) to denote an inequality up to a constant factor. For two nonnegative functions $f, g : \mathbb{R}_{\geqslant 0} \to \mathbb{R}_{\geqslant 0}$, we write $f(t) \asymp g(t)$ if there exist constants $C_1, C_2 > 0$ (independent of $t$) such that $C_1 f(t) \leqslant g(t) \leqslant C_2 f(t), \ \forall\, t \geqslant 0$.

## 2 Power-Law Kernel (PLK) Regression

Let $\mathcal{X}$ denote the input domain and $\mathcal{D}$ the input distribution, and assume labels are generated by $y = \langle \boldsymbol{\phi}(\mathbf{x}), \boldsymbol{\theta}^* \rangle + \epsilon$, where $f^*(\mathbf{x}) := \langle \boldsymbol{\phi}(\mathbf{x}), \boldsymbol{\theta}^* \rangle$ is the target function, and the label noise $\epsilon \sim \mathcal{N}(0, \sigma^2)$ is independent of $\mathbf{x}$. Here $\boldsymbol{\phi} : \mathcal{X} \to \mathbb{R}^N$ with $N \in \mathbb{N}_+ \cup \{\infty\}$ is a feature map, satisfying the following assumption:

**Assumption 2.1** (Hypercontractivity). Let $\mathbf{H} := \mathbb{E}_{\mathbf{x} \sim \mathcal{D}}[\boldsymbol{\phi}(\mathbf{x})\boldsymbol{\phi}(\mathbf{x})^\top]$ be the feature covariance. There exist constants $C_1, C_2 > 0$ such that for any PSD matrix $\mathbf{A} \in \mathbb{R}^{N \times N}$, $C_1 \operatorname{tr}(\mathbf{H}\mathbf{A}) \mathbf{A} \preceq \mathbb{E}_{\mathbf{x} \sim \mathcal{D}}\Big[ \big(\boldsymbol{\phi}(\mathbf{x})^\top \mathbf{A} \boldsymbol{\phi}(\mathbf{x})\big) \boldsymbol{\phi}(\mathbf{x})\boldsymbol{\phi}(\mathbf{x})^\top \Big] - \mathbf{H}\mathbf{A}\mathbf{H} \preceq C_2 \operatorname{tr}(\mathbf{H}\mathbf{A}) \mathbf{A}$.

This condition ensures that the feature distribution is sufficiently regular—its fourth-order moments are controlled by the second-order ones [41]. It holds, for example, for Gaussian features $\boldsymbol{\phi}(\mathbf{x}) \sim \mathcal{N}(0, \mathbf{H})$ with $C_1 = 1, C_2 = 2$ (see Lemma F.1). For simplicity, we also assume:

**Assumption 2.2.** $\mathbf{H} = \operatorname{diag}(\lambda_1, \lambda_2, \ldots, \lambda_N)$ with $\lambda_1 \geqslant \lambda_2 \geqslant \cdots \geqslant \lambda_N$.

To learn $f^*$, we consider a **model of width** $M$: $f(\mathbf{x}; \mathbf{v}) = \sum_{j=1}^M v_j \mathbf{w}_j^\top \boldsymbol{\phi}(\mathbf{x}) =: \langle \mathbf{v}, \mathbf{W}\boldsymbol{\phi}(\mathbf{x}) \rangle$, where $\mathbf{v} \in \mathbb{R}^M$ denotes trainable weights and $\mathbf{W} \in \mathbb{R}^{M \times N}$ projects the $N$-dimensional features onto an $M$-dimensional subspace. We study two choices of projection $\mathbf{W}$:

- **Top-$M$ features:** $\mathbf{w}_j = \boldsymbol{e}_j$ for $j \in [M]$, i.e., selecting the top-$M$ features $\{\phi_j\}_{j=1}^M$;
- **Random-$M$ features:** $\mathbf{w}_j \sim \mathcal{N}(0, I_N)$ independently for $j \in [M]$.

The top-$M$ setting is a particularly simple yet analytically representative case, widely adopted in prior scaling-law studies [43, 13]. For random features [3, 7, 50, 35, 8], we will show that, in certain regimes, their scaling behavior closely parallels that of the top-$M$ case. As clarified in Appendix A.3, our setup is equivalent to learning with the kernel $K_\phi(\mathbf{x}, \mathbf{x}') := \phi(\mathbf{x})^\top \phi(\mathbf{x}')$.

We now formalize the key notions of *model capacity* and *task difficulty*. Let $\widehat{\phi}_j := \phi_j / \lambda_j^{1/2}$ for $j \in [N]$. So $\{\widehat{\phi}_j\}_{j=1}^N$ forms an orthonormal basis of $L^2(\mathcal{D})$.

**Assumption 2.3** (Model capacity)**.** The spectrum of the feature map satisfies $\lambda_j \asymp j^{-\beta}, \beta > 1$.

The condition $\beta > 1$ ensures $\mathrm{tr}(\mathbf{H}) = \sum_{j=1}^N \lambda_j \leqslant C$ for some constant $C$ independent of $N$, making our analysis *dimension-free* and applicable to the infinite-dimensional setting ($N = \infty$).

For the top-$M$ features, the model takes the form $f(\cdot; \mathbf{v}) = \sum_{j=1}^M v_j \phi_j = \sum_{j=1}^M v_j \lambda_j^{1/2} \widehat{\phi}_j \asymp \sum_{j=1}^M v_j\, j^{-\beta/2} \widehat{\phi}_j$ reveals that higher-index (less significant) features are increasingly down-weighted by the factor $j^{-\beta/2}$. As $\beta$ increases, the spectrum decays more rapidly, and the model *effectively* relies on fewer features. Hence, the model's expressive power is governed by two complementary factors: (i) the **model size** $M$, which controls how many features are retained, and (ii) the **capacity exponent** $\beta$, which controls how quickly these features decay in importance.

**Assumption 2.4** (Task difficulty)**.** Suppose $|\theta_j^*|^2 \asymp j^{-1} \lambda_j^{s-1}$ for some $s > 0$.

Under Assumptions 2.3 and 2.4, the target function admits the expansion $f^* = \sum_{j=1}^N \theta_j^* \phi_j \asymp \sum_{j=1}^N j^{-1/2} \lambda_j^{s/2} \widehat{\phi}_j \asymp \sum_{j=1}^N j^{-(s\beta+1)/2} \widehat{\phi}_j$. Since $\{\hat{\phi}_j\}$ are orthonormal, this assumption implies that the spectral energy of $f^*$ decays as a power law. The exponent $\alpha := s\beta$ therefore quantifies the task's **intrinsic difficulty**, which depends only on the target function itself and is independent of the model spectrum. In contrast, $s$ measures the **relative difficulty** with respect to a model of capacity $\beta$: for a fixed $f^*$ (fixed $\alpha$), adopting a higher-capacity model (smaller $\beta$) increases $s = \alpha/\beta$, making the task relatively easier. In other words, the same task appears easier to a higher-capacity model.

We remark that similar assumptions have been widely used in the analysis of kernel methods [12, 11, 57, 9, 39]. Our work builds upon and extends this line of research.

## 3 One-Pass SGD and Intrinsic-Time SDE

Given a data point $\mathbf{z} = (\mathbf{x}, y) \in \mathcal{X} \times \mathbb{R}$ and a model $f(\cdot; \mathbf{v})$, define the loss $\ell(\mathbf{z}, \mathbf{v}) = \frac{1}{2}\big(f(\mathbf{x}; \mathbf{v}) - y\big)^2$. Then, the population risk is $\mathcal{R}(\mathbf{v}) = \mathbb{E}_{\mathbf{z}}[\ell(\mathbf{z}, \mathbf{v})] = \frac{1}{2}\|\mathbf{W}^\top \mathbf{v} - \boldsymbol{\theta}^*\|_{\mathbf{H}}^2 + \frac{\sigma^2}{2} =: \mathcal{E}(\mathbf{v}) + \frac{\sigma^2}{2}$, where $\mathcal{E}(\mathbf{v})$ denotes the excess risk. We minimize $\mathcal{R}(\mathbf{v})$ via **one-pass SGD**, given by

$$\mathbf{v}_{k+1} = \mathbf{v}_k - \frac{\eta_k}{B_k} \sum_{\mathbf{z} \in S_k} \nabla_{\mathbf{v}} \ell(\mathbf{z}, \mathbf{v}_k), \tag{3}$$

where $S_k := \{(\mathbf{x}_{k,j}, y_{k,j})\}_{j=1}^{B_k}$ is a mini-batch of *i.i.d.* samples, $\eta_k$ and $B_k$ are the learning rate and batch size, respectively. The initialization is set to $\mathbf{v}_0 = \mathbf{0}$.

Throughout, we refer to $\boldsymbol{\eta} := (\eta_0, \eta_1, \ldots, \eta_{K-1})$ as the learning rate schedule (LRS). Common choices in practice include the cosine [37, 60], WSD [21], and multi-step [5] schedules (see Appendix A.2 for details). To analyze the effect of LRS, we rewrite (3) as

$$\mathbf{v}_{k+1} = \mathbf{v}_k - \eta_k\big(\nabla \mathcal{R}(\mathbf{v}_k) + \boldsymbol{\xi}_k\big), \tag{4}$$

where the gradient noise $\boldsymbol{\xi}_k = \frac{1}{B_k} \sum_{\mathbf{z} \in S_k} \nabla \ell(\mathbf{z}, \mathbf{v}_k) - \nabla \mathcal{R}(\mathbf{v}_k)$ satisfies $\mathbb{E}[\boldsymbol{\xi}_k] = 0, \mathbb{E}[\boldsymbol{\xi}_k \boldsymbol{\xi}_k^\top] = \frac{1}{B_k} \boldsymbol{\Sigma}(\mathbf{v}_k)$, with $\boldsymbol{\Sigma}(\cdot)$ denoting the noise covariance for batch size 1.

**Continuous-time limit.** Following prior work [30, 31, 32, 33, 51, 34], we analyze the continuous-time limit of SGD rather than the discrete update (3) or (4). This perspective makes the analysis more tractable and clarifies the emergence of scaling laws. Fix a discretization step size $h > 0$ and

let $\varphi_k := \eta_k/h$ for $k \in \mathbb{N}$. Then, (4) becomes $\mathbf{v}_{k+1} = \mathbf{v}_k - \varphi_k \nabla \mathcal{R}(\mathbf{v}_k) h - \varphi_k h \boldsymbol{\xi}_k$. For sufficiently small $h$, this iteration is well approximated by the Itô-type SDE [31, 45]:

$$d\bar{\mathbf{v}}_\tau = -\varphi(\tau) \nabla \mathcal{R}(\bar{\mathbf{v}}_\tau) \, d\tau + \varphi(\tau) \sqrt{\frac{h}{b(\tau)} \boldsymbol{\Sigma}(\bar{\mathbf{v}}_\tau)} \, d\mathbf{B}_\tau, \tag{5}$$

where $\mathbf{B}_\tau \in \mathbb{R}^M$ is an $M$-dimensional Brownian motion, and $\varphi(\cdot)$ is the continuous-time LRS satisfying $\varphi(kh) = \eta_k/h$ for all $k \in \mathbb{N}$; $b(\cdot)$ is the continuous-time batch-size schedule satisfying $b(kh) = B_k$ for all $k \in \mathbb{N}$. In (5), the learning rate affects both the drift and diffusion terms, thereby coupling the deterministic and stochastic effects.

**Intrinsic-time reparametrization.** In SDE (5), the physical time $\tau$ serves as the continuous analogue of the discrete step index $k$. However, when the learning rate varies over time, the actual training progress is determined not by the number of updates $k$ but by the accumulated step size $\sum_{j=1}^k \eta_j$, which more faithfully reflects the total optimization effort. Motivated by this observation, we introduce an *intrinsic time* variable that *rescales* the physical time $\tau$ according to the LRS:

$$t = T(\tau) := \int_0^\tau \varphi(r) \, dr, \tag{6}$$

which measures the LRS-adjusted training duration. Let $\boldsymbol{\nu}_t = \bar{\mathbf{v}}_{T^{-1}(t)}$. Applying Øksendal's time change formula [44] to the SDE (5) yields the **intrinsic-time SDE**:

$$d\boldsymbol{\nu}_t = -\nabla \mathcal{R}(\boldsymbol{\nu}_t) \, dt + \sqrt{\gamma(t) \boldsymbol{\Sigma}(\boldsymbol{\nu}_t)} \, d\mathbf{B}_t \ \ \text{with} \ \ \gamma(t) = \frac{h\varphi(T^{-1}(t))}{b(T^{-1}(t))}. \tag{7}$$

Here $\gamma(t)$ quantifies the joint effect of learning-rate and batch-size scheduling. Compared with (5), the LRS dependence is absorbed from the drift and retained only in the diffusion term, thereby **decoupling the deterministic and stochastic effects**. This structural simplification greatly facilitates the subsequent scaling analysis.

For a clearer explanation of the connection between the discrete SGD (4) and the SDE formulations (5) and (7), we refer the reader to Appendix A.4.

## 4 Intrinsic-Time Functional Scaling Laws

In this section, we present our main results on the Functional Scaling Law (FSL). All proofs are deferred to Appendix D. We begin with assumptions on the learning-rate schedule and model size.

**Assumption 4.1.** Suppose Assumptions 2.1, 2.3 and 2.4 hold. Assume both $M$ and $N-M$ are sufficiently large, and the LRS satisfies $\sup_{t \geqslant 0} \gamma(t) \leqslant C_3$ for a sufficiently small constant $C_3 > 0$.

**Theorem 4.2** (Intrinsic-Time FSL, top-$M$ features, hard-regime). *Under Assumption 4.1, let $\boldsymbol{\nu}_t$ denote the solution to the intrinsic-time SDE (7) with top-$M$ features. Then, for $f^*$ with difficulty $s \in (0, 1 - 1/\beta]$ and any $\sigma \geqslant 0$, it holds for all $t \geqslant 0$ that*

$$\mathbb{E}[\mathcal{R}(\boldsymbol{\nu}_t)] - \tfrac{1}{2}\sigma^2 \asymp M^{-s\beta} + e(t) + \int_0^t \mathcal{K}(t-z)[e(z) + \sigma^2]\gamma(z) \, dz, \tag{8}$$

*where $e(t) := (1+t)^{-s}$, $\mathcal{K}(t) := (1+t)^{-(2-1/\beta)}$.*

This theorem establishes that, for hard tasks with $s \leqslant 1 - 1/\beta$, the loss dynamics are fully characterized by the FSL (8). We explain the emergence of power laws in FSL from a multi-task learning perspective in Appendix A.5. Moreover, each term in the FSL (8) admits a clear interpretation:

- **Irreducible error:** $\frac{1}{2}\sigma^2$. This term is due to label noise.

- **Approximation error:** $M^{-s\beta}$. This term corresponds to the error due to finite model size, with the scaling efficiency is determined by the task's intrinsic difficulty $s\beta$.

- **Signal learning:** $e(t)$. This term corresponds to learning under full-batch gradient descent, capturing the rate at which SGD extracts the signal $f^*$. Moreover, the rate depends on the task's relative difficulty $s$. For a fixed target $f^*$ (fixed $\alpha = s\beta$), increasing model capacity (smaller $\beta$) accelerates its convergence since $s = \alpha/\beta$ becomes larger.

- **Noise accumulation:** $\int_0^t \mathcal{K}(t-z)[e(z)+\sigma^2]\gamma(z)\,\mathrm{d}z$**.** This term characterizes how the learning-rate and batch-size schedules shape the accumulation and dissipation of stochastic noise. The integrand $[e(z)+\sigma^2]\gamma(z)$ represents the instantaneous noise magnitude, where $e(z)$ captures mini-batch noise and $\sigma^2$ captures label noise. The **forgetting kernel** $\mathcal{K}(\cdot)$ quantifies how noise injected at time $z$ still affects the loss at time $t$. Due to $\mathcal{K}(t) \asymp t^{-(2-1/\beta)}$, a higher-capacity model (smaller $\beta$) tends to forget noise more slowly.

Notably, the last two terms together constitute the optimization error and two key factors govern the trade-off between the them: (i) **Model capacity:** Increasing model capacity ($\beta \downarrow$) accelerates signal learning but simultaneously slows noise forgetting. (ii) **Learning-rate and batch-size schedules:** Smaller learning rates or larger batch sizes suppress noise injection but also shorten the intrinsic training time. However, sufficient intrinsic time is important: the signal-learning term requires it to effectively reduce the risk, while the noise-forgetting term relies on it to forget noise memorized in early training. Hence, effective schedules must balance these competing objectives—*suppressing injected noise while maintaining enough intrinsic time for both learning and forgetting.*

## 4.1 General Results

The FSL (8) is established for the hard-learning regime where $s \leqslant 1 - 1/\beta$. We now show that an analogous FSL also holds in the general case. To state the result, we define $e_M(t) = \sum_{j=1}^M \lambda_j |\theta_j^*|^2 e^{-2\lambda_j t}, \mathcal{K}_M(t) = \sum_{j=1}^M \lambda_j^2 e^{-2\lambda_j t}$. One can verify that both functions exhibit power-law decay for $1 \lesssim t \lesssim M^\beta$:

$$e_M(t) \asymp t^{-s}, \qquad \mathcal{K}_M(t) \asymp t^{-(2-1/\beta)}, \qquad 1 \lesssim t \lesssim M^\beta. \tag{9}$$

Consequently, $e_\infty(t) \asymp e(t)$ and $\mathcal{K}_\infty(t) \asymp \mathcal{K}(t)$ for $t \geqslant 0$.

The following theorem provides a characterization of the loss dynamics for general case:

---

**Theorem 4.3** (Intrinsic-Time FSL, top-$M$ features, general label noise). *Suppose Assumption 4.1 holds. Let $\boldsymbol{\nu}_t$ denote the solution to the intrinsic-time SDE (7) with the top-$M$ features. Define $\mathcal{F}_M(t;\gamma) = e_M(t) + \int_0^t \mathcal{K}_M(t-z)[e_M(z)+\sigma^2]\gamma(z)\,\mathrm{d}z$. There exists a $c > 0$ such that for $0 \leqslant t \leqslant cM^\beta$, it holds that*

$$\mathbb{E}[\mathcal{R}(\boldsymbol{\nu}_t)] - \tfrac{1}{2}\sigma^2 \asymp M^{-s\beta} + \mathcal{F}_\infty(t;\gamma). \tag{10}$$

*For all $cM^\beta \leqslant t < \infty$, it holds that*

$$M^{-s\beta} + \mathcal{F}_M(t;\gamma) \lesssim \mathbb{E}[\mathcal{R}(\boldsymbol{\nu}_t)] - \tfrac{1}{2}\sigma^2 \lesssim M^{-s\beta} + \mathcal{F}_\infty(t;\gamma). \tag{11}$$

*Notably, the constants implicit in $\asymp, \lesssim$ are independent of the noise level $\sigma$.*

---

A proof sketch is provided in Appendix C.3. The above characterization is *uniform* with respect to the label-noise level $\sigma$, and holds for all $s > 0$ and $\beta > 1$. It asserts that the exact FSL relation (10) (i.e., the FSL (8)) remains valid up to the intrinsic time $t \leqslant cM^\beta =: t_M$. For later times $t > t_M$, although the FSL may no longer hold exactly, the loss dynamics remain controlled from both sides as in (11).

At the critical time $t_M$, we have $e_M(t_M) \asymp M^{-s\beta}$, indicating that signal learning has reached the approximation-error limit. Beyond this point, further training no longer improves the learned signal; instead, the dynamics become dominated by noise effects. Depending on the interaction between the stochastic gradient noise and the decaying learning rate, additional training may either inject more noise or dissipate it. Thus, it is a priori unclear whether the total error will significantly increase or decrease after $t_M$. The upper bound in (11) ensures that the overall loss remains well-controlled, analogous to the behavior of the infinite-width limit ($M = \infty$).

Nevertheless, an FSL may still hold for all $t \geqslant 0$, under suitably stronger conditions. In Theorem 4.2, we considered the setting with tasks satisfying $s \leqslant 1 - 1/\beta$. The following result shows that a similar characterization extends to general task difficulty with constant label noise.

**Theorem 4.4** (Intrinsic-Time FSL, top-$M$ features, constant label noise). *Under Assumption 4.1, suppose $\sigma \gtrsim 1$. Let $\boldsymbol{\nu}_t$ denote the solution to the intrinsic-time SDE (7) with the top-$M$ features. Then, for any $s > 0$ and all $t \geqslant 0$, $\mathbb{E}[\mathcal{R}(\boldsymbol{\nu}_t)] - \tfrac{1}{2}\sigma^2 \asymp M^{-s\beta} + \mathcal{F}_M(t;\gamma)$.*

Theorem 4.2 implies that the finite-$M$ functions $e_M$ and $\mathcal{K}_M$ can be replaced by their infinite-width counterparts $e_\infty$ and $\mathcal{K}_\infty$ in the hard-learning regime. The next result demonstrates that the same FSL characterization naturally extends to the noiseless case $\sigma = 0$.

**Theorem 4.5** (Intrinsic-Time FSL, top-$M$ features, zero label noise). *Suppose Assumption 4.1 holds and $\sigma = 0$. Let $\boldsymbol{\nu}_t$ denote the solution to the intrinsic-time SDE (7) with the top-$M$ features. If $s \in [0, 2 - 1/\beta]$, then for all $t \geqslant 0$, $\mathbb{E}[\mathcal{R}(\boldsymbol{\nu}_t)] \ \asymp \ M^{-s\beta} + e_M(t) + \int_0^t \mathcal{K}_M(t-z) \, e_M(z) \, \gamma(z) \, \mathrm{d}z$.*

**Random-$M$ features.** For the random-features case, the modified feature covariance matrix is $\widehat{\mathbf{H}} = \mathbf{W}\mathbf{H}\mathbf{W}^\top$, whose eigenvalues we denote by $\widehat{\lambda}_1 \geqslant \widehat{\lambda}_2 \geqslant \cdots \geqslant \widehat{\lambda}_M$. We similarly define $\widehat{e}_M(t) = \sum_{j=1}^M \widehat{\lambda}_j |\theta_j^*|^2 e^{-2\widehat{\lambda}_j t}$, $\widehat{\mathcal{K}}_M(t) = \sum_{j=1}^M \widehat{\lambda}_j^2 e^{-2\widehat{\lambda}_j t}$. The next theorem establishes that the same FSL also holds when the top-$M$ features are replaced by randomly selected features.

**Theorem 4.6** (Intrinsic-Time FSL, random-$M$ features). *Suppose Assumption 4.1 holds and $s \in (0, 1]$. Let $\boldsymbol{\nu}_t$ denote the solution to the intrinsic-time SDE (7) with the random-$M$ features. Then, with probability at least $1 - \exp(-\Omega(M))$ over the randomness of the projection matrix $\mathbf{W}$, the results of Theorems 4.3, 4.4, and 4.5 continue to hold, after replacing $e_M(\cdot)$ and $\mathcal{K}_M(\cdot)$ with their random-feature counterparts $\widehat{e}_M(\cdot)$ and $\widehat{\mathcal{K}}_M(\cdot)$, respectively.*

**Lemma 4.7.** *With probability at least $1 - \exp(-\Omega(M))$ over the randomness of the projection matrix $\mathbf{W}$, it holds that $\widehat{\lambda}_j \asymp \lambda_j \asymp j^{-\beta}$ for any $j \in [M]$.*

Theorem 4.6 and Lemma 4.7 together imply that when the task difficulty satisfies $s \leqslant 1$, training with random-$M$ features is similar to using the top-$M$ features, up to exponentially small probability. We emphasize, however, that for easier tasks with $s > 1$, the behaviors of random and top feature may diverge and we leave this for future investigation.

# 5 Learning Rate Schedules Impact Scaling Efficiency

Having established the general FSL, we now instantiate it under three representative LRSs—constant, exponential decay, and warmup–stable–decay (WSD)—to examine how schedule design influences scaling efficiency. All proofs can be found in Appendix E. For clarity, we make:

**Assumption 5.1.** Assume constant label noise $\sigma^2 \gtrsim 1$ and batch size $b(\tau) = B$ for all $\tau \geqslant 0$.

Under this assumption, given a physical-time LRS function $\varphi(\cdot)$, Theorem 4.4 implies that the FSL for $t \gtrsim 1$ simplifies to $\mathbb{E}[\mathcal{R}(\boldsymbol{\nu}_t)] - \frac{1}{2}\sigma^2 \asymp M^{-s\beta} + e_M(t) + \frac{\sigma^2}{B} \int_0^t \mathcal{K}_M(t-r)\varphi(T^{-1}(r)) \, \mathrm{d}r$.

Let $\mathcal{E}_K = \mathbb{E}[\mathcal{R}(\boldsymbol{\nu}_{Kh})] - \frac{1}{2}\sigma^2$ denote the expected excess risk after $K$ training steps. For each LRS, we derive concrete scaling laws describing how $\mathcal{E}_K$ scales with the model size $M$, the total step count $K$, as well as the LRS's hyperparameters. We then reinterpret these results from a resource-allocation perspective by optimizing under two canonical constraints: (i) the data-limited regime, where the total data size $D := BK$ is fixed; and (ii) the compute-limited regime [20], where the total compute $C := MD$ is fixed. For each regime, we further examine how the **optimally tuned hyperparameters** (e.g., the peak learning rate) should scale with increasing available resources. Finally, for clarity, we distinguish between two task regimes: an **easy-learning regime**, where $s \geqslant 1 - 1/\beta$, and a **hard-learning regime**, where $s < 1 - 1/\beta$.

## 5.1 Constant LRS

**Theorem 5.2** (Scaling law for constant LRS). *Under Assumption 5.1, we have $\mathcal{E}_K \asymp M^{-s\beta} + (\eta K)^{-s} + \frac{\eta}{B}\sigma^2$.*

Let $\gamma := \eta/B$ be the *effective learning rate*. Then, the scaling law can be rewritten as $\mathcal{E}_K \asymp M^{-s\beta} + (\gamma D)^{-s} + \gamma\sigma^2 =: h(\gamma, M, D)$, where the excess risk depends the learning rate via $\gamma = \eta/B$. This suggests that we should scale the learning rate linearly with respect to batch size (a.k.a. linear scaling rule) [26, 16, 40].

**Data-optimal scaling.** Clearly, this involves minimizing $h(\cdot)$ while keeping $D$ fixed. A straightforward calculation yields: $\gamma_{\mathrm{opt}} \asymp D^{-\frac{s}{s+1}}, M_{\mathrm{opt}} \gtrsim D^{\frac{1}{(1+s)\beta}}, \mathcal{E}_{\mathrm{opt}} \asymp D^{-\frac{s}{s+1}}$. Notably, both the best achievable excess risk $\mathcal{E}_{\mathrm{opt}}$ and optimal learning rate $\gamma_{\mathrm{opt}}$ depend exclusively on the task's relative

difficulty $s$. For a fixed target (fixed $\alpha$), a higher-capacity model (smaller $\beta$) gives a larger $s = \alpha/\beta$ and is therefore more data-efficient.

**Compute-optimal scaling.** This involves minimizing $h(\cdot)$ while keeping $C := DM$ fixed. The solution is summarized as follows, with the derivation deferred to Appendix E.1: $\gamma_{\mathrm{opt}} \asymp C^{-\frac{s\beta}{1+(s+1)\beta}}$, $M_{\mathrm{opt}} \asymp C^{\frac{1}{1+(s+1)\beta}}$, $D_{\mathrm{opt}} \asymp C^{\frac{(s+1)\beta}{1+(s+1)\beta}}$, $\mathcal{E}_{\mathrm{opt}} \asymp C^{-\frac{s\beta}{1+s\beta+\beta}}$.

This shows that the performance of the compute-optimal model improves with the total compute budget $C$ in a power law. For a fixed task ($\alpha = s\beta$ fixed), we have the following observations:

- Increasing model capacity ($\beta \downarrow$) enhances compute efficiency—the extra $\beta$ in the scaling exponent $\frac{s\beta}{1+s\beta+\beta}$ quantifies this gain. This explains a well-known empirical observation in LLM pre-training: Large models are more compute-efficient than small models [25, 20].
- The optimal learning rate $\gamma_{\mathrm{opt}}$ decreases as $C$ grows, and the compute-optimal allocation favors investing more in data than in model size—again consistent with current LLM pre-training practice [5, 55, 20].

Note that [8] also investigated compute-optimal scaling for constant LRS but assumed a fixed learning rate and no label noise. In contrast, we consider a more realistic scenario where the learning rate is optimally tuned and the irreducible risk is present, leading to a compute-optimal scaling law that matches empirical observations.

## 5.2 Exponential Decay LRS

For a given number of training steps $K$ [15, 65], an exponential decay (exp-decay) LRS is given by $\varphi(\tau) = a \exp(-\lambda\tau), \varphi(Kh) = b$, where $\lambda$ is chosen such that $\varphi(Kh) = b$. For brevity, we assume $h = 1$. Note that the hyperparameters $a$ and $b$ specify the peak and final learning rates, respectively.

**Theorem 5.3** (Scaling law for exp-decay LRS). *Under Assumption 5.1, we have* $\mathcal{E}_K \asymp M^{-s\beta} + T^{-s} + \sigma^2 \left( \frac{b}{B} + (a-b)\frac{\min\{M, T^{1/\beta}\}}{BT} \right)$, *where* $T = (a-b)K/\log(a/b)$ *is the total intrinsic time.*

Let $b = a/K$. Then the intrinsic time becomes $T = a(K-1)/\log K$, whereas a constant LRS with step size $\eta = a$ yields $T = aK$. Thus, exp-decay LRS drives the learning rate down to as small as $a/K$, yet sacrifices only a logarithmic factor of intrinsic time compared to the constant schedule.

**Data-optimal scaling.** Let $\gamma = a/B$ be the effective peak learning rate. By minimizing the right hand side of the exponential decay scaling law with respect to $a, b, K, B, M$ under the constraint $KB = D$ (see Appendix E.2), We obtain $M_{\mathrm{opt}} = \infty$ and

- For $s \geqslant 1 - \frac{1}{\beta}$, $\gamma_{\mathrm{opt}} \asymp (D/\log D)^{-\frac{1+s\beta-\beta}{1+s\beta}}$ and $\mathcal{E}_{\mathrm{opt}} \asymp (D/\log D)^{-\frac{s\beta}{s\beta+1}}$.
- For $s < 1 - \frac{1}{\beta}$, $\gamma_{\mathrm{opt}} \asymp 1$ and $\mathcal{E}_{\mathrm{opt}} \asymp (D/\log D)^{-s}$.

Compared with the constant LRS, exp-decay LRS achieves a strictly faster decay of the excess risk, justifying the importance of learning-rate decay in stochastic optimization.

**Compute-optimal scaling.** A straightforward calculation (see Appendix E.2) yields:

- For $s \geqslant 1 - \frac{1}{\beta}$, $\gamma_{\mathrm{opt}} \asymp \left(\frac{C}{\log C}\right)^{-\frac{1+s\beta-\beta}{2+s\beta}}, M_{\mathrm{opt}} \asymp \left(\frac{C}{\log C}\right)^{\frac{1}{2+s\beta}}, D_{\mathrm{opt}} \asymp C^{\frac{1+s\beta}{2+s\beta}}(\log C)^{\frac{1}{2+s\beta}}$, and $\mathcal{E}_{\mathrm{opt}} \asymp \left(\frac{C}{\log C}\right)^{-\frac{s\beta}{2+s\beta}}$.
- For $s < 1 - \frac{1}{\beta}$, $\gamma_{\mathrm{opt}} \asymp 1, M_{\mathrm{opt}} \asymp \left(\frac{C}{\log C}\right)^{\frac{1}{1+\beta}}, D_{\mathrm{opt}} \asymp C^{\frac{\beta}{1+\beta}}(\log C)^{\frac{1}{1+\beta}}, \mathcal{E}_{\mathrm{opt}} \asymp \left(\frac{C}{\log C}\right)^{-\frac{s\beta}{1+\beta}}$.

In the easy-learning regime, the excess-risk rate is determined solely by the intrinsic difficulty $\alpha = s\beta$; hence, increasing model capacity alone does not lead to asymptotic gains. The compute-optimal allocation consistently favors data over model and moreover, the optimal compute split depends solely on the task's intrinsic difficulty, with ratio $D_{\mathrm{opt}}/M_{\mathrm{opt}} \asymp C^{\alpha/(2+\alpha)}$ *decreasing* as the task becomes harder. This implies that, for harder tasks, one should allocate more compute to increasing model size. The optimal $\gamma_{\mathrm{opt}}$ decreases with the compute budget $C$, and for fixed $\alpha$, higher-capacity models ($\beta \downarrow$) require smaller $\gamma_{\mathrm{opt}}$.

In the hard-learning regime, data still dominates compute allocation, but now the optimal split depends only on model capacity, independent of the task difficulty. Moreover, the optimal maximal learning rate remains constant ($\gamma_{\mathrm{opt}} \asymp 1$). These results imply that a single, universal choice of compute split and learning rate suffices to attain optimal scaling across all tasks satisfying $s < 1 - 1/\beta$, greatly simplifying hyperparameter tuning. Finally, in this regime, higher-capacity models ($\beta \downarrow$) become strictly more compute-efficient, as evidenced by the excess-risk scaling exponent $-s\beta/(1+\beta)$.

## 5.3 WSD-like LRS

We lastly turn to consider a WSD-like LRS [69, 21], which comprises a $K_1$-step **stable phase** followed by a $K_2$-step **decay phase**, for a total $K = K_1 + K_2$ steps, given by

$$\varphi(\tau) = \begin{cases} a & , \text{ if } \tau \leqslant K_1 h; \\ a\exp(-\lambda(\tau - K_1 h)) & , \text{ if } \tau > K_1 h. \end{cases} \tag{12}$$

where $\lambda$ is chosen such that $\varphi(Kh) = b$. For brevity, we assume $h = 1$ and let $r = K_2/K$. This schedule is thus characterized by three hyperparameters: the peak learning rate $a$, the final learning rate $b$, and the decay proportion $r$, which controls the duration of decay-phase.

**Theorem 5.4** (Scaling law for WSD-like LRS). *Under Assumption 5.1, we have for the LRS* (12)*:*
$\mathcal{E}_K \asymp M^{-s\beta} + (T_1 + T_2)^{-s} + \sigma^2 \left( \frac{b}{B} + (a-b)\frac{\min\{M, T_2^{1/\beta}\}}{BT_2} \right)$, *where* $T_1 = aK_1$ *and* $T_2 = (a-b)K_2/\log(a/b)$ *denote the intrinsic training times of the stable and decay phases, respectively.*

We see that WSD-like LRS can leverage the initial stable phase to boost the intrinsic training time. For a decay proportion $r < 1$, we have $T = T_1 + T_2 \geqslant (1-r)Ka$, which far exceeds the the intrinsic time $T \asymp aK/\log K$ achieved by the pure exp-decay LRS. Consequently, WSD removes logarithmic factors in the full-batch GD term, without altering the noise term's order as long as $r > 0$. Building on this insights, we show that WSD can indeed improve the scaling efficiency, as detailed below.

**Data-optimal scaling.** Assuming $b = a/K$, we have $M_{\mathrm{opt}} = \infty$ and

- For $s \geqslant 1 - \frac{1}{\beta}$, $\gamma_{\mathrm{opt}} \asymp D^{-\frac{1+s\beta-\beta}{1+s\beta}}(\log D)^{\frac{\beta-1}{1+s\beta}}$, $r_{\mathrm{opt}} \in (0,1)$, $\mathcal{E}_{\mathrm{opt}} \asymp D^{-\frac{s\beta}{s\beta+1}}(\log D)^{\frac{s\beta-s}{1+s\beta}}$.

- For $s < 1 - \frac{1}{\beta}$, $\gamma_{\mathrm{opt}} \asymp 1$, $r_{\mathrm{opt}} \gtrsim D^{\frac{s\beta+1-\beta}{\beta-1}}\log D$, $\mathcal{E}_{\mathrm{opt}} \asymp D^{-s}$.

Compared with the exp-decay LRS, both regimes enjoy a logarithmic improvement in excess-risk decay. In particular, for the hard-learning regime, the logarithmic factor disappears. This improvement requires the **decay-phase duration only needs to scale sublinearly with** $D$, as indicated by $r_{\mathrm{opt}} \to 0$ as $D \to \infty$. This matches the WSD practice in LLM pre-training, where the decay phase typically occupies only 10%-20% of the total training duration. Moreover, our theory suggests that for harder tasks, the decay fraction can be reduced further to enhance compute efficiency.

**Compute-optimal scaling.** Analogous improvements hold in the compute-limited regime. Assuming $b = a/K$ and imposing the compute constraint $MD = C$, the compute-optimal satisfies:

- For $s \geqslant 1 - \frac{1}{\beta}$, $\gamma_{\mathrm{opt}} \asymp (\frac{C}{\log C})^{-\frac{1+s\beta-\beta}{2+s\beta}}$, $r_{\mathrm{opt}} \in (0,1)$, $M_{\mathrm{opt}} \asymp (\frac{C}{\log C})^{\frac{1}{2+s\beta}}$, $D_{\mathrm{opt}} \asymp C^{\frac{1+s\beta}{2+s\beta}}(\log C)^{\frac{1}{2+s\beta}}$, and $\mathcal{E}_{\mathrm{opt}} \asymp C^{-\frac{s\beta}{2+s\beta}}(\log C)^{\frac{s\beta-s}{2+s\beta}}$.

- For $s < 1 - \frac{1}{\beta}$, $\gamma_{\mathrm{opt}} \asymp 1$, $r_{\mathrm{opt}} \gtrsim D^{-\frac{\beta-1-s\beta}{\beta-1}}\log D$, $M_{\mathrm{opt}} \asymp C^{\frac{1}{1+\beta}}$, $D_{\mathrm{opt}} \asymp C^{\frac{\beta}{1+\beta}}$, $\mathcal{E}_{\mathrm{opt}} \asymp C^{-\frac{s\beta}{1+\beta}}$.

# 6 Experiments

## 6.1 Power-Law Kernel Regression

While the FSL is derived in the continuous-time limit, we now verify that it also accurately captures the loss dynamics and scaling behavior of the discrete-time SGD (3). Specifically, we consider the PLK regression with difficulty $s = 0.5$ and capacity $\beta = 4$, corresponding to a hard-learning regime and the results are shown in Figure 1.

**FSL accurately captures the loss dynamics of SGD.** Figure 1(left) compares the loss dynamics of SGD with the predictions of the FSL under three representative LRSs: cosine, WSD, and an unconventional cyclical schedule [56]. Across all cases, the FSL provides a remarkably accurate description of the SGD's loss evolution. Comparing the WSD and cosine schedules, we observe that the loss under WSD exhibits a slower decay during the stable phase but undergoes a much

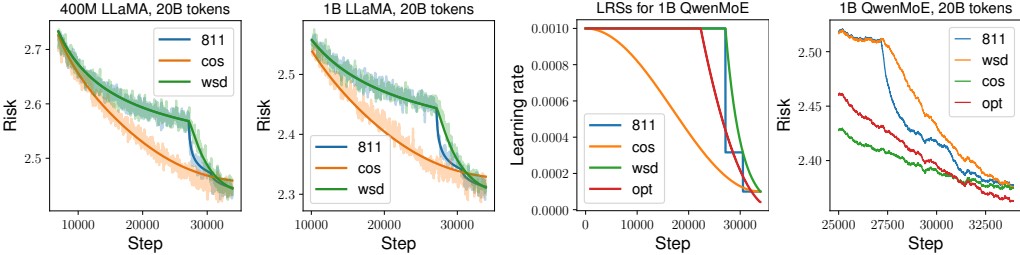

(a) Fitting and prediction using FSL.     (b) The FSL-optimal LRS and its performance

Figure 2: **Experiment on LLMs. (a)** Fitting and predictive accuracy of the FSL on dense LLaMA models. **(b)** Left: comparison of various LRSs. Right: loss trajectories of the FSL-optimal schedule versus baseline LRSs on a 1B QwenMoE model.

sharper drop once the decay phase begins, ultimately yielding a lower final loss. This seemingly counterintuitive two-phase dynamical behavior of WSD aligns well with empirical observations in practical LLM pre-training [21, 64, 58].

**FSL predicts the scaling behavior of SGD.** Figure 1(right) further validates the scaling laws derived in Section 5 for the three canonical LRSs—constant, exponential, and WSD-like (12). The results show that the final-step loss of SGD closely follows the theoretical predictions of FSL. Among these schedules, WSD yields the best scaling performance, followed by exponential decay, while the constant schedule performs the worst. More experiment details and additional results experiments with varying $(s, \beta)$ and other LRSs are provided in Appendix C.1, and exhibit consistent behaviors.

### 6.2 LLM Pre-training

We now evaluate the practical utility of FSL as a surrogate model for capturing the loss dynamics of LLM pre-training. Specifically, three popular LRSs: cosine, WSD, and the 8-1-1 [5] are considered; see Figure 2b(left) for a visualization. In the 8-1-1 LRS, the learning rate is reduced by a factor $\sqrt{10}$ at 80% and 90% of the total token budget, yielding a final value that is $0.1$ times the peak learning rate. For more experiment details, we refer to Appendix C.2.

**FSL accurately fit and predict loss curves.** We first quantify the descriptive and predictive power of FSL. Following the protocol of [59] and [38], we restrict attention to the post-warmup portion of the loss trajectory. Two Llama [60] models (400 M and 1 B) are trained on 20 B tokens under the three LRSs. For each model we (i) fit the FSL parameters on the loss curve obtained using the 8-1-1 LRS and (ii) deploy the fitted FSL to *predict* the loss curves of the cosine and WSD schedules. Figure 2a demonstrates that FSL not only fits the 8-1-1 trajectory accurately but also generalizes reliably to the unseen WSD and cosine schedules for both model sizes.

**The FSL-optimal LRS is WSD-like.** We next leverage the fitted FSL to *design* improved LRSs. Specifically, we numerically minimize the final-step loss over the space of LRSs using the fitted FSL. This experiment employs a 1B-parameter QwenMoE model [68], trained on 20B tokens using the same three LRSs. We fit the FSL using the trajectory from the 8-1-1 LRS and numerically solve for the FSL-optimal LRS. The model is then trained under this FSL-optimal LRS, using the same compute budget, and compared against the baseline LRSs. Figure 2b(left) shows that surprisingly, the FSL-optimal LRS is WSD-like and the decay phase drives the learning rate far below the conventional $0.1\eta_{\max}$ threshold. This echos recent empirical recommendations by [4, 17]. Furthermore, Figure 2b(right) demonstrates that the FSL-optimal schedule yields a strictly lower final loss than all baselines, substantiating its practical relevance. Taken together, these results suggest that FSL is a faithful surrogate for studying LLM training dynamics and a principled tool for interpreting and designing LRSs in large-scale pre-training.

## 7 Conclusion

In this paper, we present a systematic study of how LRS shapes the loss dynamics in kernel regression. Specifically, we establish a novel functional-level scaling law, which precisely characterizes the loss dynamics of SGD for general learning LRSs. The utility of our FSL is demonstrated through detailed analyses of three widely used LRSs, providing theoretical justification for several prevailing practices in LLM pre-training—most notably, offering an explanation for the effectiveness of the empirically popular but previously less-understood WSD schedules.

## Acknowledgement

Lei Wu is supported by the National Natural Science Foundation of China (NSFC12522120, NSFC92470122, and NSFC12288101). Binghui Li is supported by the Elite Ph.D. Program in Applied Mathematics at Peking University. We are grateful to Kaifeng Lyu, Kairong Luo, and Haodong Wen for generously sharing their work [38], which greatly inspired this study. We also thank Tingkai Yan, Yuhao Liu, Yunze Wu, and Zean Xu for many helpful discussions, and the anonymous reviewers for their valuable feedback.

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

# Appendix

## Table of Contents

## A  Miscellanea

### A.1  Empirical Fitting of LLM Pre-training Loss Trajectory

The Chinchilla Law [20] describes the final-step loss $\mathcal{L}$ as follows:

$$L(M, D) = L_0 + A_1 M^{-\kappa_1} + A_2 D^{-\kappa_2},$$

where $L_0$, $A_1$, $A_2$, $\kappa_1$, and $\kappa_2$ are constants, and $D$ and $M$ represent the amount of training data (tokens) and model size (number of parameters), respectively.

Later, [59] proposed the **Momentum Law**, a heuristic rule designed to capture the full loss trajectory. Given a learning rate schedule $\boldsymbol{\eta} := \{\eta_j\}_j$, the loss at the $k$-th step is modeled as

$$\mathcal{L}_k(\boldsymbol{\eta}) = L_0 + A S_1^{-\kappa} - C S_2,$$

where

$$S_1 = \sum_{i=1}^{k} \eta_i, \quad S_2 = \sum_{i=2}^{k} \sum_{j=2}^{i} (\eta_{j-1} - \eta_j) \lambda^{i-j}.$$

Here $L_0$, $A$, $C$, and $\kappa$ are constants, and $\lambda \in (0, 1)$ is a hyperparameter representing the decay factor for learning rate annealing, which typically ranges from $0.99$ to $0.999$.

Subsequently, [38] proposed the **Multi-Power Law** (MPL), which replaces the $S_2$ in the Momentum Law with additional power laws to better capture the progressive loss reduction induced by learning-rate decay. Specifically, the MPL takes the following form:

$$\mathcal{L}_k(\boldsymbol{\eta}) = L_0 + AS_1^{-\kappa} - \mathrm{LD}(k), \tag{13}$$

where

$$\mathrm{LD}(k) := C\sum_{i=2}^{k}(\eta_{i-1} - \eta_i)G(\eta_i^{-\kappa'}S_i), \quad S_i := \sum_{j=1}^{i}\eta_j, \quad G(x) := 1 - (C'x + 1)^{-\kappa''}.$$

Here $L_0, A, C, C', \kappa, \kappa', \kappa''$ are constants.

## A.2 Popular Learning Rate Schedules

Here, we introduce some widely used LRSs in the context of LLM pre-training.

- **Cosine Schedule [37].** The schedule is given by $\eta_k = \frac{1+\rho}{2}\eta_{\max} + \frac{1-\rho}{2}\eta_{\max}\cos(\frac{k-1}{K-1})$, where $\eta_{\max}$ is the maximum learning rate and the hyper-parameter $\rho$ is usually chosen as 0.1 such that the minimum learning rate is $\eta_{\max}/10$ [60].

- **Warmup-Stable-Decay (WSD) Schedule [69, 21, 17].** The schedule consists of three phases: a warm-up phase of $K_{\mathrm{warm\text{-}up}}$ steps, followed by a stable phase maintaining the learning rate $\eta_k = \eta_{\max}$, and finally a decay phase governed by $\eta_k = h(k - K_{\mathrm{stable}})\eta_{\max}$ for $K_{\mathrm{stable}} \leqslant k \leqslant K$, where $K_{\mathrm{stable}}$ represents the total duration of the first two phases. Here, the decay function $h(\cdot) \in (0, 1)$ can be linear or exponential.

- **Multi-Step Schedule [5].** The entire schedule is divided into $S$ stages, i.e., $[K_0, K_1] \cup [K_1, K_2] \cup \cdots \cup [K_{S-1}, K_S] = [0, K]$, where $0 = K_0 < K_1 < \cdots < K_S = K$. The schedule satisfies that $\eta_k = \eta_{K_i}$ for $K_{i-1} < k \leqslant K_i$ $(1 \leqslant i \leqslant S)$. In our LLM experiments, we consider a 8-1-1 LRS, corresponding to the case where $S = 3$ with $\eta_{K_1} = \eta_{\max}, \eta_{K_2} = \eta_{\max}/\sqrt{10}$, and $\eta_{K_3} = \eta_{\max}/10$, and $K_1 = 0.8K, K_2 = 0.9K$.

## A.3 Connections to Kernel Regression

In this section, we explain how our setup in Section 2 are equivalent to learning with kernels.

**Definition A.1** (Positive semidefinite (PSD) kernel). A function $K : \mathcal{X} \times \mathcal{X} \to \mathbb{R}$ is called a *continuous positive semidefinite (PSD) kernel* if it satisfies:

- Symmetry: $K(\mathbf{x}, \mathbf{x}') = K(\mathbf{x}', \mathbf{x})$ for all $\mathbf{x}, \mathbf{x}' \in \mathcal{X}$;

- Positive semidefiniteness: for any $\mathbf{x}_1, \ldots, \mathbf{x}_n \in \mathcal{X}$ and $a_1, \ldots, a_n \in \mathbb{R}$,

$$\sum_{i,j=1}^{n} a_i a_j K(\mathbf{x}_i, \mathbf{x}_j) \geqslant 0.$$

**Definition A.2** (Reproducing kernel Hilbert space (RKHS)). Given a kernel $K$, the *reproducing kernel Hilbert space* $\mathcal{H}_K$ associated with $K$ is a Hilbert space of functions $f : \mathcal{X} \to \mathbb{R}$ such that

$$\langle f, K(\mathbf{x}, \cdot)\rangle_{\mathcal{H}_K} = f(\mathbf{x}), \quad \forall f \in \mathcal{H}_K, \ \mathbf{x} \in \mathcal{X}.$$

Kernel methods learn functions from a hypothesis space defined by the associated RKHS. For instance, kernel ridge regression gives estimator:

$$\hat{f}_\lambda = \arg\min_{f \in \mathcal{H}_K} \frac{1}{n}\sum_{i=1}^{n}(f(\mathbf{x}_i) - y_i)^2 + \lambda\|f\|_{\mathcal{H}_K}^2.$$

Hence, model capacity is determined by the size of the RKHS $\mathcal{H}_K$.

Let $\mathcal{D}$ be the input distribution. Given a kernel $K$, define the associated integral operator $\mathcal{T}_K : L^2(\mathcal{D}) \to L^2(\mathcal{D})$ by

$$\mathcal{T}_K f(\cdot) = \mathbb{E}_{\mathbf{x} \sim \mathcal{D}}[K(\cdot, \mathbf{x})f(\mathbf{x})].$$

By assuming $\mathbb{E}_{\mathbf{x} \sim \mathcal{D}}[K(\mathbf{x}, \mathbf{x})] < \infty$, the operator $\mathcal{T}_K$ is compact (Mercer's theorem) and consequently, the kernel admits the following eigenvalue decomposition

$$K(\mathbf{x}, \mathbf{x}') = \sum_{j=1}^{\infty} \lambda_j e_j(\mathbf{x}) e_j(\mathbf{x}'),$$

where $\{\lambda_j\}_{j=1}^{\infty}$ and $\{e_j\}_{j=1}^{\infty}$ denotes the eigenvalues and eigenfunctions, respectively. Moreover, $\langle e_i, e_j \rangle_{L^2(\mathcal{D})} = \delta_{i,j}$, i.e., the eigenfunctions form an orthonormal basis of $L^2(\mathcal{D})$.

Using the spectral decomposition, the RKHS admits the following representation:

$$\mathcal{H}_K = \left\{ \sum_{j=1}^{\infty} a_j e_j : \sum_{j=1}^{\infty} \frac{a_j^2}{\lambda_j} < \infty \right\}.$$

To better quantify the smoothness of functions, we often consider the interpolation space $\mathcal{H}_K^s$ with $s \geqslant 0$, defined as

$$\mathcal{H}_K^s = \left\{ \sum_{j=1}^{\infty} a_j e_j : \sum_{j=1}^{\infty} \frac{a_j^2}{\lambda_j^s} < \infty \right\}.$$

Clearly, $\mathcal{H}_K^1 = \mathcal{H}_K$, and

$$\mathcal{H}_K^{s_1} \subset \mathcal{H}_K^{s_2}, \qquad \forall\, s_1 > s_2 \geqslant 0.$$

Hence, the index $s$ characterizes the smoothness of a function relative to the chosen kernel.

In the analysis of kernel methods, the following conditions are commonly used to describe the smoothness of the target function and the capacity of the kernel, respectively.

**Assumption A.3** (Source condition). There exists some $s > 0$ such that $f^* \in \mathcal{H}_K^s$.

**Assumption A.4** (Capacity condition). There exists some $\beta > 1$ such that $\lambda_j \asymp j^{-\beta}$.

These conditions yield the following interpretation:

- A smaller $s$ indicates that the target function $f^*$ belongs to a larger space, corresponding to a more difficult learning problem.

- A smaller $\beta$ implies a slower eigenvalue decay, meaning a richer hypothesis space $\mathcal{H}_K$ and thus higher model capacity.

Our formulation in Section 2 is equivalent to the above setting, but expressed in terms of the feature map $\phi$. Under Assumption 2.2, we have

$$K_\phi(\mathbf{x}, \mathbf{x}') = \sum_{j=1}^{N} \phi_j(\mathbf{x}) \phi_j(\mathbf{x}') = \sum_{j=1}^{N} \lambda_j \, \widehat{\phi}_j(\mathbf{x}) \widehat{\phi}_j(\mathbf{x}').$$

In this case, Assumption 2.3 corresponds exactly to the above capacity condition, while the task-difficulty assumption in Section 2 can be viewed as a power-law version of the source condition. Specifically, under Assumption 2.4,

$$f^* = \sum_{j=1}^{N} \theta_j^* \phi_j = \sum_{j=1}^{N} j^{-1/2} \lambda_j^s \, \widehat{\phi}_j = \sum_{j=1}^{N} a_j \, \widehat{\phi}_j.$$

Hence, for any arbitrarily small $\delta \in (0, 1)$, we have $f^* \in \mathcal{H}_{K_\phi}^{s-\delta}$, since

$$\sum_{j=1}^{N} \frac{a_j^2}{\lambda_j^{s-\delta}} = \sum_{j=1}^{N} j^{-1-\beta(s-\delta)} < \infty.$$

## A.4 The SDE Modeling

**The physical-time SDE.** In our setup, the SGD update can be written as

$$\mathbf{v}_{k+1} = \mathbf{v}_k - \varphi_k \nabla \mathcal{R}(\mathbf{v}_k) h - \varphi_k h \boldsymbol{\xi}_k.$$

The term $\boldsymbol{\xi}_k$ is the gradient noise, whose covariance is $\frac{1}{B_k}\Sigma(\mathbf{v}_k)$. By assuming the gradient noise to be Gaussian, the SGD becomes

$$\mathbf{v}_{k+1} - \mathbf{v}_k = -\varphi_k \nabla \mathcal{R}(\mathbf{v}_k) h + \varphi_k \sqrt{h} \, \mathcal{N}\left(0, \frac{h}{B_k}\Sigma(\mathbf{v}_k)\right).$$

It is exactly the Euler–Maruyama discretization of the Itô-type SDE:

$$d\bar{\mathbf{v}}_\tau = -\varphi(\tau)\nabla \mathcal{R}(\bar{\mathbf{v}}_\tau)\, dt + \varphi(\tau)\sqrt{\frac{h}{b(\tau)}\Sigma(\bar{\mathbf{v}}_\tau)}\, d\mathbf{B}_\tau,$$

where $\mathbf{B}_\tau \in \mathbb{R}^M$ denotes the $M$-dimensional Brownian motion, and $\varphi(\cdot)$, $b(\cdot)$ are the continuous version of LRS function and batch-size schedule function, respectively.

**The intrinsic-time SDE.** Intuitively, the discrete update (4) can be viewed as the Euler–Maruyama discretization of SDE (7) on the *non-uniform* grid $\{t_k = \sum_{j=0}^k \eta_j\}_{k\in\mathbb{N}}$ where the effective step size is $\Delta t_k = \eta_k$:

$$\mathbf{v}_{k+1} - \mathbf{v}_k = -\nabla \mathcal{R}(\mathbf{v}_k)(t_{k+1} - t_k) - \sqrt{t_{k+1} - t_k}\, \mathcal{N}\left(0, \frac{\eta_k}{B_k}\Sigma(\mathbf{v}_k)\right).$$

## A.5 The Emergence of Power Laws

We illustrate how the power law emerges in our setting from a multi-task learning viewpoint. For brevity, consider the case of the top-$M$ features and an infinitesimal learning rate, where the SDE (7) reduces to the gradient flow ODE: $d\boldsymbol{\nu}_t = -\mathbf{W}^\top \mathbf{W} \mathbf{H} \boldsymbol{\nu}_t\, dt$. Noting that $\mathbf{W}^\top \mathbf{W} \mathbf{H} = \text{diag}\{\lambda_1, \lambda_2, \cdots, \lambda_M, 0, \cdots, 0\}$ is diagonal, consequently the ODE is solvable and gives the following expression of the excess risk:

$$\mathcal{R}(\boldsymbol{\nu}_t) - \tfrac{1}{2}\sigma^2 \approx \underbrace{\sum_{j=1}^M \lambda_j |\theta_j^*|^2 e^{-2\lambda_j t} + \int_0^t}_{\text{learned sub-tasks}} + \underbrace{\sum_{j=M+1}^N \lambda_j |\theta_j^*|^2}_{\text{unlearned sub-tasks}}.$$

Intuitively, we can view the learning of each eigenfunction as a sub-task. Due to the limited model size, student model can at most learn the top-$M$ eigenfunctions.

(i) **Intrinsic-time power law.** For each sub-task, the sub-task risk converges exponentially w.r.t. the intrinsic time $t$. However, owing to the power-law structure of $\lambda_j, \theta_j^*$, the total multi-task risk exhibits a power-law decay for sufficiently large $M$ due to $\sum_{j=1}^M \lambda_j |\theta_j^*|^2 e^{-2\lambda_j t} \approx \int_0^1 u^{s-1} e^{-2ut}\, du \approx \frac{1}{t^s}$ if $M \gg 1$.

(ii) **Model-size power law.** Approximation error accounts for total risk of the $N-M$ unlearned sub-tasks, which follows a power-law decay due to $\sum_{j=M+1}^N \lambda_j |\theta_j^*|^2 \approx M^{-s\beta}$ if $N-M \gg 1$.

> **Summary.** The emergence of power law arises from the accumulation effect, requiring both the number of learned tasks and unlearned tasks to be large (ideally infinite).

# B Related Work

**Theoretical explanation of scaling laws.** Among the growing body of work seeking to theoretically explain scaling laws [54, 22, 39, 63, 23, 42, 43, 2, 14, 3, 7, 35, 50, 8, 46], the most closely related are [7, 50, 8, 35], which also analyze PLK regression (often written in the equivalent linear-regression form). Specifically, [7] studies gradient flow, [50, 8] analyze SGD with a constant LRS, and [35]

considers an exponential-decay LRS. In contrast, we establish a unified scaling law applicable to general LRSs, which not only recovers these prior results as special cases but also substantially extends them by capturing the loss dynamics rather than only the final-step loss. This unification is enabled by introducing the key notion of *intrinsic time*, which more faithfully captures the effective training progress than the raw number of training steps.

**Predicting loss trajectories in LLM pre-training.** Recent empirical studies [59, 38] have shown that the entire loss trajectory of LLM pre-training—not merely the final loss—can be accurately captured by suitable scaling relations. A detailed description of the corresponding fitting procedures is provided in Appendix A.1. Our theory offers a theoretical explanation for these empirical findings. Interestingly, the *multi-power-law* (MPL) model proposed by [38] is closely connected to our FSL: through an integration-by-parts transformation, the FSL expression can be recast into a form that is nearly equivalent to the MPL formulation (see Appendix C.2).

**Warmup-Stable-Decay (WSD) LRS.** A WSD schedule [69, 21] maintains a constant learning rate for a long stable phase, followed by a learning rate decay only near the end of training. Although unconventional, WSD has become popular in LLM pre-training [21, 17] and is already deployed in training industry-scale LLMs such as `DeepSeek-V3` [36] and `Kimi-K2` [58]. Yet its mechanism remains poorly understood. While recent works [64, 53] offer partial insights, we show—perhaps surprisingly—that even *quadratic optimization*, corresponding to a kernel regression problem, already reproduces the essential advantage of WSD. Furthermore, we quantify this advantage through explicit comparisons of scaling efficiency against constant and exponential-decay schedules.

## C   Experiment Details and Additional Results

In this section, we present the details of our experiments as well as additional results.

### C.1   Power-Law Kernel Regression

**Physical-time FSL.** The FSL (10) is presented in terms of intrinsic time, but in practice, it is often more convenient to use physical time (training steps). By a suitable change of times, after $\tau$ steps (equivalently, $\tau/h$ discrete steps), the FSL maintains the form (10), with adjustments:

$$t^{-s} = T(\tau)^{-s},$$

$$\mathcal{N}(\varphi, b) = \int_0^\tau \mathcal{K}(T(\tau) - T(u)) \left(e(T(u)) + \sigma^2\right) \frac{h\varphi(u)^2}{b(u)} \, \mathrm{d}u.$$

**Fitting FSL on SGD Average-Risks.** To validate that the Functional Scaling law (FSL) can accurately capture the risk curve of SGD, we conducted a series of SGD experiments under different configurations of $s$ and $\beta$. Subsequently, we fitted the FSL to these risk curves. Our results demonstrate that FSL indeed provides a close fit to the SGD trajectories.

In each experiment, we adopt a PLKR configuration with $M = N = 128$, $\sigma = 3$ and employ the top-$M$ projection matrix, thereby eliminating the approximation error term $M^{-s\beta}$. We explore a range of values for $s \in [0.5, 1]$ and $\beta \in [1.5, 5]$, encompassing both easy- ($s \geqslant 1 - 1/\beta$) and hard-learning ($s < 1 - 1/\beta$) regimes. For each parameter configuration, we execute 200 independent SGD runs with a batch size of 1 over 10,000 steps. The resulting average trajectory across these runs serves as the fitting target. The FSL fitting is performed using the physical-time FSL formulation.

$$\mathcal{E}_k = c_1 T(k)^{-s} + c_2 \sum_{i=1}^{k} \mathcal{K}(T(k) - T(i)) e(T(i)) \eta_i^2 + c_3 \sigma^2 \sum_{i=1}^{k} \mathcal{K}(T(k) - T(i)) \eta_i^2,$$

where $c_1$, $c_2$, $c_3$ are constants to fit, $\{\eta_i\}_{i=1}^{k}$ is the learning rate schedule, and $T(i) = \sum_{j=1}^{i} \eta_j$.

When fitting the SGD trajectory, we minimize the mean squared error (MSE) between the empirical risk trajectory of SGD (without the irreducible risk $\frac{\sigma^2}{2}$), denoted by $\mathcal{E}_{\mathrm{SGD}}(k)$, and the theoretical prediction from FSL, $\mathcal{E}_k$. Formally, we solve the following optimization problem:

$$\min_{c_1, c_2, c_3} \frac{1}{K} \sum_{k=1}^{K} \left(\mathcal{E}_{\mathrm{SGD}}(k) - \mathcal{E}_k\right)^2,$$

where $K$ represents the total number of training steps. This minimization is performed using ordinary least squares (OLS), with the integrals in the FSL expression $\mathcal{E}_k$ evaluated numerically via quadrature methods.

We display the learning rate schedules (LRSs) used in the SGD experiments in the top-left panel of Figure 3. Complementing Figure 1 (middle and right), additional experimental results for various values of $s$ and $\beta$ are presented in Figure 3.

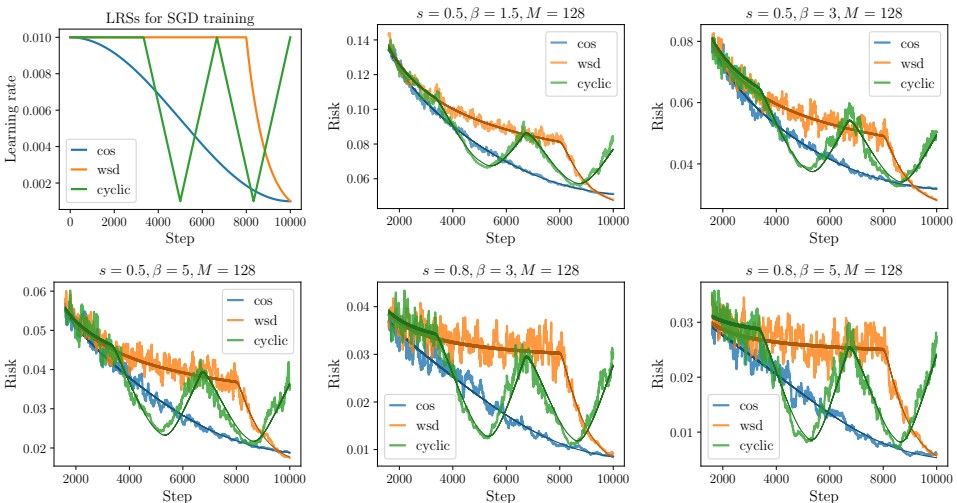

Figure 3: **Fitting results of FSL on SGD trajectories.** The shaded curves are the average over 200 independent SGD runs, while the solid curves show the predictions of FSL.

**Scaling law experiments.** These experiments are designed to evaluate the correctness of the scaling laws predicted by our analytical analysis. To this end, we conduct two complementary sets of experiments:

- **FSL experiments.** This experiment is intended to validate the theoretical predictions derived from the FSL. We compute the predicted risk by numerically discretizing the FSL (10), with all untracked constants set to 1. For each LRS, following the theoretical analysis, we set $\eta_{\max} = 0.05D^{-r}$, where $r = s/(1+s)$ for the constant learning rate schedule, and $r = 1$ for exponential decay and WSD schedules. We fix the batch size to $B = 1$; thus, for each data budget $D$, we compute the intrinsic time and evaluate the final-step loss using the discretized FSL.

- **SGD experiments.** This experiment serves to assess whether the scaling behavior predicted by our continuous-time FSL faithfully captures that of discrete-time SGD. We simulate stochastic gradient descent (SGD) with 200 independent trajectories and a fixed batch size $B = 1$. For each data budget $D$, the maximum learning rate is set as $\eta_{\max} = 0.05D^{-r}$, using the same theoretical values of $r$ as in the FSL experiments. We run SGD for $D$ steps under each corresponding LRS and record the final-step excess risk.

### C.2 LLM pre-training

**Practical FSL Ansatz for LLM pre-training** In this section, in order to fit real LLM pre-training loss curves, we will derive an approximation form of the FSL in Theorem 4.2.

First, by the physical-time for of the FSL (10) with $h = 1$ and $B(u) \equiv B$, we have

$$\mathcal{E}_k \approx \frac{1}{T(k)^s} + M^{-s\beta} + \frac{1}{B} \int_0^k \mathcal{K}(T(k) - T(u))(\sigma^2 + e(T(u))) \cdot \varphi(u)^2 \, \mathrm{d}u.$$

Here we focus on the integral term. Since $\varphi(u) = \int_0^u \varphi'(r) \, \mathrm{d}r + \varphi(0)$, and that

$$\int_0^k \mathcal{K}(T(k) - T(u))(\sigma^2 + e(T(k)) \cdot \varphi(u)\varphi(0) \, \mathrm{d}u = \varphi(0) \int_0^{T(k)} \mathcal{K}(T(k) - t)(\sigma^2 + e(t)) \, \mathrm{d}t. \quad (14)$$

Note that this is exactly the SGD noise term at the constant LRS $\eta(0)$ for a total intrinsic-time $T(\tau)$. By results of constant LRS (as seen in the proof of Theorem E.1), we have

$$(14) \asymp \varphi(0)(\sigma^2 + e(T(k))).$$

As $\varphi(0) \lesssim 1$, we have

$$\mathcal{E}_k \asymp \frac{1}{T(k)^s} + M^{-s\beta} - \text{LRD}(k), \tag{15}$$

where

$$\begin{aligned}
\text{LRD}(k) &:= -\frac{1}{B} \int_0^k \mathcal{K}(T(k) - T(u))(\sigma^2 + e(T(u)))\varphi(u) \int_0^u \varphi'(r) \, dr \, du \\
&= -\frac{1}{B} \int_0^k \varphi'(r) \int_r^k \mathcal{K}(T(k) - T(u))(\sigma^2 + e(T(u)))\varphi(u) \, du \, dr \\
&= -\frac{1}{B} \int_0^k \varphi'(r) \int_{T(r)}^{T(k)} \mathcal{K}(T(k) - t)(\sigma^2 + e(t)) \, dt \, dr.
\end{aligned}$$

We discretize the outer integral at integer nodes $r = 0, 1, \ldots, k$,

$$\text{LRD}(k) \approx \frac{1}{B} \sum_{i=1}^k (\eta_{i-1} - \eta_i) \int_{T(i)}^{T(k)} \mathcal{K}(T(k) - t)(\sigma^2 + e(t)) \, dt.$$

By the integral mean value theorem, we can take $(\sigma^2 + e(t))$ outside the integral, which gives

$$\text{LRD}(k) \approx \frac{1}{B} \sum_{i=1}^k (\eta_{i-1} - \eta_i)(\sigma^2 + e(\xi_i)) \int_{T(i)}^{T(k)} \mathcal{K}(T(k) - t) \, dt,$$

where $\xi_i \in [T(i), T(k)]$. Now since

$$\int_{T(i)}^{T(k)} \mathcal{K}(T(k) - t) \, dt \approx \int_{T(i)}^{T(k)} \frac{1}{(1 + ct)^{2 - 1/\beta}} \, dt \, du \asymp 1 - \frac{1}{(1 + c(T(k) - T(i))^{1 - 1/\beta}},$$

we then further simplify it as

$$\text{LRD}(k) \approx \frac{1}{B} \sum_{i=1}^k (\eta_{i-1} - \eta_i)(\sigma^2 + e(T(i)))(1 - (1 + c(T(k) - T(i)))^{-\gamma}).$$

Here, we approximate $\xi_i$ as $T(i)$ and introduce a new parameter $\gamma$ to replace $1 - \frac{1}{\beta}$ for simplicity.

Therefore, combining with (15), when the batch size $B$ is fixed, after renaming some constants, the final discrete ansatz can be written as

$$\begin{aligned}
\mathcal{R}_k \approx {}& c_0 + \frac{c_1}{T(k)^s} + c_2 M^{-s\beta} \\
& - c_3 \sum_{i=1}^k (\eta_{i-1} - \eta_i) \left( c_4 + \frac{1}{T(i)^s} \right) \left( 1 - (1 + c_5(T(k) - T(i)))^{-\gamma} \right),
\end{aligned} \tag{16}$$

where $c_0, c_1, c_2, c_3, c_4, c_5, s, \beta, \gamma$ are constants to fit.

**Fitting the Practical FSL**   The objective of this experiment is to analyze and fit the loss function using our functional scaling law, by (16), since we do not explore the effect of varying the model size $M$ in our experiments, we drop the term $M^{-s\beta}$ and get

$$\mathcal{L}_\Theta(k) = L_0 + \frac{c_1}{T(k)^s} - \text{LRD}(k)$$

where $T(k) = \sum_{i=1}^k \eta_i$ and

$$\text{LRD}(k) := c_2 \sum_{i=1}^k (\eta_{i-1} - \eta_i) \left( c_3 + \frac{1}{T(i)^s} \right) \left( 1 - \frac{1}{(1 + c_4(T(k) - T(i)))^\gamma} \right),$$

and $\Theta = (L_0, c_1, c_2, c_3, c_4, s, \gamma)$.

Following [59], we utilize the Huber loss as the objective function.

$$\min_{\Theta} \sum_{k=1}^{K} \mathrm{Huber}_{\delta} \left( \log \mathcal{L}_{\Theta}(k) - \log \mathcal{L}_{\mathrm{gt}}(k) \right),$$

where $\delta = 1 \times 10^{-3}$, $\mathcal{L}_{\mathrm{gt}}$ denotes the ground truth of the validation losses. We adopt the Adam optimizer, with a learning rate of $5 \times 10^{-2}$ for the index parameters in our law and $5 \times 10^{-3}$ for the coefficient or constant parameters. Each optimization takes over 10,000 steps.

We fit the law on the 400M model and 1B model trained with 20B tokens and an 8-1-1 LRS We then predict the loss curve for the 400M model and 1B model with cosine LRS and WSD LRS. The experiment result is present in Figure 2a.

**FSL-optimal LRS via numerical variation.** We propose to obtain a numerical optimal LRS by directly minimizing the final-step loss over the space of LRS using the fitted FSL, termed FSL-optimal LRS.

**Step 1: Fitting FSL.** Fit FSL on the loss curve of a 1B QwenMoE model trained on 20B tokens with batch size 288, maximum learning rate $\eta_0 = 0.001$, and the 8-1-1 scheduler over a total step of $K = 33907$ , following the same procedure described earlier.

**Step 2: Optimize LRS.** To improve optimization stability, we reparameterize the learning rate schedule by defining

$$\delta_i = \eta_i - \eta_{i+1}, \text{ for } i = 0, 1, \ldots, K - 1.$$

Then, the $i$-th step learning rate can be recovered by $\eta_i = \eta_0 - \sum_{k=0}^{i-1} \delta_k$, which defines a one-to-one correspondence between the learning rate schedule $\{\eta_i\}$ and $\{\delta_i\}$. The optimization problem is

$$\min_{\{\delta_i\}_{i=1}^{K}} \mathcal{L}_{\Theta}(\{\eta_i\}_{i=1}^{K}), \quad \text{subject to } \sum_{k=0}^{K-1} \delta_k \leqslant \eta_0, \ d\eta_i \geqslant 0, i = 0, 1, \ldots, K - 1. \quad (17)$$

To solve the above constraint optimization, we use the projected gradient descent (PGD) [10]. The learning rate of PGD is searched ranging from $1 \times 10^{-8}$ to $5 \times 10^{-10}$, and the optimization step number ranges from 50,000 to 100,000.

The resulting FSL-optimal LRS is presented in Figure 2b (left), where cosine, WSD, and 8-1-1 LRSs are also given for a comparison.

**Step 3: Evaluate our LRS.** We then evaluate the performance of the resulting FSL-optimal LRS, and the three LRSs in Figure 2b (left) are used as baseline. All comparisons are conducted on the same 1B QwenMoE model under identical training conditions: 33,907 total steps, batch size 288, and 20B training tokens. Full loss curves are shown in Figure 2b.

**Additional Experiments** We have further conducted ablation experiments with different model sizes and architectures, different total steps and different WSD schedules.

We validate our functional scaling law in models with various sizes, ranging from 100M to 1B, and diverse architectures including GPT-2 [52], LLaMA [60] and QwenMoE [68]. For each model, we first fit the FSL using the 8-1-1 LRS and subsequently employ it to predict the loss curve under a WSD LRS. Next we numerically solve the FSL-optimal LRS and empirically validate its efficacy by comparing the final pre-training loss against those obtained using other commonly adopted learning rate schedules. We present the results in Figure 4 for the 1B LLaMA dense model, Figure 5 for the 100M GPT-2 dense model. The consistent alignment between predicted and observed performance across architectures and sizes underscores the robustness and generalizability FSL.

We further validate the applicability of our functional scaling law (FSL) across varying training durations. Using a 100M LLaMA dense model, we conduct experiments with total training steps set to 17k, 34k, 68k, and 134k. As demonstrated in Figures 6 and 7, our FSL accurately models the loss trajectories across all evaluated step counts, confirming its robustness to different total training steps.

Finally, we conduct a comprehensive empirical comparison between our FSL-optimal learning rate schedule and various WSD baselines, examining different decay ratios and minimum learning rate

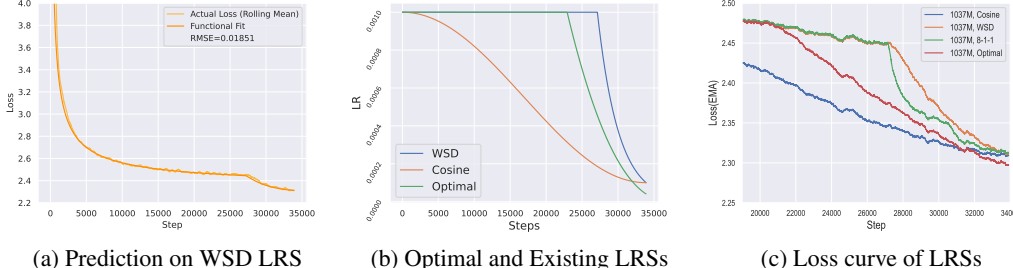

(a) Prediction on WSD LRS      (b) Optimal and Existing LRSs      (c) Loss curve of LRSs

Figure 4: **Experiment on the 1B LLaMA (dense) model.** Figure (a): We fit our functional scaling law on the loss curve of 1B LLaMA (dense) model with 20B tokens training data and 8-1-1 LRS. Figures (b)(c): The comparison on the 1B model between the optimal LRS, cosine LRS, WSD LRS with exponential decay and 8-1-1 LRS.

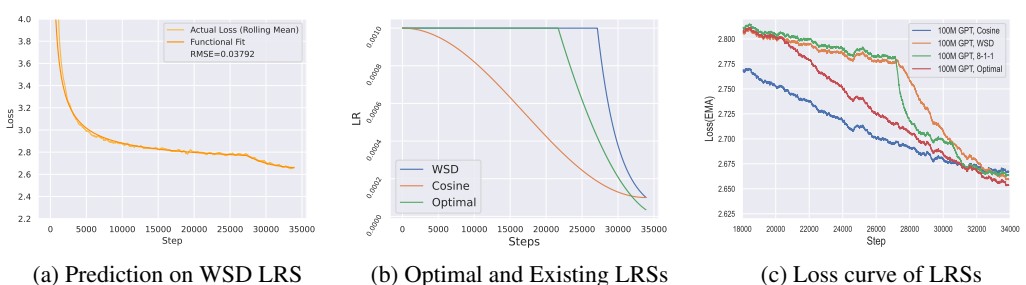

(a) Prediction on WSD LRS      (b) Optimal and Existing LRSs      (c) Loss curve of LRSs

Figure 5: **Experiment on the 100M GPT2 (dense) model.** Figure (a): We fit our functional scaling law on the loss curve of 100M GPT2 (dense) model with 20B tokens training data and 8-1-1 LRS. Figures (b)(c): The comparison on the 100M model between the optimal LRS, cosine LRS, WSD LRS with exponential decay and 8-1-1 LRS.

configurations. As evidenced by Figures 8 and 9, our FSL-optimal LRS consistently outperforms all WSD variants, achieving superior final pre-training loss across all experimental conditions. This systematic evaluation demonstrates both the effectiveness of our theoretically-derived schedule and its practical advantages over conventional heuristic approaches.

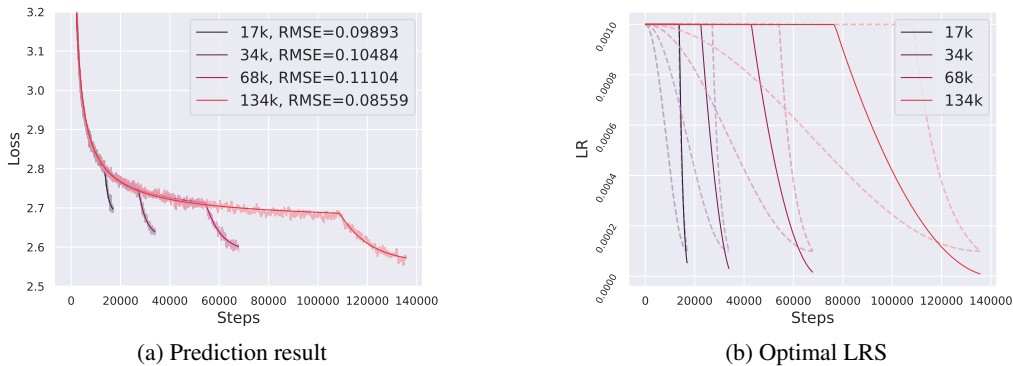

(a) Prediction result      (b) Optimal LRS

Figure 6: **Experiments with different total steps.** Figure (a): Fitted functional scaling laws on 100M LLaMA model with different total training steps 17k, 34k, 68k and 134k (corresponding to 10B, 20B, 40B and 80B tokens respectively). Figure (b): Optimal LRSs compared with cosine and WSD LRSs. The solid lines are optimal LRSs, and the dashed lines are cosine/WSD LRSs.

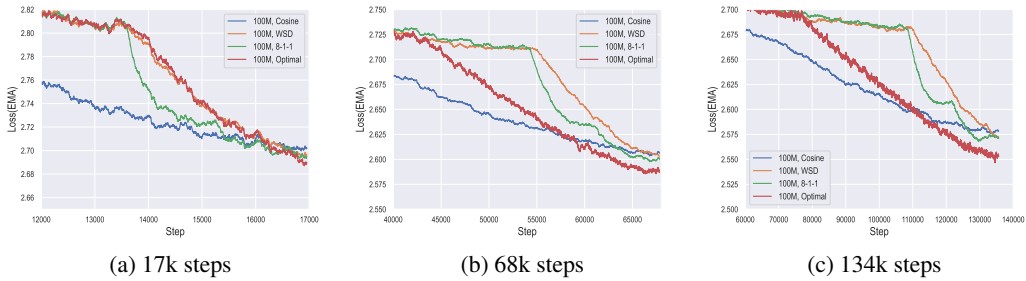

(a) 17k steps      (b) 68k steps      (c) 134k steps

Figure 7: **Experiments with different total steps.** We compare loss curves of existing LRSs and optimal LRS on the 100M LLaMA model with different total training steps 17k, 68k and 134k.

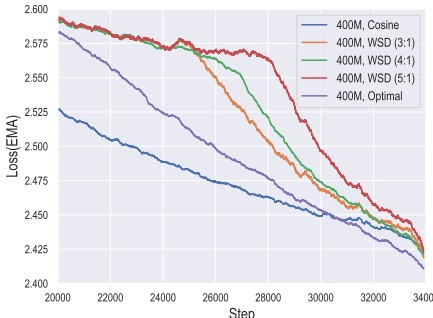

Figure 8: **WSD with different decay ratios:** We train a 400M LLaMA (dense) model with 20B tokens of training data and WSD LRSs with the ratios between stable time and decay time of 3:1, 4:1, and 5:1. All WSD LRSs exhibit a final loss similar to that of the Cosine LRS, and the optimal LRS derived from our functional scaling law outperforms all other LRSs by a loss gap of approximately 0.01.

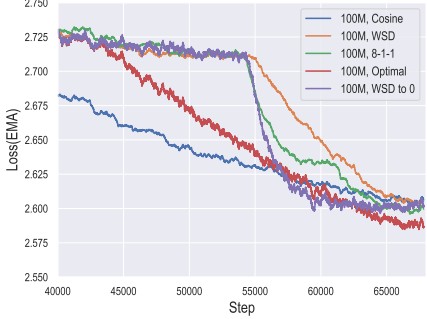

Figure 9: **Comparison between optimal LRS and WSD with a near-zero final learning rate:** We train a 100M LLaMA (dense) model with 40B tokens training data and various LRSs with the same $\eta_{\max} = 10^{-3}$, including WSD LRS with $\eta_{\min} = \frac{1}{10}\eta_{\max}$, WSD LRS with $\eta_{\min} = 10^{-7}$, cosine LRS with $\eta_{\min} = \frac{1}{10}\eta_{\max}$, 8-1-1 LRS with $\eta_{\min} = \frac{1}{10}\eta_{\max}$, and optimal LRS. The experimental results show that decaying to (near) zero does not result in significant loss reduction.

## C.3 Key Proof Steps and Core Insights

In this section, we outline the main ideas behind the proof of the FSL (10), highlighting the key techniques. Complete proofs of the above theorems are deferred to Appendix D.

We will need the following characterization of the gradient noise structure.

**Lemma C.1** (Noise structure). *For any* $\mathbf{v} \in \mathbb{R}^M$, *it holds that*

$$(2\rho_- \mathcal{E}(\mathbf{v}) + \sigma^2) \nabla^2 \mathcal{R}(\mathbf{v}) \preceq \mathbf{\Sigma}(\mathbf{v}) \preceq (2\rho_+ \mathcal{E}(\mathbf{v}) + \sigma^2) \nabla^2 \mathcal{R}(\mathbf{v}),$$

*where* $\nabla^2 \mathcal{R}(\mathbf{v}) = \mathbf{W}\mathbf{H}\mathbf{W}^\top$, *and the constants* $\rho_-$ *and* $\rho_+$ *are the same as in Assumption 2.1.*

The proof is provided in Appendix D.1. Let $\boldsymbol{\xi}(\mathbf{v})$ denote the gradient noise at $\mathbf{v}$. Since $\mathcal{R}(\mathbf{v}) = \mathcal{E}(\mathbf{v}) + \frac{1}{2}\sigma^2$, it follows that for any direction $\boldsymbol{n} \in \mathbb{S}^{M-1}$,

$$\mathbb{E}[|\boldsymbol{\xi}(\mathbf{v})^\top \boldsymbol{n}|^2] = \boldsymbol{n}^\top \mathbf{\Sigma}(\mathbf{v})\boldsymbol{n} \approx \mathcal{R}(\mathbf{v})\,\boldsymbol{n}^\top \nabla^2 \mathcal{R}(\mathbf{v})\boldsymbol{n},$$

where $\boldsymbol{n}^\top \nabla^2 \mathcal{R}(\mathbf{v})\boldsymbol{n}$ represents the local curvature of the risk landscape along $\boldsymbol{n}$. Hence, the noise energy in each direction is proportional to the product of the population risk and the curvature along that direction. This **anisotropic structure** of the gradient noise—scaling with the risk and shaped by curvature—has also been reported in prior work [67, 62].

For clarity, in this section, we focus on the case of top-$M$ features, for which the population risk takes the form

$$2\mathcal{E}(\mathbf{v}) = \sum_{j=1}^{M} \lambda_j (v_j - \theta_j^*)^2 + \sum_{j=M+1}^{N} \lambda_j |\theta_j^*|^2. \tag{18}$$

For the intrinsic-time SDE (7), each coordinate of $\boldsymbol{\nu}_t$ evolves as

$$\mathrm{d}\nu_j(t) = -\lambda_j(\nu_j - \theta_j^*)\,\mathrm{d}t + \sqrt{\gamma(t)}\sum_{k=1}^{M}\sqrt{\mathbf{\Sigma}(\boldsymbol{\nu}_t)}_{jk}\,\mathrm{d}B_t^{(k)},$$

where $B_t^{(k)}$ is the $k$-th coordinate of the $M$-dimensional Browian motion $\mathbf{B}_t$.

Observe that the summation of white noises is equivalent to a single stochastic term $\sqrt{q_j(t)}\,\mathrm{d}B_j(t)$ with variance $q_j(t) := \sum_{k=1}^{M}|\sqrt{\mathbf{\Sigma}(\boldsymbol{\nu}_t)}_{jk}|^2 = \|\sqrt{\mathbf{\Sigma}(\boldsymbol{\nu}_t)}\mathbf{e}_j\|_2^2 = \mathbf{e}_j^\top \mathbf{\Sigma}(\boldsymbol{\nu}_t)\mathbf{e}_j$, $\mathbf{e}_j$ is the $j$-th canonical basis vector for $j \in [M]$. Therefore each coordinate of $\boldsymbol{\nu}_t$ satisfies

$$\mathrm{d}\nu_j(t) = -\lambda_j(\nu_j - \theta_j^*)\,\mathrm{d}t + \sqrt{\gamma(t)q_j(t)}\,\mathrm{d}B_j(t),$$

where $q_j(t) = \mathbf{e}_j^\top \mathbf{\Sigma}(\boldsymbol{\nu}_t)\mathbf{e}_j$ is the variance of gradient noise along $\mathbf{e}_j$. By applying Itô's formula to $(\nu_j - \theta_j^*)^2$ and noting $\boldsymbol{\nu}(0) = \mathbf{0}$, we obtain

$$\mathbb{E}[(\nu_j(t) - \theta_j^*)^2] = (0 - \theta_j^*)^2 e^{-2\lambda_j t} + \int_0^t e^{-2\lambda_j(t-z)}\gamma(z)\mathbb{E}[q_j(z)]\,\mathrm{d}z.$$

Let $\mathcal{E}_t = \mathcal{E}(\boldsymbol{\nu}_t)$ and plugging the above equation into (18) gives

$$2\mathbb{E}[\mathcal{E}_t] = \sum_{j=1}^{M}\lambda_j|\theta_j^*|^2 e^{-2\lambda_j t} + \sum_{j=1}^{M}\lambda_j\int_0^t e^{-2\lambda_j(t-z)}\gamma(z)\mathbb{E}[q_j(z)]\,\mathrm{d}z + \sum_{j=M+1}^{N}\lambda_j|\theta_j^*|^2. \tag{19}$$

By Lemma C.1 and noting $\nabla^2 \mathcal{R}(\mathbf{v}) = \mathrm{diag}(\lambda_1, \ldots, \lambda_M)$, we have

$$q_j(t) = \mathbf{e}_j^\top \mathbf{\Sigma}(\boldsymbol{\nu}_t)\mathbf{e}_j \approx \lambda_j \mathcal{R}(\boldsymbol{\nu}_t) = \lambda_j(\mathcal{E}(\boldsymbol{\nu}_t) + \sigma^2/2).$$

Let $\delta_M = \sum_{j=M+1}^{N}\lambda_j|\theta_j^*|^2$, $e_M(t) = \sum_{j=1}^{M}\lambda_j|\theta_j^*|^2 e^{-2\lambda_j t}$, and $\mathcal{K}_M(t) = \sum_{j=1}^{M}\lambda_j^2 e^{-2\lambda_j t}$. Plugging them back into (19) gives the following **Volterra equation**:

$$\mathbb{E}[\mathcal{E}_t] \approx \delta_M + e_M(t) + \int_0^t \mathcal{K}_M(t-z)\gamma(z)(\mathbb{E}[\mathcal{E}_z] + \sigma^2)\,\mathrm{d}z. \tag{20}$$

The above equation characterizes the expected loss dynamics of SGD under a general spectrum and has been derived in prior works such as [47, 48, 28, 49, 50]. Our key observation is that, under the power-law assumptions on $\{\theta_j^*\}_j$ and $\{\lambda_j\}_j$ (Assumptions 2.3 and 2.4), the solution to (20) admits a *sharp asymptotic characterization*, providing explicit upper and lower bounds that precisely capture its scaling behavior.

Let $f(t) := \mathbb{E}[\mathcal{E}_t]$, $g(t) := \delta_M + e_M(t) + \sigma^2 \int_0^t \mathcal{K}_M(t-z)\gamma(z)\,\mathrm{d}z$, and define the linear operator

$$\mathcal{T}f(t) = \int_0^t \mathcal{K}_M(t-z)\gamma(z)f(z)\,\mathrm{d}z.$$

Then, the Volterra equation (20) can be expressed in the compact form $f = g + \mathcal{T}f$. Formally, its solution can be expanded as an infinite series:

$$f = (\mathcal{I} - \mathcal{T})^{-1}g = g + \mathcal{T}g + \mathcal{T}^2 g + \mathcal{T}^3 g + \cdots. \tag{21}$$

The key observation is that, under Assumptions 2.3 and 2.4, **the higher-order terms $\mathcal{T}^k g$ for $k \geqslant 2$ can be well controlled by the first-order term $\mathcal{T}g$.** This is due to the scale invariance of the forgetting kernel:

---

**Lemma C.2** (Scale invariance). *It holds for any $t \lesssim M^\beta$ that $\frac{\mathcal{K}_M(t/2)}{\mathcal{K}_M(t)} \simeq 1$.*

---

*Proof.* The result follows from the fact that $\mathcal{K}_M$ exhibits a power-law decay for $t \lesssim M^\beta$ (see (25)). Consequently, $\mathcal{K}_M(t/2) \asymp (t/2)^{-(2-1/\beta)} \asymp t^{-(2-1/\beta)} \asymp \mathcal{K}_M(t)$, which establishes the claim. $\quad\square$

**Corollary C.3** (Subconvolution property). *For any $t \lesssim M^\beta$, it holds $\mathcal{K}_M * \mathcal{K}_M(t) \lesssim \mathcal{K}_M(t)$.*

*Proof.* Noting $\mathcal{K}_M$ is non-increasing and integrable and applying Lemma C.2, we obtain

$$(\mathcal{K}_M * \mathcal{K}_M)(t) = \int_0^{t/2} \mathcal{K}_M(t-z)\mathcal{K}_M(z)\,\mathrm{d}z + \int_{t/2}^t \mathcal{K}_M(t-z)\mathcal{K}_M(z)\,\mathrm{d}z$$

$$\leqslant 2\,\mathcal{K}_M(t/2)\int_0^{t/2} \mathcal{K}_M(z)\,\mathrm{d}z \;\lesssim\; \mathcal{K}_M(t/2) \;\lesssim\; \mathcal{K}_M(t),$$

which proves the claim. $\quad\square$

Let $\|\gamma\|_\infty = \sup_{t \geqslant 0} \gamma(t)$. By Corollary C.3, it holds for any $t \lesssim M^\beta$ that

$$\mathcal{T}^2 g(t) \leqslant \|\gamma\|_\infty \int_0^t \mathcal{K}_M * \mathcal{K}_M(t-z)\gamma(z)g(z)\,\mathrm{d}z$$

$$\lesssim \|\gamma\|_\infty \int_0^t \mathcal{K}_M(t-z)\gamma(z)g(z)\,\mathrm{d}z = \|\gamma\|_\infty \mathcal{T}g(t).$$

When $\|\gamma\|_\infty$ is sufficiently small, there exists a constant $0 < c < 1$ such that $\mathcal{T}^k g(t) \leqslant c^{k-1}\mathcal{T}g(t)$ holds for any $0 \leqslant t \lesssim M^\beta$. Hence,

$$f(t) \leqslant g(t) + \mathcal{T}g(t) + \sum_{k=2}^\infty c^{k-1}\mathcal{T}g(t) \lesssim g(t) + \mathcal{T}g(t) + \tfrac{c}{1-c}\mathcal{T}g(t).$$

Combining $f(t) \geqslant g(t) + \mathcal{T}g(t)$ with the above upper bound, we conclude $f(t) \asymp g(t) + \mathcal{T}g(t)$, i.e., we have

$$\mathbb{E}[\mathcal{R}(\boldsymbol{\nu}_t)] - \tfrac{1}{2}\sigma^2 \;\asymp\; \delta_M + e_M(t) + \int_0^t \mathcal{K}_M(t-z)\left[e_M(z) + \sigma^2\right]\gamma(z)\,\mathrm{d}z. \tag{22}$$

**The emergence of power-law scaling.** We now illustrate how power-law scaling arises in our setting from a multi-task learning viewpoint. Intuitively, we can view the learning of each eigenfunction as a sub-task. Due to the limited model size, student model can at most learn the top-$M$ eigenfunctions. Within this view, our FSL framework reveals three distinct manifestations of power-law behavior:

(i) **Approximation error.** Approximation error accounts for total risk of the $N - M$ unlearned sub-tasks, which follows a power-law decay due to

$$\delta_M = \sum_{j=M+1}^{N} \lambda_j |\theta_j^*|^2 \asymp M^{-s\beta}, \text{ if } N - M \gtrsim M. \tag{23}$$

(ii) **Signal learning.** For each sub-task, the sub-task risk converges exponentially w.r.t. the intrinsic time $t$. However, owing to the power-law structure of $\{\lambda_j\}, \{\theta_j^*\}$, the total multi-task risk exhibits a power-law decay for sufficiently large $M$ due to

$$e_M(t) = \sum_{j=1}^{M} \lambda_j |\theta_j^*|^2 e^{-2\lambda_j t} \asymp \int_{M^{-\beta}}^{1} u^{s-1} e^{-2ut} \, \mathrm{d}u \asymp \frac{1}{t^s}, \text{ if } 1 \lesssim t \lesssim M^\beta. \tag{24}$$

(iii) **Noise forgetting.** Analogously, for the forgetting kernel, we also have

$$\mathcal{K}_M(t) = \sum_{j=1}^{M} \lambda_j^2 e^{-2\lambda_j t} \asymp \int_{M^{-\beta}}^{1} z^{1-\frac{1}{\beta}} e^{-2zt} \, \mathrm{d}z \asymp t^{-(2-1/\beta)}, \text{ if } 1 \lesssim t \lesssim M^\beta. \tag{25}$$

---

*Remark* C.4 (Task Accumulation Effect). The above derivation reveals that, although a single task may not exhibit an exact power-law scaling with respect to intrinsic time, the *collective accumulation of many tasks* gives rise to such scaling behavior. As the number of tasks increases, the power-law regime progressively extends, and in the idealized limit $M \to \infty$, it spans the entire training process. In our setting, each sub-task naturally follows an exponential learning curve $\exp(-\lambda t)$, yet the underlying mechanism appears more universal. We anticipate that similar power-law behavior may arise even when individual task dynamics deviate from this form, which we leave for future investigation.

---

# D  Proofs for Section 4

## D.1  Volterra Integral Equation Governing the Loss Dynamics

In this section, we derive a Volterra-type integral equation that exactly characterizes the evolution of expected loss under the intrinsic-time SDE. This equation serves as the starting point for all subsequent theoretical analysis. Recall the intrinsic-time SDE:

$$\mathrm{d}\boldsymbol{\nu}_t = -\nabla \mathcal{R}(\boldsymbol{\nu}_t) \, \mathrm{d}t + \sqrt{\gamma_t \boldsymbol{\Sigma}(\boldsymbol{\nu}_t)} \, \mathrm{d}\mathbf{B}_t, \tag{26}$$

where we write $\gamma_t = \gamma(t)$ for simplicity.

By the definition of $\mathcal{R}(\mathbf{v})$, we have $\nabla \mathcal{R}(\mathbf{v}) = \mathbf{W}\mathbf{H}(\mathbf{W}^\top \mathbf{v} - \boldsymbol{\theta}^*)$. Let $\mathbf{u}_t = \mathbf{W}^\top \boldsymbol{\nu}_t - \boldsymbol{\theta}^*$. Then, we have

$$\mathcal{E}_t = \mathcal{E}(\mathbf{u}_t) = \frac{1}{2} \|\mathbf{u}_t\|_\mathbf{H}^2$$

To obtain the estimate of $\mathcal{E}_t$, we consider the intrinsic-time SDE for $\mathbf{u}_t$ given by:

**Lemma D.1.** *We have*

$$\mathrm{d}\mathbf{u}_t = -\mathbf{W}^\top \mathbf{W}\mathbf{H}\mathbf{u}_t \, \mathrm{d}t + \sqrt{\gamma_t \mathbf{W}^\top \boldsymbol{\Sigma}_t \mathbf{W}} \, \mathrm{d}\mathbf{B}_t, \tag{27}$$

*where $\boldsymbol{\Sigma}_t := \boldsymbol{\Sigma}(\boldsymbol{\nu}_t)$.*

*Proof.* By Eq. (26),

$$\mathrm{d}\mathbf{u}_t = \mathrm{d}(\mathbf{W}^\top \boldsymbol{\nu}_t - \mathbf{v}^*) = \mathbf{W}^\top \, \mathrm{d}\boldsymbol{\nu}_t = -\mathbf{W}^\top \mathbf{W}\mathbf{H}\mathbf{u}_t \, \mathrm{d}t + \mathbf{W}^\top \sqrt{\gamma_t \boldsymbol{\Sigma}_t} \, \mathrm{d}\tilde{\mathbf{B}}_t.$$

Here $\tilde{\mathbf{B}}_t$ is an $N$ dimensional standard Brownian motion, we are going to replace it with an $M$ dimensional standard Brownian motion $\mathbf{B}_t$.

It is easy to see that the diffusion term $\mathbf{W}^\top \sqrt{\gamma_t \Sigma_t}\, \mathrm{d}\tilde{\mathbf{B}}_t$ has the same distribution as $\sqrt{\gamma_t \mathbf{W}^\top \Sigma_t \mathbf{W}}\, \mathrm{d}\mathbf{B}_t$, hence the SDE can be written in $\mathbf{B}_t$ as

$$\mathrm{d}\mathbf{u}_t = -\mathbf{W}^\top \mathbf{W}\mathbf{H}\mathbf{u}_t\, \mathrm{d}t + \sqrt{\gamma_t \mathbf{W}^\top \Sigma_t \mathbf{W}}\, \mathrm{d}\mathbf{B}_t.$$

$\square$

A key insight for tractability is that the gradient noise exhibits the following anisotropic structure:

Our analytic analysis also relies on the noise structure characterized by the following lemma.

**Lemma D.2** (Noise Structure). *For any $\mathbf{v} \in \mathbb{R}^M$, it holds that*

$$\left(2\rho_- \mathcal{E}(\mathbf{v}) + \sigma^2\right) \mathbf{W}\mathbf{H}\mathbf{W}^\top \preceq \Sigma(\mathbf{v}) \preceq \left(2\rho_+ \mathcal{E}(\mathbf{v}) + \sigma^2\right) \mathbf{W}\mathbf{H}\mathbf{W}^\top.$$

Noting $\nabla^2 \mathcal{R}(\mathbf{v}) = \mathbf{W}\mathbf{H}\mathbf{W}^\top$ and $\mathcal{R}(\mathbf{v}) = \mathcal{E}(\mathbf{v}) + \frac{1}{2}\sigma^2$, this lemma means $\Sigma(\mathbf{v}) \asymp \mathcal{R}(\mathbf{v})\nabla^2 \mathcal{R}(\mathbf{v})$. That is, the gradient noise scales proportionally with the population risk and aligns with the local curvature. Notably, the noise has two distinct sources: (i) the fit-dependent term $\mathcal{E}(\mathbf{v})$, which arises purely from minibatching and persists even in the absence of label noise; (ii) the $\sigma^2$ term, which captures the contribution from label noise. This anisotropic structure of SGD noise – scaling with risk and shaped by curvature – has also been observed in prior work [67, 66].

*Proof.* Noting $\ell(\mathbf{z}; \mathbf{v}) = \frac{1}{2}(\mathbf{v}^\top \mathbf{W}\phi(\mathbf{x}) - y)^2$, we have

$$\nabla \ell(\mathbf{z}; \mathbf{v}) = \mathbf{W}\phi(\mathbf{x})\phi(\mathbf{x})^\top \left(\mathbf{W}^\top \mathbf{v} - \boldsymbol{\theta}^*\right) - \mathbf{W}\phi(\mathbf{x})\epsilon$$
$$\nabla \mathcal{R}(\mathbf{v}) = \mathbb{E}[\nabla \ell(\mathbf{z}; \mathbf{v})] = \mathbf{W}\mathbf{H}\left(\mathbf{W}^\top \mathbf{v} - \boldsymbol{\theta}^*\right).$$

Hence, the covariance matrix of the noise $\boldsymbol{\xi} := \nabla \ell(\mathbf{z}; \mathbf{v}) - \nabla \mathcal{R}(\mathbf{v})$ is given by

$$\Sigma(\mathbf{v}) = \mathbb{E}[\boldsymbol{\xi}\boldsymbol{\xi}^\top | \mathbf{v}]$$
$$= \mathbf{W}\left(\mathbb{E}\left[\phi(\mathbf{x})\phi(\mathbf{x})^\top \mathbf{u}\mathbf{u}^\top \phi(\mathbf{x})\phi(\mathbf{x})^\top\right] - \mathbf{H}\mathbf{u}\mathbf{u}^\top \mathbf{H}\right)\mathbf{W}^\top + \sigma^2 \mathbf{W}\mathbf{H}\mathbf{W}^\top.$$

Noting

$$\mathbb{E}\left[\phi(\mathbf{x})\phi(\mathbf{x})^\top \mathbf{u}\mathbf{u}^\top \phi(\mathbf{x})\phi(\mathbf{x})^\top\right] - \mathbf{H}\mathbf{u}\mathbf{u}^\top \mathbf{H} = \mathbb{E}\left[\phi(\mathbf{x})^\top \mathbf{u}\mathbf{u}^\top \phi(\mathbf{x})\phi(\mathbf{x})\phi(\mathbf{x})^\top\right] - \mathbf{H}\mathbf{u}\mathbf{u}^\top \mathbf{H},$$

then applying Assumption 2.1, we have

$$\Sigma(\mathbf{v}) \preceq \rho_+ \mathbf{W}\mathrm{tr}(\mathbf{H}\mathbf{u}\mathbf{u}^\top)\mathbf{H}\mathbf{W}^\top + \sigma^2 \mathbf{W}\mathbf{H}\mathbf{W}^\top = \left(2\rho_+ \mathcal{E}(\mathbf{u}) + \sigma^2\right)\mathbf{W}\mathbf{H}\mathbf{W}^\top,$$

where the last step follows from $\mathrm{tr}(\mathbf{H}\mathbf{u}\mathbf{u}^\top) = \|\mathbf{u}\|_{\mathbf{H}}^2 = 2\mathcal{E}(\mathbf{v})$. The lower bound follows the same proof. $\square$

**The excess-risk dynamics** is then given by the following Volterra integral equation:

**Proposition D.3.** *For the intrinsic-time SDE, we have*

$$2\mathbb{E}[\mathcal{E}_t] = \mathbf{u}_0^\top \mathbf{A}_t^\top \mathbf{H}\mathbf{A}_t \mathbf{u}_0 + \int_0^t \mathrm{tr}(\mathbf{S}\mathbf{A}_{t-\tau}^\top \mathbf{H}\mathbf{A}_{t-\tau}\mathbf{S}) \cdot \gamma_\tau (c_\tau \mathbb{E}[\mathcal{E}_\tau] + \sigma^2)\, \mathrm{d}\tau, \qquad (28)$$

*where $\mathbf{A}_t := e^{-\mathbf{W}^\top \mathbf{W}\mathbf{H}t}$, $\mathbf{S} := \sqrt{\mathbf{W}^\top \mathbf{W}\mathbf{H}\mathbf{W}^\top \mathbf{W}}$, and $c_\tau \in [2\rho_-, 2\rho_+]$ for any $\tau \geqslant 0$.*

*Proof.* By Itô's formula,

$$\mathrm{d}(e^{\mathbf{W}^\top \mathbf{W}\mathbf{H}t}\mathbf{u}_t) = e^{\mathbf{W}^\top \mathbf{W}\mathbf{H}t}(\mathbf{W}^\top \mathbf{W}\mathbf{H}\mathbf{u}_t\, \mathrm{d}t + \mathrm{d}\mathbf{u}_t)$$
$$= e^{\mathbf{W}^\top \mathbf{W}\mathbf{H}t}\sqrt{\gamma_t \mathbf{W}^\top \Sigma_t \mathbf{W}}\, \mathrm{d}\mathbf{B}_t.$$

Integrating both sides, we get

$$e^{\mathbf{W}^\top \mathbf{W}\mathbf{H}t}\mathbf{u}_t - \mathbf{u}_0 = \int_0^t e^{\mathbf{W}^\top \mathbf{W}\mathbf{H}\tau}\sqrt{\gamma_\tau \mathbf{W}^\top \Sigma_\tau \mathbf{W}}\, \mathrm{d}\mathbf{B}_\tau.$$

Now write $\mathbf{A}_t = e^{-\mathbf{W}^\top \mathbf{W} \mathbf{H} t}$, we have

$$\mathbf{u}_t = \mathbf{A}_t \mathbf{u}_0 + \int_0^t \mathbf{A}_{t-\tau} \sqrt{\gamma_\tau \mathbf{W}^\top \mathbf{\Sigma}_\tau \mathbf{W}} \, d\mathbf{B}_\tau.$$

Note that the integral with respect to $\mathbf{B}_t$ always has zero expectation, therefore we have

$$2\mathbb{E}\mathcal{E}_t = \mathbb{E}(\mathbf{u}_t^\top \mathbf{H} \mathbf{u}_t)$$

$$= \mathbf{u}_0^\top \mathbf{A}_t^\top \mathbf{H} \mathbf{A}_t \mathbf{u}_0 + \mathbb{E} \int_0^t \gamma_\tau \mathrm{tr}\left(\sqrt{\mathbf{W}^\top \mathbf{\Sigma}_\tau \mathbf{W}} \mathbf{A}_{t-\tau}^\top \mathbf{H} \mathbf{A}_{t-\tau} \sqrt{\mathbf{W}^\top \mathbf{\Sigma}_\tau \mathbf{W}}\right) d\tau.$$

By Lemma F.2 and Lemma D.2, we have

$$\mathrm{tr}\left(\sqrt{\mathbf{W}^\top \mathbf{\Sigma}_\tau \mathbf{W}} \mathbf{A}_{t-\tau}^\top \mathbf{H} \mathbf{A}_{t-\tau} \sqrt{\mathbf{W}^\top \mathbf{\Sigma}_\tau \mathbf{W}}\right) \leqslant (2\rho_+ \mathcal{E}_\tau + \sigma^2)\mathrm{tr}(\mathbf{S}\mathbf{A}_{t-\tau}^\top \mathbf{H}\mathbf{A}_{t-\tau}\mathbf{S}),$$

$$\mathrm{tr}\left(\sqrt{\mathbf{W}^\top \mathbf{\Sigma}_\tau \mathbf{W}} \mathbf{A}_{t-\tau}^\top \mathbf{H} \mathbf{A}_{t-\tau} \sqrt{\mathbf{W}^\top \mathbf{\Sigma}_\tau \mathbf{W}}\right) \geqslant (2\rho_- \mathcal{E}_\tau + \sigma^2)\mathrm{tr}(\mathbf{S}\mathbf{A}_{t-\tau}^\top \mathbf{H}\mathbf{A}_{t-\tau}\mathbf{S}).$$

Hence there exists some constant $c_\tau \in [2\rho_-, 2\rho_+]$ such that

$$\mathbb{E}\mathrm{tr}\left(\sqrt{\mathbf{W}^\top \mathbf{\Sigma}_\tau \mathbf{W}} \mathbf{A}_{t-\tau}^\top \mathbf{H} \mathbf{A}_{t-\tau} \sqrt{\mathbf{W}^\top \mathbf{\Sigma}_\tau \mathbf{W}}\right) = (c_\tau \mathbb{E}[\mathcal{E}_\tau] + \sigma^2)\mathrm{tr}(\mathbf{S}\mathbf{A}_{t-\tau}^\top \mathbf{H}\mathbf{A}_{t-\tau}\mathbf{S}),$$

from which the lemma follows. $\square$

### D.2 Power-law Decay of the $e_M$ and $\mathcal{K}_M$ Functions

In this section, we rigorously establish the power-law decay behavior of the functions $e_M$ and $\mathcal{K}_M$.

**Lemma D.4.** *For $e_M(t)$, $\mathcal{K}_M(t)$ we have the following estimation:*

$$e_M(t) \lesssim t^{-s}, \quad \mathcal{K}_M(t) \lesssim t^{-(2-\frac{1}{\beta})}.$$

*Moreover,*

$$e_M(t) \asymp t^{-s}, \quad \mathcal{K}_M(t) \asymp t^{-(2-\frac{1}{\beta})}, \quad 1 \lesssim t \lesssim M^\beta.$$

*Proof.* Recall that

$$e_M(t) = \sum_{j=1}^M e^{-2\lambda_j t} \lambda_j |\theta_j^*|^2 \asymp \sum_{j=1}^M e^{-2\lambda_j t} j^{-1-s\beta}.$$

Since $\lambda_j \asymp j^{-\beta}$, there exists a constant $c$ such that

$$\lambda_j \leqslant cj^{-\beta} \implies 2\lambda_j t \leqslant 1 \text{ when } j \geqslant (2ct)^{\frac{1}{\beta}}.$$

We have

$$e_M(t) \asymp \sum_{j=1}^{[(2ct)^{\frac{1}{\beta}}]} e^{-2\lambda_j t} j^{-1-s\beta} + \sum_{j=[(2ct)^{\frac{1}{\beta}}]+1}^M e^{-2\lambda_j t} j^{-1-s\beta}$$

$$\lesssim (2ct)^{\frac{1}{\beta}} \max_j e^{-2\lambda_j t} j^{-1-s\beta} + \sum_{j=[(2ct)^{\frac{1}{\beta}}]+1}^M j^{-1-s\beta}.$$

Now we bound the two terms separately:

$$\max_j e^{-2\lambda_j t} j^{-1-s\beta} \asymp \max_j e^{-2\lambda_j t} (\lambda_j t)^{s+\frac{1}{\beta}} \cdot t^{-s-\frac{1}{\beta}} \leqslant t^{-s-\frac{1}{\beta}} \sup_{x>0} e^{-2x} x^{s+\frac{1}{\beta}} \lesssim t^{-s-\frac{1}{\beta}}.$$

where the last inequality is because $\sup_{x>0} e^{-2x} x^{s+\frac{1}{\beta}}$ is a constant only depending on $s$ and $\beta$.

We have

$$\sum_{j=[(2ct)^{\frac{1}{\beta}}]+1}^M j^{-1-s\beta} \lesssim \int_{[(2ct)^{\frac{1}{\beta}}]}^\infty x^{-1-s\beta} \, dx \asymp [(2ct)^{\frac{1}{\beta}}]^{-s\beta} \asymp t^{-s}.$$

Therefore we have for all $t > 0$,

$$e_M(t) \lesssim (2ct)^{\frac{1}{\beta}} t^{-s-\frac{1}{\beta}} + t^{-s} \asymp t^{-s}.$$

On the other hand, when $M > 2([(2ct)^{\frac{1}{\beta}} + 1])$, i.e., $t \lesssim M^\beta$,

$$e_M(t) \asymp \sum_{j=1}^{[(2ct)^{\frac{1}{\beta}}]} e^{-2\lambda_j t} j^{-1-s\beta} + \sum_{j=[(2ct)^{\frac{1}{\beta}}]+1}^{M} e^{-2\lambda_j t} j^{-1-s\beta}$$

$$\gtrsim e^{-1} \sum_{j=[(2ct)^{\frac{1}{\beta}}]+1}^{M} j^{-1-s\beta}$$

$$\gtrsim \int_{[(2ct)^{\frac{1}{\beta}}]+1}^{2[(2ct)^{\frac{1}{\beta}}]+1} x^{-1-s\beta} \, \mathrm{d}x$$

$$= \frac{1}{s\beta} (([(2ct)^{\frac{1}{\beta}} + 1])^{-s\beta} - (2[(2ct)^{\frac{1}{\beta}}] + 1)^{-s\beta})$$

$$\asymp t^{-s},$$

where the last line requires $t \gtrsim 1$. Therefore we have shown that $e_M(t) \asymp t^{-s}$ when $1 \lesssim t \lesssim M^\beta$. The proof for $\mathcal{K}_M(t)$ follows the same way. □

**Lemma D.5.** *We can bound the gap between $e_M(t)$ and $e_\infty(t)$ as follows:*

$$e_\infty(t) - e_M(t) \lesssim M^{-s\beta}.$$

*As a consequence, we have*

$$e_M(t) + M^{-s\beta} \asymp e_\infty(t) + M^{-s\beta}.$$

*Proof.* We have

$$e_\infty(t) - e_M(t) = \sum_{j=M+1}^{\infty} e^{-2\lambda_j t} \lambda_j |\theta_j^*|^2$$

$$\lesssim \sum_{j=M+1}^{\infty} j^{-1-s\beta}$$

$$\leqslant \int_M^{\infty} x^{-1-s\beta} \, \mathrm{d}x \asymp M^{-s\beta}.$$

□

## D.3 The Case of Top-$M$ Features

First, we prove the general label-noise case of FSL (Theorem 4.3). Applying this result, we then derive the constant label-noise case (Theorem 4.4), the noiseless case (Theorem 4.5) and the hard regime case (Theorem 4.2)

### D.3.1 Proof of Theorem 4.3

In the top-$M$ feature case, the matrix $\mathbf{W}$ satisfies $\mathbf{w}_j = \mathbf{e}_j$ for each $j \in [M]$, therefore we can simplify the equation for $\mathbb{E}[\mathcal{E}_t]$ as follows.

**Theorem D.6** (Volterra equation of the top-$M$ case)**.** *In the top-$M$ case, we have*

$$\mathbb{E}[\mathcal{E}_t] \asymp M^{-s\beta} + e_M(t) + \int_0^t \mathcal{K}_M(t-\tau)\gamma_\tau(\mathbb{E}[\mathcal{E}_\tau] + \sigma^2) \, \mathrm{d}\tau, \tag{29}$$

*where the function $e_M$ and $\mathcal{K}_M$ are defined as*

$$e_M(t) := \sum_{j=1}^{M} \lambda_j (\theta_j^*)^2 e^{-2\lambda_j t}, \quad \mathcal{K}_M(t) := \sum_{j=1}^{M} \lambda_j^2 e^{-2\lambda_j t}. \tag{30}$$

*Proof.* By (28) in Proposition D.3, note that $\mathbf{W}^\top \mathbf{W} \mathbf{H} = \mathbf{H}_{0:M} \in \mathbb{R}^{N \times N}$ is the top-$M$ part of the matrix $\mathbf{H}$, i.e. $\mathbf{H}_{0:M} = \mathrm{diag}\{\lambda_1, \ldots, \lambda_M, 0, \ldots, 0\}$, we get

$$\mathbb{E}[\mathcal{E}_t] \asymp \mathbf{u}_0^\top \mathbf{H} e^{-2\mathbf{H}_{0:M}t} \mathbf{u}_0 + \int_0^t \mathrm{tr}(\mathbf{H}_{0:M}^2 e^{-2\mathbf{H}_{0:M}}) \gamma_\tau (\mathbb{E}[\mathcal{E}_t] + \sigma^2) \, \mathrm{d}\tau,$$

which can be further written in terms of the eigenvalues $\{\lambda_j\}$ as

$$\mathbb{E}[\mathcal{E}_t] \asymp \sum_{j=1}^M \lambda_j (u_0^{(j)})^2 e^{-2\lambda_j t} + \sum_{j=M+1}^\infty \lambda_j (u_0^{(j)})^2 + \int_0^t \sum_{j=1}^M \lambda_j^2 e^{-2\lambda_j t} \gamma_\tau (\mathbb{E}[\mathcal{E}_t] + \sigma^2) \, \mathrm{d}\tau.$$

Note that $u_0^{(j)}$, the $j$-th component of $\mathbf{u}_0$, is equal to $\theta_j^*$ because of the zero initialization of $\boldsymbol{\nu}_0$.

Therefore by the definition of $e_M$ and $\mathcal{K}_M$ we arrive at the Volterra-type integral equation of $\mathbb{E}[\mathcal{E}_t]$. $\qquad\square$

**Lemma D.7.** *For the forgetting kernel $\mathcal{K}_M$ and $t \leqslant M^\beta$, there exists a constant $C$ independent of $t$, such that*

$$\mathcal{K}_M * \mathcal{K}_M(t) \leqslant C\mathcal{K}_M(t), \quad \forall t \leqslant M^\beta.$$

*where $*$ denotes convolution:*

$$\mathcal{K}_M * \mathcal{K}_M(t) := \int_0^t \mathcal{K}_M(\tau) \mathcal{K}_M(t - \tau) \, \mathrm{d}\tau.$$

*Proof.* Observe that $\mathcal{K}_M(t)$ is a monotonically decreasing function. By the symmetry of the convolution, we can write

$$\mathcal{K}_M * \mathcal{K}_M(t) = \int_0^t \mathcal{K}_M(\tau) \mathcal{K}_M(t - \tau) \, \mathrm{d}\tau$$
$$= 2 \int_0^{t/2} \mathcal{K}_M(\tau) \mathcal{K}_M(t - \tau) \, \mathrm{d}\tau.$$

Since $\mathcal{K}_M$ is decreasing, for $0 \leqslant \tau \leqslant t/2$ we have $\mathcal{K}_M(t - \tau) \leqslant \mathcal{K}_M(t/2)$. This observation yields the upper bound

$$\mathcal{K}_M * \mathcal{K}_M(t) \leqslant 2 \mathcal{K}_M\left(\tfrac{t}{2}\right) \int_0^\infty \mathcal{K}_M(\tau) \, \mathrm{d}\tau \leqslant C \mathcal{K}_M\left(\tfrac{t}{2}\right),$$

where the constant $C$ is finite and given by

$$C = 2 \int_0^\infty \mathcal{K}_M(\tau) \, \mathrm{d}\tau \leqslant 2 \int_0^\infty \sum_{j=1}^\infty \lambda_j^2 e^{-2\lambda_j \tau} \, \mathrm{d}\tau = \sum_{j=1}^\infty \lambda_j = \mathrm{tr}(\mathbf{H}) < \infty.$$

It remains to show that when $t \leqslant M^\beta$,

$$\mathcal{K}_M\left(\frac{t}{2}\right) \leqslant C' \mathcal{K}_M(t)$$

for some constant $C' > 0$. Recall that by Lemma D.4 we have

$$\mathcal{K}_M(t) \asymp t^{-(2-\frac{1}{\beta})}, \quad \mathcal{K}_M\left(\frac{t}{2}\right) \asymp \left(\frac{t}{2}\right)^{-(2-\frac{1}{\beta})}.$$

Therefore when $1 \lesssim t \lesssim M^\beta$ the conclusion follows. When $t \lesssim 1$, the function $\mathcal{K}_M(t)$ is decreasing and $\mathcal{K}_M(t) \leqslant \sum_{j=1}^M \lambda_j^2 < \infty$, we can always find a constant $C'$ satisfying the condition.

Combining the above estimates, we obtain

$$(\mathcal{K}_M * \mathcal{K}_M)(t) \leqslant C \mathcal{K}_M\left(\tfrac{t}{2}\right) \leqslant CC_1 \mathcal{K}_M(t) =: C' \mathcal{K}_M(t).$$

The proof is complete. $\qquad\square$

We now prove Theorem 4.3.

*Proof.* The lower bound is trivial by $\mathbb{E}[\mathcal{E}_t] \geqslant e_M(t)$ and the Volterra equation (29):

$$\mathbb{E}[\mathcal{E}_t] \asymp M^{-s\beta} + e_M(t) + \int_0^t \mathcal{K}_M(t-\tau)\gamma_\tau(\mathbb{E}[\mathcal{E}_\tau] + \sigma^2)\,\mathrm{d}\tau$$

$$\gtrsim M^{-s\beta} + e_M(t) + \int_0^t \mathcal{K}_M(t-\tau)\gamma_\tau(e_M(\tau) + \sigma^2)\,\mathrm{d}\tau.$$

For the upper bounds, we first prove a slightly stronger bound with $\mathcal{K}_\infty$, that is,

**Proposition D.8.** *For all $t > 0$, $s > 0$ and $\beta > 1$, we have*

$$\mathbb{E}[\mathcal{E}_t] \lesssim M^{-s\beta} + e_M(t) + \int_0^t \mathcal{K}_\infty(t-\tau)(e_M(\tau) + \sigma^2)\gamma_\tau\,\mathrm{d}\tau. \qquad (31)$$

*Proof of D.8.* By Equation (29), we have

$$\mathbb{E}[\mathcal{E}_t] \asymp M^{-s\beta} + e_M(t) + \int_0^t \mathcal{K}_M(t-\tau)\gamma_\tau(\mathbb{E}[\mathcal{E}_\tau] + \sigma^2)\,\mathrm{d}\tau$$

$$\leqslant M^{-s\beta} + e_M(t) + \int_0^t \mathcal{K}_\infty(t-\tau)\gamma_\tau(\mathbb{E}[\mathcal{E}_\tau] + \sigma^2)\,\mathrm{d}\tau.$$

Define $f(t) := \int_0^t \mathcal{K}_\infty(t-\tau)\gamma_\tau(\mathbb{E}[\mathcal{E}_\tau] + \sigma^2)\,\mathrm{d}\tau$ and $\gamma_{\max} = \sup_t \gamma(t)$, substituting the above inequality into the definition of $f(t)$, we get

$$f(t) \lesssim \int_0^t \mathcal{K}_\infty(t-\tau)(M^{-s\beta} + e_M(\tau) + \sigma^2)\gamma_\tau\,\mathrm{d}\tau$$

$$+ \int_0^t \int_0^\tau \mathcal{K}_\infty(t-\tau)\mathcal{K}_\infty(\tau-r)\gamma_\tau\gamma_r(\mathbb{E}[\mathcal{E}_r] + \sigma^2)\,\mathrm{d}r\,\mathrm{d}\tau$$

$$\lesssim \gamma_{\max}M^{-s\beta} + \int_0^t \mathcal{K}_\infty(t-\tau)(e_M(\tau) + \sigma^2)\gamma_\tau\,\mathrm{d}\tau$$

$$+ \gamma_{\max}\int_0^t \int_r^t \mathcal{K}_\infty(t-\tau)\mathcal{K}_\infty(\tau-r)\,\mathrm{d}\tau\gamma_r(\mathbb{E}[\mathcal{E}_r] + \sigma^2)\,\mathrm{d}r$$

$$= \gamma_{\max}M^{-s\beta} + \int_0^t \mathcal{K}_\infty(t-\tau)(e_M(\tau) + \sigma^2)\gamma_\tau\,\mathrm{d}\tau$$

$$+ \gamma_{\max}\int_0^t (\mathcal{K}_\infty * \mathcal{K}_\infty)(t-r)\gamma_r(\mathbb{E}[\mathcal{E}_r] + \sigma^2)\,\mathrm{d}r$$

$$\overset{\text{Lemma D.7}}{\lesssim} \gamma_{\max}M^{-s\beta} + \int_0^t \mathcal{K}_\infty(t-\tau)(e_M(\tau) + \sigma^2)\gamma_\tau\,\mathrm{d}\tau + \gamma_{\max}f(t).$$

Therefore when $\gamma_{\max}$ is sufficiently small ($\gamma_{\max} \leqslant \frac{c}{\mathrm{tr}(\mathbf{H})}$ for some absolute constant $c$), the constant factor of $f(t)$ on the right-hand side will be less than $\frac{1}{2}$, hence we may subtract $\gamma_{\max}f(t)$ from both sides and get

$$\int_0^t \mathcal{K}_\infty(t-\tau)\gamma_\tau\mathbb{E}[\mathcal{E}_\tau]\,\mathrm{d}\tau \lesssim \gamma_{\max}M^{-s\beta} + \int_0^t \mathcal{K}_\infty(t-\tau)(e_M(\tau) + \sigma^2)\gamma_\tau\,\mathrm{d}\tau.$$

Therefore substituting this back to the Volterra equation yields

$$\mathbb{E}[\mathcal{E}_t] \lesssim M^{-s\beta} + e_M(t) + \int_0^t \mathcal{K}_\infty(t-\tau)(e_M(\tau) + \sigma^2)\gamma_\tau\,\mathrm{d}\tau.$$

∎

The preceding proposition relies exclusively on two properties of $\mathcal{K}_\infty$: the identity established in Lemma D.7 and the finiteness of its integral, $\int_0^\infty \mathcal{K}_\infty(t)\,\mathrm{d}t < \infty$. Since the kernel $\mathcal{K}_M$ also possesses these properties for $t \leqslant M^\beta$, the derivation of the upper bound

$$\mathbb{E}[\mathcal{E}_t] \lesssim M^{-s\beta} + e_M(t) + \int_0^t \mathcal{K}_M(t-\tau)(e_M(\tau) + \sigma^2)\gamma_\tau\,\mathrm{d}\tau, \quad t \leqslant M^\beta$$

follows analogously by substituting $\mathcal{K}_\infty$ with $\mathcal{K}_M$. $\qquad\square$

### D.3.2  Proof of Theorem 4.4

*Proof.* It is clear that, by our assumption, $\sup_t \gamma_t \lesssim 1$, and that

$$e_M(t) \asymp \sum_{j=1}^M j^{-1-s\beta} e^{-2\lambda_j t} \leqslant \sum_{j=1}^M j^{-1-s\beta} \asymp 1.$$

By Proposition D.8, we have

$$\mathbb{E}[\mathcal{E}_t] \lesssim M^{-s\beta} + e_M(t) + \int_0^t \mathcal{K}_\infty(t-\tau)(e_M(t) + \sigma^2)\gamma_\tau\,\mathrm{d}\tau \tag{32}$$

$$\lesssim 1 + \int_0^t \mathcal{K}_\infty(t-\tau)(\sigma^2 + 1)\,\mathrm{d}\tau \lesssim \sigma^2, \tag{33}$$

where the last inequality is by $\sigma^2 \gtrsim 1$ and that

$$\int_0^t \mathcal{K}_\infty(\tau)\,\mathrm{d}\tau = \int_0^t \sum_{j=1}^\infty \lambda_j^2 e^{-2\lambda_j \tau}\,\mathrm{d}\tau = \frac{1}{2}\sum_{j=1}^\infty \lambda_j \asymp 1.$$

Therefore by the Volterra equation (29), note that $\mathbb{E}[\mathcal{E}_\tau] + \sigma^2 \asymp \sigma^2 \asymp e_M(\tau) + \sigma^2$, we have

$$\mathbb{E}[\mathcal{E}_t] \asymp M^{-s\beta} + e_M(t) + \int_0^t \mathcal{K}_M(t-\tau)(e_M(\tau) + \sigma^2)\gamma_\tau\,\mathrm{d}\tau.$$

$\qquad\square$

### D.3.3  Proof of Theorem 4.5

*Proof.* When $\sigma^2 = 0$, by FSL in general case (Proposition D.8), we have

$$\mathbb{E}[\mathcal{E}_t] \lesssim M^{-s\beta} + e_M(t) + \int_0^t \mathcal{K}_\infty(t-\tau)e_M(\tau)\gamma_\tau\,\mathrm{d}\tau.$$

In order to get the upper bound with $\mathcal{K}_M$, we first prove a lemma.

**Lemma D.9.** *We will bound the gap introduced by $\mathcal{K}_\infty$ and $\mathcal{K}_M$:*

$$\int_0^t (\mathcal{K}_\infty(t-\tau) - \mathcal{K}_M(t-\tau))e_M(\tau)\gamma_\tau\,\mathrm{d}\tau \lesssim M^{\max\{-s\beta, -2\beta+1\}}. \tag{34}$$

*Proof of Lemma.* First note that

$$\mathcal{K}_\infty(t) - \mathcal{K}_M(t) = \sum_{j=M+1}^\infty \lambda_j^2 e^{-2\lambda_j t} \lesssim \sum_{j=M+1}^\infty j^{-2\beta} \asymp M^{-2\beta+1}.$$

Therefore we can bound the integral as

$$\int_0^t (\mathcal{K}_\infty(t-\tau) - \mathcal{K}_M(t-\tau))e_M(\tau)\gamma_\tau\,\mathrm{d}\tau$$

$$\lesssim M^{-2\beta+1}\int_0^t e_M(\tau)\gamma_\tau\,\mathrm{d}\tau$$

$$\leqslant \gamma_{\max} M^{-2\beta+1} \int_0^\infty e_M(\tau)\, \mathrm{d}\tau$$

$$= \gamma_{\max} M^{-2\beta+1} \int_0^\infty \sum_{j=1}^M \lambda_j(\theta_j^*)^2 e^{-2\lambda_j\tau}\, \mathrm{d}\tau$$

$$\eqsim \gamma_{\max} M^{-2\beta+1} \sum_{j=1}^M j^{-s\beta-1+\beta}$$

$$\lesssim \gamma_{\max} M^{\max\{-s\beta-\beta+1,-2\beta+1\}} \lesssim \gamma_{\max} M^{\max\{-s\beta,-2\beta+1\}}$$

∎

By $s \leqslant 2 - \frac{1}{\beta}$ and $\gamma_{\max}$ is sufficiently small, we can combine the upper bound with (34), and directly conclude that

$$\mathbb{E}[\mathcal{E}_t] \lesssim M^{-s\beta} + e_M(t) + \int_0^t \mathcal{K}_M(t-\tau)e_M(\tau)\gamma_\tau\, \mathrm{d}\tau.$$

Now with the lower bound in FSL Theorem 4.3, the result follows. □

### D.3.4 Proof of Theorem 4.2

*Proof.* In the hard regime $s \leqslant 1 - \frac{1}{\beta}$, by Propostion D.8, we have

$$\mathbb{E}[\mathcal{E}_t] \lesssim M^{-s\beta} + e_M(t) + \int_0^t \mathcal{K}_\infty(t-\tau)(e_M(\tau) + \sigma^2)\gamma_\tau\, \mathrm{d}\tau.$$

**Lemma D.10.** *It holds for $s \leqslant 1 - \frac{1}{\beta}$ that*

$$\int_0^t (\mathcal{K}_\infty(t-\tau) - \mathcal{K}_M(t-\tau))\gamma_\tau\, \mathrm{d}\tau \lesssim M^{-\beta+1}. \tag{35}$$

*Proof of Lemma.* First from the definition of $\mathcal{K}$ we obtain

$$\mathcal{K}_\infty(t) - \mathcal{K}_M(t) = \sum_{j=M+1}^\infty \lambda_j^2 e^{-2\lambda_j t}.$$

Therefore, we have

$$\int_0^t (\mathcal{K}_\infty(t-\tau) - \mathcal{K}_M(t-\tau))\gamma_\tau\, \mathrm{d}\tau$$

$$= \gamma_{\max} \int_0^\infty \sum_{j=1}^M \lambda_j^2 e^{-2\lambda_j\tau}\, \mathrm{d}\tau$$

$$\eqsim \gamma_{\max} \sum_{j=1}^M j^{-\beta} \lesssim \gamma_{\max} M^{-\beta+1}.$$

∎

By $s \leqslant 1 - \frac{1}{\beta}$ and that $\gamma_{\max}$ is sufficiently small, we can combine the upper bound with (35), (34), and directly conclude that

$$\mathbb{E}[\mathcal{E}_t] \lesssim M^{-s\beta} + e_M(t) + \int_0^t \mathcal{K}_\infty(t-\tau)(e_M(\tau) + \sigma^2)\gamma_\tau\, \mathrm{d}\tau$$

$$\lesssim M^{-s\beta} + e_M(t) + \int_0^t \mathcal{K}_M(t-\tau)(e_M(\tau) + \sigma^2)\gamma_\tau\, \mathrm{d}\tau$$

$$+ \int_0^t (\mathcal{K}_\infty(t-\tau) - \mathcal{K}_M(t-\tau))(e_M(\tau) + \sigma^2)\gamma_\tau\, \mathrm{d}\tau$$

$$\text{(by (34),(35))} \quad \lesssim M^{-s\beta} + e_M(t) + \int_0^t \mathcal{K}_M(t-\tau)(e_M(\tau) + \sigma^2)\gamma_\tau \, d\tau$$
$$+ M^{\max\{-s\beta, -2\beta+1\}} + \sigma^2 M^{-\beta+1}$$
$$\lesssim M^{-s\beta} + e_M(t) + \int_0^t \mathcal{K}_M(t-\tau)(e_M(\tau) + \sigma^2)\gamma_\tau \, d\tau.$$

Combining with the lower bound in FSL Theorem 4.3, the result follows. $\qquad\square$

### D.4 The Case of Random-$M$ Features

**Notations.** Throughout this section, for a power series $P(x) = \sum_{i=0}^\infty a_i x^i$, we write $P(\mathbf{A}) = \sum_{i=0}^\infty a_i \mathbf{A}^i$ to denote the result of substituting a square matrix $\mathbf{A} \in \mathbb{R}^{d \times d}$ into $P(\cdot)$, where the power $\mathbf{A}^i$ is obtained through matrix multiplications.

For a vector $\mathbf{v} \in \mathbb{R}^d$, we write the norm $\|\mathbf{v}\|$ for $\|\mathbf{v}\|_2$ if not otherwise specified. For a matrix $\mathbf{A} \in \mathbb{R}^{d \times d}$, we use $\|\mathbf{A}\|_2$ to denote the operator norm induced by the vector 2-norm, and $\mu_1(\mathbf{A}) \geqslant \mu_2(\mathbf{A}) \geqslant \ldots \geqslant \mu_d(\mathbf{A})$ to denote the eigenvalues of $\mathbf{A}$. For a positive semi-definite matrix $\mathbf{C} \in \mathbb{R}^{d \times d}$ and a vector $\mathbf{v} \in \mathbb{R}^d$, we define $\|\mathbf{v}\|_{\mathbf{C}}$ as

$$\|\mathbf{v}\|_{\mathbf{C}} = \mathbf{v}^\top \mathbf{C} \mathbf{v}.$$

First, for the random-$M$ feature setting, we can establish the following Volterra equation.

**Proposition D.11.** *Suppose $0 < s \leqslant 1$. Then, with probability at least $1 - e^{-\Omega(M)}$, a similar Volterra equation as derived in Theorem D.6 continues to hold for the random-feature case; that is,*

$$\mathbb{E}[\mathcal{E}_t] \eqsim M^{-s\beta} + \widehat{e}_M(t) + \int_0^t \widehat{\mathcal{K}}_M(t-\tau)\,\gamma_\tau\big(\mathbb{E}[\mathcal{E}_\tau] + \sigma^2\big)\,d\tau, \tag{36}$$

*where*

$$\widehat{e}_M(t) := \sum_{j=1}^M \hat{\lambda}_j |\theta_j^*|^2 e^{-2\hat{\lambda}_j t}, \quad \widehat{\mathcal{K}}_M := \sum_{j=1}^M \hat{\lambda}_j^2 e^{-2\hat{\lambda}_j t},$$

*and $\hat{\lambda}_j := \mu_j(\mathbf{W}\mathbf{H}\mathbf{W}^\top)$ is the $j$-th eigenvalue of the random matrix.*

Theorem 4.6 then follows by applying exactly the same argument as in the top-$M$ case. It therefore remains to prove Proposition D.11. The key idea is to show that the spectrum of the random matrix $\mathbf{W}\mathbf{H}\mathbf{W}^\top \in \mathbb{R}^{M \times M}$ closely matches that of the top-$M$ truncation, namely,

$$\hat{\lambda}_j := \mu_j(\mathbf{W}\mathbf{H}\mathbf{W}^\top) \eqsim \lambda_j, \quad \forall\, 1 \leqslant j \leqslant M.$$

#### D.4.1 Concentration Inequalities

Recall that we derived the following recursive equation in Eq. (28):

$$2\mathbb{E}\mathcal{E}_t = \mathbf{u}_0^\top \mathbf{A}_t^\top \mathbf{H} \mathbf{A}_t \mathbf{u}_0 + \int_0^t \text{tr}(\mathbf{S}\mathbf{A}_{t-\tau}^\top \mathbf{H} \mathbf{A}_{t-\tau} \mathbf{S}) \cdot \gamma_\tau(c_\tau \mathbb{E}[\mathcal{E}_\tau] + \sigma^2)\,d\tau,$$

where $\mathbf{A}_t = e^{-\mathbf{W}^\top \mathbf{W} \mathbf{H} t}$ and $\mathbf{S} = (\mathbf{W}^\top \mathbf{W} \mathbf{H} \mathbf{W}^\top \mathbf{W})^{\frac{1}{2}}$.

We first introduce the following notation: for integers $0 \leqslant a < b \leqslant N$ (we allow $b = \infty$, in this case we regard it as the same as $b = N$),

$$\mathbf{H}_{a:b} = \text{diag}\{\lambda_{a+1}, \ldots, \lambda_b\} \in \mathbb{R}^{(b-a)\times(b-a)}, \quad \mathbf{u}_{a:b} = ((\mathbf{u})_{a+1}, \ldots, (\mathbf{u})_b) \in \mathbb{R}^{b-a},$$

while

$$\mathbf{W}_{a:b} = [\mathbf{W}_{a+1}, \ldots, \mathbf{W}_b] \in \mathbb{R}^{M \times (b-a)}$$

is the $(a+1)$-th to $b$-th columns of $\mathbf{W}$.

To understand this equation with random projection matrix $\mathbf{W}$, we leverage the following concentration results developed in [35].

**Lemma D.12** (Lemma G.4 in [35]). *For* $\mathbf{H} = \mathrm{diag}\{\lambda_1, \lambda_2, \ldots, \lambda_N\}$ *such that* $\lambda_j \asymp j^{-\beta}$ *where* $\beta > 1$, *there exists* $\beta$-*dependent constants* $0 < c_1 < c_2$ *such that it holds with probability at least* $1 - e^{-\Omega(M)}$ *for all* $j \in [M]$ *that*

$$c_1 \lambda_j \leqslant \mu_j(\mathbf{W}\mathbf{H}\mathbf{W}^\top) \leqslant c_2 \lambda_j$$

**Lemma D.13** (Lemma G.5 in [35]). *There exists some* $\beta$-*dependent constant* $c$ *such that for all* $k \geqslant 1$, *the ratio between the* $\frac{M}{2}$-*th and* $M$-*th eigenvalue*

$$\frac{\mu_{\frac{M}{2}}(\mathbf{W}_{k:\infty}\mathbf{H}_{k:\infty}\mathbf{W}_{k:\infty}^\top)}{\mu_M(\mathbf{W}_{k:\infty}\mathbf{H}_{k:\infty}\mathbf{W}_{k:\infty}^\top)} \leqslant c$$

*with probability at least* $1 - e^{-\Omega(M)}$.

### D.4.2 Upper and Lower Bounds

**Lemma D.14.** *With probability at least* $1 - e^{-\Omega(M)}$, *for* $s > 0$ *we have*

$$\mathrm{tr}(\mathbf{S}\mathbf{A}_{t-\tau}^\top \mathbf{H}\mathbf{A}_{t-\tau}\mathbf{S}) = \sum_{j=1}^{M} e^{-2(t-\tau)\hat{\lambda}_j}\hat{\lambda}_j^2 =: \widehat{\mathcal{K}}_M(t-\tau).$$

*Proof.* We can compute that

$$\mathrm{tr}(\mathbf{S}\mathbf{A}_{t-\tau}^\top \mathbf{H}\mathbf{A}_{t-\tau}\mathbf{S}) = \mathrm{tr}(\mathbf{W}^\top \mathbf{W}\mathbf{H}\mathbf{W}^\top \mathbf{W}\mathbf{A}_{t-\tau}^\top \mathbf{H}\mathbf{A}_{t-\tau})$$

$$= \mathrm{tr}\left(\mathbf{W}^\top \mathbf{W}\mathbf{H}\mathbf{W}^\top \mathbf{W} \sum_{a,b=0}^{\infty} \frac{1}{a!b!}(-(t-\tau))^{a+b}(\mathbf{H}\mathbf{W}^\top \mathbf{W})^a \mathbf{H}(\mathbf{W}^\top \mathbf{W}\mathbf{H})^b\right)$$

$$= \mathrm{tr}\left(\sum_{a,b=0}^{\infty} \frac{1}{a!b!}(-t+\tau)^{a+b} \cdot \mathbf{W}^\top (\mathbf{W}\mathbf{H}\mathbf{W}^\top)^{a+b+1}\mathbf{W}\mathbf{H}\right)$$

$$= \mathrm{tr}\left(\sum_{a,b=0}^{\infty} \frac{1}{a!b!}(-t+\tau)^{a+b} \cdot (\mathbf{W}\mathbf{H}\mathbf{W}^\top)^{a+b+2}\right)$$

$$= \sum_{a,b=0}^{\infty} \frac{1}{a!b!}(-t+\tau)^{a+b} \sum_{j=1}^{M} \hat{\lambda}_j^{a+b+2}$$

$$= \sum_{j=1}^{M} e^{-2(t-\tau)\hat{\lambda}_j}\hat{\lambda}_j^2,$$

which completes the proof. $\qquad\square$

For the first term in Eq. (28), following [35], we have

**Lemma D.15.** *With probability at least* $1 - e^{-\Omega(M)}$, *for* $0 < s \leqslant 1$ *we have*

$$\mathbf{u}_0^\top \mathbf{A}_t^\top \mathbf{H}\mathbf{A}_t\mathbf{u}_0 \lesssim M^{-s\beta} + \hat{e}_M(t).$$

*Proof.* Note

$$\mathbf{A}_t^\top \mathbf{H}\mathbf{A}_t = \sum_{a,b=0}^{\infty} \frac{1}{a!b!}(-t)^{a+b}(\mathbf{H}\mathbf{W}^\top \mathbf{W})^a \mathbf{H}(\mathbf{W}^\top \mathbf{W}\mathbf{H})^b$$

$$= \sum_{a,b=0}^{\infty} \frac{1}{a!b!}(-t)^{a+b}\mathbf{H}\mathbf{W}^\top (\mathbf{W}\mathbf{H}\mathbf{W}^\top)^{a+b-1}\mathbf{W}\mathbf{H}$$

$$= \mathbf{H}\mathbf{W}^\top (\mathbf{W}\mathbf{H}\mathbf{W}^\top)^{-1}\mathbf{M}_t(\mathbf{W}\mathbf{H}\mathbf{W}^\top)^{-1}\mathbf{W}\mathbf{H}$$

where $\mathbf{M}_t = P_t(\mathbf{WHW}^\top)$ with $P_t$ being the power series

$$P_t(x) := \sum_{a+b \geqslant 0} \frac{1}{a!b!}(-t)^{a+b}x^{a+b+1}.$$

Note that when $x \in \mathbb{R}$, we have

$$P_t(x) = x \sum_{a,b \geqslant 0} \frac{(-tx)^a}{a!} \cdot \frac{(-tx)^b}{b!} = xe^{-tx} \cdot e^{-tx} = xe^{-2tx}.$$

Hence the eigenvalues of $\mathbf{M}_t$ is exactly $P_t(\hat{\lambda}_j)$. Since

$$\mathbf{u}_0^\top \mathbf{A}_t^\top \mathbf{H} \mathbf{A}_t \mathbf{u}_0 = \mathbf{u}_0^\top (\mathbf{HW}^\top (\mathbf{WHW}^\top)^{-1}\mathbf{M}_t(\mathbf{WHW}^\top)^{-1}\mathbf{WH})\mathbf{u}_0,$$

for any positive integer $k \leqslant \frac{M}{2}$, note that $\mathbf{WHu} = \mathbf{W}_{0:k}\mathbf{H}_{0:k}\mathbf{u}_{0:k} + \mathbf{W}_{k:\infty}\mathbf{H}_{k:\infty}\mathbf{u}_{k:\infty}$, we have

$$\mathbf{u}_0^\top \mathbf{A}_t^\top \mathbf{H} \mathbf{A}_t \mathbf{u}_0 \leqslant 2(T_1 + T_2),$$

where

$$T_1 = \mathbf{u}_{0:k}^\top (\mathbf{H}_{0:k}\mathbf{W}_{0:k}^\top (\mathbf{WHW}^\top)^{-1}\mathbf{M}_t(\mathbf{WHW}^\top)^{-1}\mathbf{W}_{0:k}\mathbf{H}_{0:k})\mathbf{u}_{0:k},$$
$$T_2 = \mathbf{u}_{k:\infty}^\top (\mathbf{H}_{k:\infty}\mathbf{W}_{k:\infty}^\top (\mathbf{WHW}^\top)^{-1}\mathbf{M}_t(\mathbf{WHW}^\top)^{-1}\mathbf{W}_{k:\infty}\mathbf{H}_{k:\infty})\mathbf{u}_{k:\infty}.$$

Then by Lemma D.18, we can derive an upper bound. Since $s \leqslant 1$,

$$T_1 + T_2 \lesssim \frac{1}{t}\|\mathbf{u}_{0:k}\|_2^2 + \|\mathbf{u}_{k:\infty}\|_{\mathbf{H}_{k:\infty}}^2$$

$$\approx \frac{1}{t}\sum_{j=1}^{k} j^{-1+\beta(1-s)} + \sum_{j=k+1}^{N} j^{-1-s\beta} \approx \frac{k^{\beta(1-s)}}{t} + k^{-s\beta}.$$

By setting $k = \min\{t^{1/\beta}, \frac{M}{3}\}$, we have

$$\mathbf{u}_0^\top \mathbf{A}_t^\top \mathbf{H} \mathbf{A}_t \mathbf{u}_0 \lesssim \max\{t^{-s}, M^{-s\beta}\} \approx t^{-s} + M^{-s\beta}.$$

Clearly $\mathbf{u}_0^\top \mathbf{A}_t^\top \mathbf{H} \mathbf{A}_t \mathbf{u}_0 \lesssim 1$, therefore by Lemma D.5,

$$\mathbf{u}_0^\top \mathbf{A}_t^\top \mathbf{H} \mathbf{A}_t \mathbf{u}_0 \lesssim \min\{t^{-s}, 1\} + M^{-s\beta} \approx \widehat{e}_\infty(t) + M^{-s\beta} \approx \widehat{e}_M(t) + M^{-s\beta}.$$

$\square$

**Lemma D.16.** *For $s > 0$, it holds with probability at least $1 - e^{-\Omega(M)}$ that*

$$\mathbf{u}_0^\top \mathbf{A}_t^\top \mathbf{H} \mathbf{A}_t \mathbf{u}_0 \gtrsim \max\{\widehat{e}_M(t), M^{-s\beta}\},$$

*where $\widehat{e}_M(t) := \sum_{j=1}^{M} \hat{\lambda}_j |\theta_j^*|^2 e^{-2t\hat{\lambda}_j}$.*

*Proof.* Following the proof of Lemma D.15, we have

$$\mathbf{u}_0^\top \mathbf{A}_t^\top \mathbf{H} \mathbf{A}_t \mathbf{u}_0 = \mathbf{u}_0 \mathbf{HW}^\top (\mathbf{WHW}^\top)^{-1}\mathbf{M}_t(\mathbf{WHW}^\top)^{-1}\mathbf{WH}\mathbf{u}_0$$

$$= \mathrm{tr}\left((\mathbf{WHW}^\top)^{-1}\mathbf{M}_t(\mathbf{WHW}^\top)^{-1} \cdot \mathbf{WH}\mathbf{u}_0\mathbf{u}_0^\top \mathbf{HW}^\top\right)$$

$$\geqslant \sum_{i=1}^{M} \mu_{M-i+1}\left((\mathbf{WHW}^\top)^{-1}\mathbf{M}_t(\mathbf{WHW}^\top)^{-1}\right) \cdot \mu_i\left(\mathbf{WH}\mathbf{u}_0\mathbf{u}_0^\top \mathbf{HW}^\top\right),$$

where the last inequality is by Von Neumann's trace inequality: For two symmetric matrices $\mathbf{A}$, $\mathbf{B} \in \mathbb{R}^{d \times d}$ with eigenvalues $a_1 \geqslant a_2 \geqslant \ldots \geqslant a_d$ and $b_1 \geqslant b_2 \geqslant \ldots \geqslant b_d$, we have

$$\mathrm{tr}(\mathbf{AB}) \geqslant \sum_{i=1}^{d} a_i b_{d+1-i}.$$

Note that $\mathbf{M}_t = P_t(\mathbf{WHW}^\top)$, we then get

$$\mathbf{u}_0^\top \mathbf{A}_t^\top \mathbf{H}\mathbf{A}_t\mathbf{u}_0 \geqslant \sum_{i=1}^{M} \mu_i \left((\mathbf{WHW}^\top)^2 \mathbf{M}_t^{-1}\right)^{-1} \mu_i \left(\mathbf{WHu}_0\mathbf{u}_0^\top \mathbf{HW}^\top\right)$$

$$= \sum_{i=1}^{M} e^{-2t\hat{\lambda}_i} \hat{\lambda}_i^{-1} \mu_i \left(\mathbf{WHu}_0\mathbf{u}_0^\top \mathbf{HW}^\top\right).$$

Note that $\mathbf{u}_0 = \boldsymbol{\theta}^*$, by Assumption 2.3 and 2.4,

$$\mathbf{u}_0^\top \mathbf{A}_t^\top \mathbf{H}\mathbf{A}_t\mathbf{u}_0 \geqslant \sum_{i=1}^{M} e^{-2t\hat{\lambda}_i} \hat{\lambda}_i^{-1} \mu_i \left(\mathbf{WHu}_0\mathbf{u}_0^\top \mathbf{HW}^\top\right)$$

$$\asymp \sum_{i=1}^{M} e^{-2t\hat{\lambda}_i} \hat{\lambda}_i^{-1} i^{-1-\beta-s\beta}$$

$$\asymp \sum_{i=1}^{M} e^{-2t\hat{\lambda}_i} \hat{\lambda}_i |\theta_j^*|^2 = \hat{e}_M(t).$$

Here in the second line we used Lemma D.12 for matrix $\mathbf{Hu}_0\mathbf{u}_0^\top \mathbf{H}$ as

$$\mu_j(\mathbf{Hu}_0\mathbf{u}_0^\top \mathbf{H}) = \lambda_j^2 |\theta_j^*|^2 \asymp j^{-1-\beta-s\beta} \implies \mu_j(\mathbf{WHu}_0\mathbf{u}_0^\top \mathbf{HW}^\top) \asymp j^{-1-\beta-s\beta},$$

and the third line follows from again from Lemma D.12 that $\hat{\lambda}_j \asymp j^{-\beta}$.

On the other hand, we prove the lower bound on $M^{-s\beta}$. First, we claim that

$$\mathbf{u}_0^\top \mathbf{A}_t^\top \mathbf{H}\mathbf{A}_t\mathbf{u}_0 \geqslant \|(\mathbf{I} - \mathbf{H}^{\frac{1}{2}}\mathbf{W}^\top(\mathbf{WHW}^\top)^{-1}\mathbf{WH}^{\frac{1}{2}})\mathbf{H}^{\frac{1}{2}}\mathbf{u}_0\|^2 =: T_3,$$

which will be proved in the Lemma D.17

Notice that

$$T_3 = \left\langle \mathbf{I}_N - \mathbf{H}^{1/2}\mathbf{W}^\top(\mathbf{WHW}^\top)^{-1}\mathbf{WH}^{1/2}, \mathbf{H}^{1/2}\mathbf{u}_0\mathbf{u}_0^\top \mathbf{H}^{1/2} \right\rangle,$$

where the inner product $\langle \mathbf{A}, \mathbf{B} \rangle = \mathrm{tr}(\mathbf{A}^\top \mathbf{B})$ denotes the trace inner product between matrices. Therefore note that $\mu_i(\mathbf{H}^{\frac{1}{2}}\mathbf{u}_0\mathbf{u}_0^\top \mathbf{H}^{\frac{1}{2}}) = i^{-1-s\beta}$ by source and capacity conditions,

$$T_3 \geqslant \sum_{i=1}^{N} \mu_i \left(\mathbf{I}_N - \mathbf{H}^{1/2}\mathbf{W}^\top(\mathbf{WHW}^\top)^{-1}\mathbf{WH}^{1/2}\right) \cdot \mu_{N+1-i}(\mathbf{H}^{\frac{1}{2}}\mathbf{u}_0\mathbf{u}_0^\top \mathbf{H}^{\frac{1}{2}})$$

$$\gtrsim \sum_{i=1}^{N} \mu_i(\mathbf{M}) \cdot (N+1-i)^{-1-s\beta}$$

where the second line follows again from Von Neumann's trace inequality. Since $\mathbf{M} = \mathbf{I}_N - \mathbf{H}^{1/2}\mathbf{W}^\top(\mathbf{WHW}^\top)^{-1}\mathbf{WH}^{1/2}$ is a projection matrix such that $\mathbf{M}^2 = \mathbf{M}$ and $\mathrm{rank}(\mathbf{I}_N - \mathbf{M}) = M$ with probability 1, $\mathbf{M}$ must have $M$ eigenvalues 0 and $N - M$ eigenvalues 1.

Hence we have

$$T_3 \gtrsim \sum_{i=M}^{N} i^{-1-s\beta} \gtrsim M^{-s\beta}.$$

$\square$

**Lemma D.17.**

$$\mathbf{u}_0^\top \mathbf{A}_t^\top \mathbf{H}\mathbf{A}_t\mathbf{u}_0 \geqslant \|(\mathbf{I} - \mathbf{H}^{\frac{1}{2}}\mathbf{W}^\top(\mathbf{WHW}^\top)^{-1}\mathbf{WH}^{\frac{1}{2}})\mathbf{H}^{\frac{1}{2}}\mathbf{u}_0\|^2$$

*Proof.* By the definition of positive semi-definite, we only need to prove that

$$\mathbf{A}_t^\top \mathbf{H}\mathbf{A}_t \succeq \mathbf{H}^{\frac{1}{2}}(\mathbf{I} - \mathbf{H}^{\frac{1}{2}}\mathbf{W}^\top(\mathbf{WHW}^\top)^{-1}\mathbf{WH}^{\frac{1}{2}})^2\mathbf{H}^{\frac{1}{2}}$$

Notice that

$$\mathbf{A}_t^\top \mathbf{H} \mathbf{A}_t = e^{-\mathbf{H}\mathbf{W}^\top \mathbf{W}t} \mathbf{H} e^{-\mathbf{W}^\top \mathbf{W}\mathbf{H}t}$$

$$= \mathbf{H}^{\frac{1}{2}} \left( \mathbf{I} + \sum_{a+b \geqslant 1} \frac{1}{a!b!}(-t)^{a+b} \mathbf{H}^{\frac{1}{2}} \mathbf{W}^\top (\mathbf{W}\mathbf{H}\mathbf{W}^\top)^{-1} \mathbf{W}\mathbf{H}^{\frac{1}{2}} \right) \mathbf{H}^{\frac{1}{2}}$$

Notice that $\mathbf{H}$ is a positive definite matrix, and now we only need to prove

$$\mathbf{I} + \sum_{a+b \geqslant 1} \frac{1}{a!b!}(-t)^{a+b} \mathbf{H}^{\frac{1}{2}} \mathbf{W}^\top (\mathbf{W}\mathbf{H}\mathbf{W}^\top)^{a+b-1} \mathbf{W}\mathbf{H}^{\frac{1}{2}} \succeq (\mathbf{I} - \mathbf{H}^{\frac{1}{2}} \mathbf{W}^\top (\mathbf{W}\mathbf{H}\mathbf{W}^\top)^{-1} \mathbf{W}\mathbf{H}^{\frac{1}{2}})^2.$$

Let $\mathbf{P} = \mathbf{W}\mathbf{H}^{\frac{1}{2}}$. After simplification, we only need to prove that

$$\mathbf{I} + \sum_{a+b \geqslant 1} \frac{1}{a!b!}(-t)^{a+b} \mathbf{P}^\top (\mathbf{P}\mathbf{P}^\top)^{a+b-1} \mathbf{P} \succeq \mathbf{I} - \mathbf{P}^\top (\mathbf{P}\mathbf{P}^\top)^{-1} \mathbf{P}.$$

Notice that, by the definition of matrix exponential, we have

$$\mathbf{I} + \sum_{a+b \geqslant 1} \frac{1}{a!b!}(-t)^{a+b} \mathbf{P}^\top (\mathbf{P}\mathbf{P}^\top)^{a+b-1} \mathbf{P}$$

$$= \mathbf{I} - \mathbf{P}^\top (\mathbf{P}\mathbf{P}^\top)^{-1} \mathbf{P} + \mathbf{P}^\top (\mathbf{P}\mathbf{P}^\top)^{-1} \left( \sum_{a+b \geqslant 0} \frac{2^{a+b}}{(a+b)!}(-t)^{a+b} (\mathbf{P}\mathbf{P}^\top)^{a+b} \right) \mathbf{P}$$

$$= \mathbf{I} - \mathbf{P}^\top (\mathbf{P}\mathbf{P}^\top)^{-1} \mathbf{P} + \mathbf{P}^\top (\mathbf{P}\mathbf{P}^\top)^{-1} e^{-2\mathbf{P}\mathbf{P}^\top t} \mathbf{P}.$$

Notice that the matrix $\mathbf{P}^\top \mathbf{P}$ and $e^{-2\mathbf{P}\mathbf{P}^\top t}$ are both positive semi-definite, we have $\mathbf{P}^\top (\mathbf{P}\mathbf{P}^\top)^{-1} e^{-2\mathbf{P}\mathbf{P}^\top t} \mathbf{P}$ is positive semi-definite. As a result,

$$\mathbf{I} + \sum_{a+b \geqslant 1} \frac{1}{a!b!}(-t)^{a+b} \mathbf{P}^\top (\mathbf{P}\mathbf{P}^\top)^{a+b-1} \mathbf{P} \succeq \mathbf{I} - \mathbf{P}^\top (\mathbf{P}\mathbf{P}^\top)^{-1} \mathbf{P}.$$

which completes the proof. $\qquad\square$

**Lemma D.18.** *With probability* $1 - e^{-\Omega(M)}$*, we have*

$$T_1 \leqslant c \frac{\|\mathbf{u}_{0:k}\|_2^2}{t} \left( \frac{\mu_{\frac{M}{2}}(\mathbf{W}_{0:k}\mathbf{H}_{0:k}\mathbf{W}_{0:k}^\top)}{\mu_M(\mathbf{W}_{0:k}\mathbf{H}_{0:k}\mathbf{W}_{0:k}^\top)} \right)^2, \quad T_2 \leqslant \|\mathbf{u}_{k:\infty}\|_{\mathbf{H}_{k:\infty}}^2.$$

*where $c$ is some constant.*

*Proof.* Recall that the eigenvalues of $\mathbf{M}_t = P_t(\mathbf{W}\mathbf{H}\mathbf{W}^\top)$ is $P_t(\hat{\lambda}_j)$, so we have

$$\|\mathbf{M}_t\|_2 = \max_{1 \leqslant j \leqslant M} P_t(\hat{\lambda}_j) = \max_{1 \leqslant j \leqslant M} \hat{\lambda}_j e^{-2t\hat{\lambda}_j} = \frac{1}{2t} \max_{1 \leqslant j \leqslant M} 2t\hat{\lambda}_j e^{-2t\hat{\lambda}_j} \leqslant \frac{1}{2t} \sup_{x > 0} xe^{-x} \leqslant \frac{1}{2et},$$

where the last step follows from $\sup_{x>0} xe^{-x} \leqslant e^{-1}$.

By the definition of $T_1$, we have

$$T_1 \leqslant \|\mathbf{H}_{0:k}\mathbf{W}_{0:k}^\top (\mathbf{W}\mathbf{H}\mathbf{W}^\top)^{-1}\mathbf{M}_t(\mathbf{W}\mathbf{H}\mathbf{W}^\top)^{-1}\mathbf{W}_{0:k}\mathbf{H}_{0:k}\| \|\mathbf{u}_{0:k}\|_2^2$$

$$\leqslant \|\mathbf{M}_t\|_2 \|(\mathbf{W}\mathbf{H}\mathbf{W}^\top)^{-1}\mathbf{W}_{0:k}\mathbf{H}_{0:k}\|_2^2 \|\mathbf{u}_{0:k}\|_2^2$$

$$\leqslant \frac{c}{t} \|(\mathbf{W}\mathbf{H}\mathbf{W}^\top)^{-1}\mathbf{W}_{0:k}\mathbf{H}_{0:k}\|_2^2 \|\mathbf{u}_{0:k}\|_2^2.$$

We only need to show

$$\|(\mathbf{W}\mathbf{H}\mathbf{W}^\top)^{-1}\mathbf{W}_{0:k}\mathbf{H}_{0:k}\|_2 \leqslant c \left( \frac{\mu_{\frac{M}{2}}(\mathbf{W}_{k:\infty}\mathbf{H}_{k:\infty}\mathbf{W}_{k:\infty}^\top)}{\mu_M(\mathbf{W}_{k:\infty}\mathbf{H}_{k:\infty}\mathbf{W}_{k:\infty}^\top)} \right).$$

We denote $\mathbf{A}_k = \mathbf{W}_{k:\infty}\mathbf{H}_{k:\infty}\mathbf{W}_{k:\infty}^\top$, and since $\mathbf{WHW}^\top = \mathbf{W}_{0:k}\mathbf{H}_{0:k}\mathbf{W}_{0:k}^\top + \mathbf{A}_k$, we have

$$
\begin{aligned}
(\mathbf{WHW}^\top)^{-1}\mathbf{W}_{0:k}\mathbf{H}_{0:k} &= (\mathbf{A}_k^{-1} - \mathbf{A}_k^{-1}\mathbf{W}_{0:k}[\mathbf{H}_{0:k}^{-1} + \mathbf{W}_{0:k}^\top\mathbf{A}_k^{-1}\mathbf{W}_{0:k}]^{-1}\mathbf{W}_{0:k}^\top\mathbf{A}_k^{-1}\mathbf{W}_{0:k})\mathbf{W}_{0:k}\mathbf{H}_{0:k} \\
&= \mathbf{A}_k^{-1}\mathbf{W}_{0:k}\mathbf{H}_{0:k} - \mathbf{A}_k^{-1}\mathbf{W}_{0:k}[\mathbf{H}_{0:k}^{-1} + \mathbf{W}_{0:k}^\top\mathbf{A}_k^{-1}\mathbf{W}_{0:k}]^{-1}\mathbf{W}_{0:k}^\top\mathbf{A}_k^{-1}\mathbf{W}_{0:k}\mathbf{H}_{0:k} \\
&= \mathbf{A}_k^{-1}\mathbf{W}_{0:k}[\mathbf{H}_{0:k}^{-1} + \mathbf{W}_{0:k}^\top\mathbf{A}_k^{-1}\mathbf{W}_{0:k}]^{-1}\mathbf{H}_{0:k}^{-1}\mathbf{H}_{0:k} \\
&= \mathbf{A}_k^{-1}\mathbf{W}_{0:k}[\mathbf{H}_{0:k}^{-1} + \mathbf{W}_{0:k}^\top\mathbf{A}_k^{-1}\mathbf{W}_{0:k}]^{-1}
\end{aligned}
$$

where the first line uses Sherman–Morrison-Woodbury's identity: For matrices $\mathbf{A}$, $\mathbf{C}$, $\mathbf{U}$, $\mathbf{V}$ with suitable shapes, we have

$$
(\mathbf{A} + \mathbf{UCV})^{-1} = \mathbf{A}^{-1} - \mathbf{A}^{-1}\mathbf{U}(\mathbf{C}^{-1} + \mathbf{VA}^{-1}\mathbf{U})^{-1}\mathbf{VA}^{-1}.
$$

Since

$$
\mathbf{H}_{0:k}^{-1} + \mathbf{W}_{0:k}^\top\mathbf{A}_k^{-1}\mathbf{W}_{0:k} \succeq \mathbf{W}_{0:k}^\top\mathbf{A}_k^{-1}\mathbf{W}_{0:k}.
$$

it follows that

$$
\|[\mathbf{H}_{0:k}^{-1} + \mathbf{W}_{0:k}^\top\mathbf{A}_k^{-1}\mathbf{W}_{0:k}]^{-1}\|_2 \leqslant \|[\mathbf{W}_{0:k}^\top\mathbf{A}_k^{-1}\mathbf{W}_{0:k}]^{-1}\|_2.
$$

Therefore, with probability at least $1 - e^{-\Omega(M)}$

$$
\begin{aligned}
\|\mathbf{A}_k^{-1}\mathbf{W}_{0:k}[\mathbf{H}_{0:k}^{-1} + \mathbf{W}_{0:k}^\top\mathbf{A}_k^{-1}\mathbf{W}_{0:k}]^{-1}\|_2 &\leqslant \|\mathbf{A}_k^{-1}\|_2 \cdot \|\mathbf{W}_{0:k}\|_2 \cdot \|[\mathbf{H}_{0:k}^{-1} + \mathbf{W}_{0:k}^\top\mathbf{A}_k^{-1}\mathbf{W}_{0:k}]^{-1}\|_2 \\
&\leqslant \|\mathbf{A}_k^{-1}\|_2 \cdot \|\mathbf{W}_{0:k}\|_2 \cdot \|[\mathbf{W}_{0:k}^\top\mathbf{A}_k^{-1}\mathbf{W}_{0:k}]^{-1}\|_2 \\
&\leqslant \frac{\|\mathbf{A}_k^{-1}\|_2 \cdot \|\mathbf{W}_{0:k}\|_2}{\mu_{\min}(\mathbf{W}_{0:k}^\top\mathbf{A}_k^{-1}\mathbf{W}_{0:k})}.
\end{aligned}
$$

Assume $k \leqslant \frac{M}{2}$ and with probability at least $1 - e^{-\Omega(M)}$ for some constant $c > 0$, $\|\mathbf{W}_{0:k}\|_2 \leqslant c$.

We may write $\mathbf{W}_{0:k}^\top\mathbf{A}_k^{-1}\mathbf{W}_{0:k} = \sum_{i=1}^{M} \frac{1}{\widehat{\lambda}_{M-i}}\boldsymbol{s}_i\boldsymbol{s}_i^\top$, where $\boldsymbol{s}_i \overset{\text{i.i.d.}}{\sim} \mathcal{N}(0, \mathbf{I}_k/N)$ and $(\widehat{\lambda}_i)_{i=1}^{M}$ are eigenvalues of $\mathbf{A}_k$ in non-increasing order. Therefore, for $k \leqslant M/3$,

$$
\sum_{i=1}^{M} \frac{1}{\widehat{\lambda}_{M-i}}\boldsymbol{s}_i\boldsymbol{s}_i^\top \succeq \sum_{i=1}^{M/2} \frac{1}{\widehat{\lambda}_{M-i}}\boldsymbol{s}_i\boldsymbol{s}_i^\top \succeq \frac{1}{\widehat{\lambda}_{M/2}} \sum_{i=1}^{M/2} \boldsymbol{s}_i\boldsymbol{s}_i^\top \succeq \frac{c\mathbf{I}_k}{\widehat{\lambda}_{M/2}}.
$$

with probability at least $1 - e^{-\Omega(M)}$, where in the last inequality we again use the concentration properties of Gaussian covariance matrices (see e.g., Theorem 6.1 in [61]).

$$
\begin{aligned}
\|\mathbf{A}_k^{-1}\mathbf{W}_{0:k}[\mathbf{H}_{0:k}^{-1} + \mathbf{W}_{0:k}^\top\mathbf{A}_k^{-1}\mathbf{W}_{0:k}]^{-1}\|_2 &\lesssim \frac{\|\mathbf{A}_k^{-1}\|_2}{\mu_{\min}(\mathbf{W}_{0:k}^\top\mathbf{A}_k^{-1}\mathbf{W}_{0:k})} \\
&\leqslant \frac{\mu_{M/2}(\mathbf{A}_k)}{\mu_M(\mathbf{A}_k)}.
\end{aligned}
$$

Now we focus on $T_2$, by definition of $T_2$ we have

$$
\begin{aligned}
T_2 &= \mathbf{u}_{k:\infty}{}^\top\mathbf{H}_{k:\infty}\mathbf{W}_{k:\infty}^\top(\mathbf{WHW}^\top)^{-1/2}\exp(-2t\mathbf{WHW}^\top)(\mathbf{WHW}^\top)^{-1/2}\mathbf{W}_{k:\infty}\mathbf{H}_{k:\infty}\mathbf{u}_{k:\infty} \\
&\leqslant \mathbf{u}_{k:\infty}{}^\top\mathbf{H}_{k:\infty}\mathbf{W}_{k:\infty}^\top(\mathbf{WHW}^\top)^{-1}\mathbf{W}_{k:\infty}\mathbf{H}_{k:\infty}\mathbf{u}_{k:\infty} \\
&\leqslant \|\mathbf{H}_{k:\infty}^{1/2}\mathbf{W}_{k:\infty}^\top(\mathbf{WHW}^\top)^{-1}\mathbf{W}_{k:\infty}\mathbf{H}_{k:\infty}^{1/2}\| \cdot \|\mathbf{u}_{k:\infty}\|_{\mathbf{H}_{k:\infty}}^2 \\
&\leqslant \|\mathbf{u}_{k:\infty}\|_{\mathbf{H}_{k:\infty}}^2,
\end{aligned}
$$

where the last line follows from

$$
\begin{aligned}
&\|\mathbf{H}_{k:\infty}^{1/2}\mathbf{W}_{k:\infty}^\top(\mathbf{WHW}^\top)^{-1}\mathbf{W}_{k:\infty}\mathbf{H}_{k:\infty}^{1/2}\|_2 \\
&\quad = \|\mathbf{H}_{k:\infty}^{1/2}\mathbf{W}_{k:\infty}^\top(\mathbf{W}_{0:k}\mathbf{H}_{0:k}\mathbf{W}_{0:k}^\top + \mathbf{W}_{k:\infty}\mathbf{H}_{k:\infty}\mathbf{W}_{k:\infty}^\top)^{-1}\mathbf{W}_{k:\infty}\mathbf{H}_{k:\infty}^{1/2}\|_2 \\
&\quad \leqslant \|\mathbf{H}_{k:\infty}^{1/2}\mathbf{W}_{k:\infty}^\top\mathbf{A}_k^{-1}\mathbf{W}_{k:\infty}\mathbf{H}_{k:\infty}^{1/2}\|_2 = 1.
\end{aligned}
$$

The last line is because a nonzero projection matrix has norm 1. $\qquad\square$

### D.4.3 Proof of Proposition D.11

Combining Lemma D.14, D.15 and D.16, we get with probability at least $1 - e^{-\Omega(M)}$,

$$\mathbb{E}[\mathcal{E}_t] \asymp \mathbf{u}_0^\top \mathbf{A}_t^\top \mathbf{H} \mathbf{A}_t \mathbf{u}_0 + \int_0^t \operatorname{tr}(\mathbf{S} \mathbf{A}_{t-\tau}^\top \mathbf{H} \mathbf{A}_{t-\tau} \mathbf{S}) \gamma_\tau (\mathbb{E}[\mathcal{E}_\tau] + \sigma^2) \, \mathrm{d}\tau, \tag{37}$$

$$\asymp M^{-s\beta} + \widehat{e}_M(t) + \int_0^t \widehat{\mathcal{K}}_M(t-\tau) \gamma_\tau (\mathbb{E}[\mathcal{E}_\tau] + \sigma^2) \, \mathrm{d}\tau, \tag{38}$$

Here we arrive at the Volterra equation for random feature case. Since we have $\hat{\lambda}_j \asymp \lambda_j$, the proof in the top-$M$ case also holds for $\widehat{e}_M$ and $\widehat{\mathcal{K}}_M$.

## E  Proofs for Section 5

When the condition $\sigma \gtrsim 1$ holds – indicating a constant label-noise level – the FSL simplifies to

$$\mathbb{E}[\mathcal{E}(\boldsymbol{\nu}_t)] \asymp \frac{1}{M^{s\beta}} + \frac{1}{t^s} + \frac{\sigma^2}{B} \mathcal{N}(\varphi), \quad \text{with} \quad \mathcal{N}(\varphi) = \int_0^t \mathcal{K}_M(t-r) \varphi(T^{-1}(r)) \, \mathrm{d}r, \tag{39}$$

where $e_M(t) + M^{-s\beta} \asymp e_\infty(t) + M^{-s\beta} \asymp t^{-s} + M^{-s\beta}$ as $e_\infty(t) - e_M(t) \lesssim M^{-s\beta}$, and the fit-dependent noise term $e_M(r)$ is absorbed by the label noise term due to $\sigma \gtrsim 1$. Extending to the full range $\sigma \geqslant 0$ is possible but makes the statements and derivations much more involved. We therefore focus on the above case to streamline the exposition.

### E.1  Proofs for Constant LRS

In this section, we prove Theorem 5.2 and present the data-optimal scaling strategy, as well as some results related to the compute-optimal allocation.

**Theorem E.1** (Restatement of Theorem 5.2). *Under Assumption 5.1, when the learning rate $\eta(k) \equiv \eta$, for the top-$M$ selection of the projection matrix $\mathbf{W}$ or for the random case with probability at least $1 - e^{-\Omega(M)}$, we have*

$$\mathbb{E}[\mathcal{R}_K] - \frac{\sigma^2}{2} \asymp \frac{1}{(\eta K)^s} + \frac{\eta}{B} \sigma^2 + M^{-s\beta}.$$

*Proof.* By our main Theorem 4.2, when the learning rate $\eta(k) \equiv \eta$, denote $\gamma := \frac{\eta}{B}$, by (39) we have

$$\mathbb{E}[\mathcal{E}_K] \asymp \gamma \sigma^2 + \frac{1}{t^s} + M^{-s\beta}.$$

Now we may write it as

$$\mathbb{E}[\mathcal{R}_K] - \frac{\sigma^2}{2} \asymp \gamma \sigma^2 + \frac{1}{t^s} + M^{-s\beta}.$$

Notice that $t = \eta K$ and $\gamma = \frac{\eta}{B}$, we have

$$\mathbb{E}[\mathcal{R}_K] - \frac{\sigma^2}{2} \asymp \frac{1}{(\eta K)^s} + \frac{\eta}{B} \sigma^2 + M^{-s\beta}.$$

$\square$

**Theorem E.2.** *Given a total data size of $D \gg 1$, the optimal strategy for minimizing the final population risk, in terms of the effective learning rate $\gamma$ and model size $M$ is:*

$$\gamma_{\mathrm{opt}} \asymp D^{-\frac{s}{s+1}}, \quad M_{\mathrm{opt}} \gtrsim D^{\frac{1}{(1+s)\beta}}, \quad \mathcal{E}_{\mathrm{opt}} \asymp D^{-\frac{s}{s+1}}. \tag{40}$$

*Proof.* Since we have

$$\mathbb{E}[\mathcal{E}_K] \asymp \gamma \sigma^2 + \frac{1}{(\gamma D)^s} + M^{-s\beta},$$

By weighted AM-GM inequality, we have that when $\mathcal{E}_K$ is minimized, it must hold that

$$\gamma\sigma^2 \eqsim \frac{1}{(\gamma D)^s}$$

which gives

$$\gamma_{\text{opt}} \eqsim D^{-\frac{s}{s+1}}.$$

Substituting this into the error expression yields

$$\mathcal{E}_{\text{opt}} \eqsim D^{-\frac{s}{s+1}} + M^{-s\beta}.$$

To balance the two terms and achieve the optimal rate, we require

$$M_{\text{opt}} \gtrsim D^{\frac{1}{(1+s)\beta}}.$$

Consequently, the optimal loss rate becomes

$$\mathcal{E}_{\text{opt}} \eqsim D^{-\frac{s}{s+1}}.$$

$\square$

Next we consider the compute optimal strategy for constant learning rates. We define the compute $C = MKB$ to be the product of the model size, training steps and batch size.

**Theorem E.3.** *Given a total compute budget of $C \gg 1$, the optimal strategy for minimizing the final population risk, in terms of the effective learning rate $\gamma$, model size $M$, and data size $D := BK$, is:*

$$\gamma_{\text{opt}} \eqsim C^{-\frac{s\beta}{1+\beta+s\beta}}, M_{\text{opt}} \eqsim C^{\frac{1}{1+\beta+s\beta}}, D_{\text{opt}} \eqsim C^{\frac{\beta+s\beta}{1+\beta+s\beta}},$$

*Proof.* Since we have

$$\mathbb{E}[\mathcal{E}_K] \eqsim \gamma\sigma^2 + \frac{1}{(\eta K)^s} + M^{-s\beta},$$

substituting $K = \frac{C}{MB}$, we get

$$\mathbb{E}[\mathcal{E}_K] \eqsim \gamma\sigma^2 + \frac{M^s}{(C\gamma)^s} + M^{-s\beta}.$$

By weighted AM-GM inequality, we have that when $\mathcal{E}_K$ is minimized, it must hold that

$$\gamma\sigma^2 \eqsim \frac{M^s}{(C\gamma)^s}, \quad \frac{M^s}{(C\gamma)^s} \eqsim M^{-s\beta},$$

which gives

$$\gamma_{\text{opt}} \eqsim C^{-\frac{s\beta}{1+\beta+s\beta}}, \quad M_{\text{opt}} \eqsim C^{\frac{1}{1+\beta+s\beta}}.$$

Now we can further compute $D = BK = CM^{-1} \eqsim C^{\frac{\beta+s\beta}{1+\beta+s\beta}}.$ $\square$

## E.2 Proof for The Exponential-Decay LRS

Recall that the LRS given by

$$\varphi(\tau) = ae^{-\lambda\tau}, \text{ with } \varphi(K) = b,$$

where $\lambda = \log(a/b)/K =: 1/\bar{K}$. Note that the intrinsic-time transform is given by

$$T(\tau) = \int_0^\tau \varphi(r)\,\mathrm{d}r = \frac{a}{\lambda}\left(1 - e^{-\lambda\tau}\right).$$

Thus, we have

- The total intrinsic time is:

$$T(K) = \frac{a}{\lambda}(1 - e^{-\lambda K}) = \frac{K}{\log(a/b)}(a - b) =: \bar{K}(a - b).$$

For simplicity, we shall write $T = T(K)$ in what follows.

- The LRS-adjusted function in intrinsic time is given by

$$\gamma_\varphi(t) = \varphi(T^{-1}(t)) = a - \lambda t.$$

**Lemma E.4.** *The noise term satisfies* $\mathcal{N}(\varphi) = bI_1 + (a - b)I_2$ *with*

$$I_1 = \int_{M^{-\beta}}^1 \frac{1 - e^{-2uT}}{2u^{1/\beta}}\, du, \qquad I_2 = \int_{M^{-\beta}}^1 \left( \frac{1 - e^{-2uT} - 2uTe^{-2uT}}{4Tu^{1+1/\beta}} \right) du.$$

*Proof.* We use the integral to approximate the forgetting kernel $\mathcal{K}_M$ as

$$\mathcal{K}_M(t) \approx \sum_{j=1}^M j^{-2\beta} e^{-2j^{-\beta}t} \approx \int_1^M x^{-2\beta} e^{-2x^{-\beta}t}\, dx \approx \int_{M^{-\beta}}^1 u^{1-1/\beta} e^{-2ut}\, du.$$

Noticing $b = a - \lambda T$ and $\lambda T = a - b$, we have

$$\int_0^T \mathcal{K}_M(T-t)\gamma_\varphi(t)\, dt = \int_0^T \left( \int_{M^{-\beta}}^1 u^{1-1/\beta} e^{-2u(T-t)}\, du \right)(a - \lambda t)\, dt$$

$$= \int_{M^{-\beta}}^1 u^{1-1/\beta} e^{-2uT} \left( \int_0^T e^{2ut}(a - \lambda t)\, dt \right) du$$

$$= \int_{M^{-\beta}}^1 u^{1-1/\beta} e^{-2uT} \left[ \frac{a}{2u}\left(e^{2uT} - 1\right) - \frac{\lambda}{2u}\left( Te^{2uT} - \frac{e^{2uT} - 1}{2u} \right) \right] du$$

$$= \int_{M^{-\beta}}^1 \left[ \frac{a}{2u^{1/\beta}} - \frac{ae^{-2uT}}{2u^{1/\beta}} - \frac{\lambda T}{2u^{1/\beta}} + \frac{\lambda(1 - e^{-2uT})}{4u^{1+1/\beta}} \right] du$$

$$= \int_{M^{-\beta}}^1 \left[ \frac{a - \lambda T}{2u^{1/\beta}} - \frac{(a - \lambda T + \lambda T)e^{-2uT}}{2u^{1/\beta}} + \frac{\lambda(1 - e^{-2uT})}{4u^{1+1/\beta}} \right] du$$

$$= (a - \lambda T)\int_{M^{-\beta}}^1 \frac{1 - e^{-2uT}}{2u^{1/\beta}}\, du + \lambda T \int_{M^{-\beta}}^1 \left( -\frac{e^{-2uT}}{2u^{1/\beta}} + \frac{1 - e^{-2uT}}{4Tu^{1+1/\beta}} \right) du.$$

Thus, we complete the proof. $\qquad\square$

We next bound $I_1$ and $I_2$ separately.

**Lemma E.5.** *If $T$ and $M$ is sufficiently large, then* $I_1 = \frac{\beta}{2\beta-1} + o_{T,M}(1)$.

*Proof.* Note that

$$\int_{M^{-\beta}}^1 \frac{1}{2u^{1/\beta}}\, du = \frac{\beta(1 - M^{-(\beta-1)})}{2(\beta - 1)} =: A.$$

and

$$\int_{M^{-\beta}}^1 \frac{e^{-2uT}}{2u^{1/\beta}}\, du = \frac{1}{2(2T)^{1-1/\beta}} \int_{T/M^\beta}^T \frac{e^{-r}}{r^{1/\beta}}\, dr \leqslant \frac{\Gamma(1 + \frac{1}{\beta})}{2^{2-1/\beta}} \frac{1}{T^{1-1/\beta}} =: B.$$

Then, we complete the proof by noting $I_1 = A - B$. $\qquad\square$

**Lemma E.6.** *If $T$ and $M$ is sufficiently large, then*

$$I_2 \approx \frac{\beta \min(M, T^{1/\beta})}{4T}.$$

*Proof.* Let $r = uT$. Then, by a change of variable, we obtain

$$I_2 = \frac{1}{4T^{1-1/\beta}} \int_{\frac{T}{M^\beta}}^T \frac{1 - e^{-2r} - 2re^{-2r}}{r^{1+1/\beta}}\, dr =: \frac{1}{4T^{1-1/\beta}} \int_{\frac{T}{M^\beta}}^T q_\beta(r)\, dr.$$

It is easy to verify that for any $\beta \geqslant 1$, $\inf_{r \geqslant 0} q_\beta(r) \geqslant 0$ and $q_\beta(r) \approx r^{-1-1/\beta}$ when $r$ is sufficiently large. We refer to Figure 10 for an illustration of $q_\beta(\cdot)$.

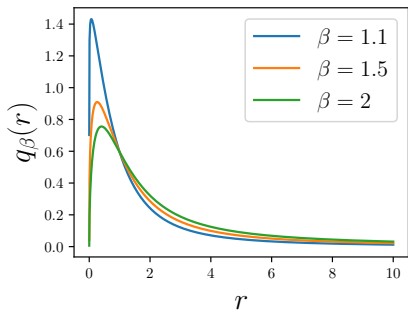

Figure 10: Illustration of the function $q_\beta(\cdot)$.

- When $T/M^\beta \leqslant 1$ and $T$ is sufficiently large such that $\int_{\frac{T}{M^\beta}}^T q_\beta(r)\,\mathrm{d}r \approx \beta$ and thus we have

$$I_2 = \frac{\beta + o_{M,T}(1)}{4T^{1-1/\beta}}.$$

- When $T/M^\beta > 1$, it holds for all $r \geqslant 1$ that $0.5 \leqslant 1 - e^{-2r} - 2re^{-2r} \leqslant 1$. Thus, there exists a $C_{T,M} \in [0.5, 1]$ such that

$$I_2 = C_{T,M} \frac{1}{4T^{1-1/\beta}} \int_{\frac{T}{M^\beta}}^T r^{-1-1/\beta}\,\mathrm{d}r$$

$$= \frac{C_{T,M}\beta}{4T^{1-1/\beta}} \left( \left(\frac{T}{M^\beta}\right)^{-1/\beta} - T^{-1/\beta} \right) = \frac{C_{T,M}\beta(M-1)}{4T}.$$

Combining the two cases, we complete the proof. $\qquad\square$

**Theorem E.7** (Theorem 5.3 in the main paper). *We consider the exponentially decaying learning rate schedule*

$$\varphi(\tau) = ae^{-\lambda\tau}, \text{ with } \varphi(K) = b,$$

*Under this learning rate schedule, for the top-$M$ projection matrix or the random projection with probability at least $1 - e^{-\Omega(M)}$, we have*

$$\mathcal{E}_K \approx M^{-s\beta} + T^{-s} + \frac{\sigma^2}{B}\left(b + (a-b)\frac{\min\{M, T^{1/\beta}\}}{T}\right),$$

*where $T = (a-b)K/\log(a/b)$ is the total intrinsic training time.*

*Proof.* By the functional scaling laws (39),

$$\mathcal{E}_K \approx M^{-s\beta} + T^{-s} + \frac{\sigma^2}{B}\mathcal{N}(\varphi).$$

The noise term $\mathcal{N}(\varphi)$ is estimated by Lemma E.4 and the bound on $I_1, I_2$ as

$$\mathcal{N}(\varphi) = bI_1 + (a-b)I_2 \approx b + (a-b)\frac{\min(M, T^{1/\beta})}{T},$$

which gives

$$\mathcal{E}_K \approx M^{-s\beta} + T^{-s} + \frac{\sigma^2}{B}\left(b + (a-b)\frac{\min(M, T^{1/\beta})}{T}\right),$$

so we complete the proof. $\qquad\square$

**Theorem E.8.** *Given a total data size $D \gg 1$, the optimal strategy for minimizing the final population risk when $b = \frac{a}{K}$ is given by $M_{\mathrm{opt}} = \infty$ and*

- *If $s > 1 - \frac{1}{\beta}$, then $\gamma_{\mathrm{opt}} \approx (D/\log D)^{-\frac{1+s\beta-\beta}{1+s\beta}}$ and $\mathcal{E}_{\mathrm{opt}} \approx (D/\log D)^{-\frac{s\beta}{s\beta+1}}$.*

- *If $s \leqslant 1 - \frac{1}{\beta}$, then $\gamma_{\mathrm{opt}} \asymp 1$ and $\mathcal{E}_{\mathrm{opt}} \asymp (D/\log D)^{-s}$.*

*Proof.* Denote $\tilde{D} := \frac{D}{\log K}$, then by Theorem 5.3,

$$\mathcal{E}_K \asymp M^{-s\beta} + (\gamma\tilde{D})^{-s} + \frac{\min(M, (\gamma\tilde{D})^{\frac{1}{\beta}})}{\tilde{D}}.$$

**Case 1.** When $M^\beta \leqslant \gamma\tilde{D}$,

$$\mathcal{E}_K \asymp M^{-s\beta} + (\gamma\tilde{D})^{-s} + \frac{M}{\tilde{D}}.$$

We see that in this case $\gamma$ should be as large as possible, since $a \lesssim 1$, we set $\gamma \asymp 1$ accordingly.

In this case $M^{-s\beta} + \frac{M}{\tilde{D}} \gtrsim \tilde{D}^{-\frac{s\beta}{1+s\beta}}$, with equality at $M \asymp \tilde{D}^{\frac{1}{1+s\beta}}$.

When $s > 1 - \frac{1}{\beta}$, the above equality condition can be acheived as $M^\beta = \tilde{D}^{\frac{\beta}{1+s\beta}} < \tilde{D}$. Hence we have that

$$M_{\mathrm{opt}} \asymp \tilde{D}^{\frac{1}{1+s\beta}}, \quad \gamma_{\mathrm{opt}} \asymp 1, \quad \mathcal{E}_{\mathrm{opt}} \asymp \tilde{D}^{-\frac{s\beta}{1+s\beta}}.$$

Note that $\gamma = \frac{a}{B} \asymp 1$ and $a \lesssim 1$, which forces $B \asymp 1$, hence $\tilde{D} \asymp \frac{D}{\log D}$.

When $s \leqslant 1 - \frac{1}{\beta}$, the quantity $M^{-s\beta} + \frac{M}{\tilde{D}}$ is decreasing with respect to $M$, hence the optimal $M$ in this case is $M = (\gamma\tilde{D})^{\frac{1}{\beta}}$, which transfers to case 2.

**Case 2.** When $M^\beta > \gamma\tilde{D}$,

$$\mathcal{E}_K \asymp M^{-s\beta} + (\gamma\tilde{D})^{-s} + \gamma^{\frac{1}{\beta}} \frac{1}{\tilde{D}^{1-\frac{1}{\beta}}}.$$

Clearly in this case $M_{\mathrm{opt}} = \infty$, and by AM-GM inequality,

$$(\gamma\tilde{D})^{-s} + \gamma^{\frac{1}{\beta}} \frac{1}{\tilde{D}^{1-\frac{1}{\beta}}} \gtrsim \tilde{D}^{-\frac{s\beta}{1+s\beta}},$$

with equality at $\gamma \asymp \tilde{D}^{\frac{\beta-1-s\beta}{1+s\beta}}$.

When $s > 1 - \frac{1}{\beta}$, the equality can be achieved, hence we have

$$M_{\mathrm{opt}} = \infty, \quad \gamma_{\mathrm{opt}} \asymp \tilde{D}^{-\frac{1+s\beta-\beta}{1+s\beta}}, \quad \mathcal{E}_{\mathrm{opt}} \asymp \tilde{D}^{-\frac{s\beta}{1+s\beta}}.$$

When $s \leqslant 1 - \frac{1}{\beta}$, since $\gamma \lesssim 1$, we must have

$$M_{\mathrm{opt}} = \infty, \quad \gamma_{\mathrm{opt}} \asymp 1, \quad \mathcal{E}_{\mathrm{opt}} \asymp \tilde{D}^{-s}.$$

Similarly, as $a \lesssim 1$, we have $B \lesssim \tilde{D}^{1-\frac{\beta}{1+s\beta}}$, which means $K \gtrsim \tilde{D}^{\frac{\beta}{1+s\beta}}$, hence $\log K \asymp \log D$, $\tilde{D} \asymp \frac{D}{\log D}$.

**Summary.** Combining the two cases together, we see that $M_{\mathrm{opt}} = \infty$ can always achieves the optimal rate, hence the conclusion follows. $\square$

**Theorem E.9.** *Given a large total compute budget $C \gg 1$, the optimal strategy for minimizing the final population risk – expressed in terms of the effective maximum learning rate $\gamma$, model size $M$, and data size $D$ – is given by:*

- *When $s > 1 - \frac{1}{\beta}$, the optimal scaling laws are:*

$$\gamma_{\mathrm{opt}} \asymp \left(\frac{C}{\log C}\right)^{-\frac{1+\beta(s-1)}{2+s\beta}}, \quad M_{\mathrm{opt}} \asymp \left(\frac{C}{\log C}\right)^{\frac{1}{2+s\beta}}, \quad D_{\mathrm{opt}} \asymp C^{\frac{1+s\beta}{2+s\beta}} (\log C)^{\frac{1}{2+s\beta}},$$

  *which leads to the following optimal final population risk:*

$$\mathcal{E}_{\mathrm{opt}}(C) \asymp \left(\frac{C}{\log C}\right)^{-\frac{s\beta}{2+s\beta}}.$$

- *When $s \leqslant 1 - \frac{1}{\beta}$, the optimal scaling laws are*

$$\gamma_{\mathrm{opt}} \asymp 1, \quad M_{\mathrm{opt}} \asymp \left(\frac{C}{\log C}\right)^{\frac{1}{1+\beta}}, \quad D_{\mathrm{opt}} \asymp C^{\frac{\beta}{1+\beta}} (\log C)^{\frac{1}{1+\beta}},$$

*which leads to the following optimal final population risk:*

$$\mathcal{E}_{\mathrm{opt}}(C) \asymp \left(\frac{C}{\log C}\right)^{-\frac{s\beta}{1+\beta}}.$$

*Proof.* Denote $\tilde{D} = D/\log K$. For similar reasons as in the derivation of data-optimal scaling, we may assume $\log K \asymp \log C$ to simplify the proof. At this point, the loss can be reformulated as follows.

$$\mathcal{E}_K \asymp M^{-s\beta} + \frac{1}{(\gamma \tilde{D})^s} + \sigma^2 \frac{\min\{M, (\gamma \tilde{D})^{1/\beta}\}}{\tilde{D}}.$$

**Case 1.** $M^\beta < \gamma \tilde{D}$ and we have

$$\mathcal{E}_K \asymp M^{-s\beta} + \frac{1}{(\gamma \tilde{D})^s} + \sigma^2 \frac{M}{\tilde{D}}$$

As $\gamma$ only appears in the second term, and $\frac{1}{(\gamma \tilde{D})^s}$ is monotone decreasing with $\gamma$, we have that when $\mathcal{E}_K$ is minimized, it must hold that

$$M = (\gamma \tilde{D})^{1/\beta}.$$

When $s > 1 - \frac{1}{\beta}$, we then consider a weighted AM-GM inequality, we have

$$M^{-s\beta} = \sigma^2 \frac{M}{\tilde{D}}.$$

Combining with $C = MD$ and $M = (\gamma \tilde{D})^{1/\beta}$, we have

$$\gamma_{\mathrm{opt}} \asymp \left(\frac{C}{\log C}\right)^{-\frac{1+\beta(s-1)}{2+s\beta}}, \quad M_{\mathrm{opt}} \asymp \left(\frac{C}{\log C}\right)^{\frac{1}{2+s\beta}}, \quad D_{\mathrm{opt}} \asymp C^{\frac{1+s\beta}{2+s\beta}} (\log C)^{\frac{1}{2+s\beta}},$$

and

$$\mathcal{E}_{\mathrm{opt}}(C) \asymp \left(\frac{C}{\log C}\right)^{-\frac{s\beta}{2+s\beta}}.$$

When $s \leqslant 1 - \frac{1}{\beta}$, since $a \lesssim 1$, we set $\gamma_{\mathrm{opt}} \asymp 1$ accordingly, and proceed as follows:

$$M_{\mathrm{opt}} \asymp \left(\frac{C}{\log C}\right)^{\frac{1}{1+\beta}}, \quad D_{\mathrm{opt}} \asymp C^{\frac{\beta}{1+\beta}} (\log C)^{\frac{1}{1+\beta}},$$

and

$$\mathcal{E}_{\mathrm{opt}}(C) \asymp \left(\frac{C}{\log C}\right)^{-\frac{s\beta}{1+\beta}}.$$

**Case 2.** $M^\beta \geqslant \gamma \tilde{D}$ and we have

$$\mathcal{E}_K \asymp M^{-s\beta} + \frac{1}{(\gamma \tilde{D})^s} + \sigma^2 \frac{(\gamma \tilde{D})^{1/\beta}}{\tilde{D}}$$

As $M$ only appears in the second term, and $M^{-s\beta}$ is monotonically decreasing in $M$, we have that when $\mathcal{E}_K$ is minimized, it must hold that

$$M = (\gamma \tilde{D})^{1/\beta}.$$

And then the rest is identical to the first case. $\qquad \square$

### E.3 Proof for the WSD-Like LRS

To prove Theorem 5.4, we first present the following lemma, which gives an upper bound for the SGD noise induced by the stable phase.

**Lemma E.10.** *For $T_2 > 0$, we have*

$$\int_0^\infty \mathcal{K}_M(T_2 + t)\,\mathrm{d}t \lesssim \frac{\min\{M, T_2^{\frac{1}{\beta}}\}}{T_2}.$$

*Proof.* Similar to the previous section, we use integral to approximate the forgetting kernel $\mathcal{K}_M$ and get

$$\int_0^\infty \mathcal{K}_M(T_2 + t)\,\mathrm{d}t = \int_0^\infty \int_{M^{-\beta}}^1 u^{1-\frac{1}{\beta}} e^{-2u(T_2+t)}\,\mathrm{d}u\,\mathrm{d}t$$

$$= \int_{M^{-\beta}}^1 u^{1-\frac{1}{\beta}} e^{-2uT_2} \frac{\mathrm{d}u}{2u}$$

$$\approx \frac{1}{T_2^{1-\frac{1}{\beta}}} \int_{T_2 M^{-\beta}}^{T_2} u^{-\frac{1}{\beta}} e^{-2u}\,\mathrm{d}u.$$

Since the integral $\int_0^\infty u^{-\frac{1}{\beta}} e^{-2u}\,\mathrm{d}u$ is convergent, we have

$$\int_0^\infty \mathcal{K}_M(T_2 + t)\,\mathrm{d}t \lesssim \frac{1}{T_2^{1-\frac{1}{\beta}}}.$$

When $T_2 > M^\beta$, similarly we have

$$\int_0^\infty \mathcal{K}_M(T_2 + t)\,\mathrm{d}t \approx M^{1-\beta} \int_1^{M^\beta} u^{-\frac{1}{\beta}} e^{-2u \frac{T_2}{M^\beta}}\,\mathrm{d}u.$$

Let $p = \frac{T_2}{M^\beta} \geqslant 1$, we have

$$\int_0^\infty \mathcal{K}_M(T_2 + t)\,\mathrm{d}t \approx \frac{M}{T_2} p \int_1^{M^\beta} u^{-\frac{1}{\beta}} e^{-2up}\,\mathrm{d}u$$

$$\lesssim \frac{M}{T_2} \int_1^{M^\beta} u^{-\frac{1}{\beta}} e^{-2u}\,\mathrm{d}u \approx \frac{M}{T_2}.$$

Where the last line is because $pe^{-2up}$ is decreasing in $p$ when $u, p \geqslant 1$. $\qquad\square$

**Theorem E.11** (Theorem 5.4 in the main paper). *Suppose the FSL (10) hold and $M, K$ are sufficiently large. Then, we have*

$$\mathcal{E}_K \approx M^{-s\beta} + T^{-s} + \sigma^2 \left( \frac{b}{B} + (a - b)\frac{\min\{M, T_2^{1/\beta}\}}{BT_2} \right),$$

*where $T = aK_1 + (a - b)K_2/\log(a/b)$ is the total intrinsic training time, and $T_2 = (a - b)K_2/\log(a/b)$ is the decay-phase intrinsic training time.*

*Proof.* By the results of the exponential decay LRS, let $\lambda = \log(a/b)/K_2$, we have

$$\int_0^{T(K)} \mathcal{K}_M(T(K) - t)\gamma_\varphi(t)\,\mathrm{d}t = \int_0^{T_1} \mathcal{K}_M(T(K) - t)a\,\mathrm{d}t + \int_0^{T_2} \mathcal{K}_M(T_2 - t)(a - \lambda t)\,\mathrm{d}t,$$

Hence by the estimation of the noise term of the exponential decay LRS (see the proof of Theorem 5.3), we have

$$\int_0^{T_2} \mathcal{K}_M(T_2 - t)(a - \lambda t)\,\mathrm{d}t \approx b + \frac{(a - b)\min\{M, T_2^{\frac{1}{\beta}}\}}{T_2}.$$

Thus, we know

$$\int_0^{T(K)} \mathcal{K}_M(T(K)-t)\gamma_\varphi(t)\,\mathrm{d}t \approx \int_0^{T_1} \mathcal{K}_M(T(K)-t)a\,\mathrm{d}t + b + \frac{(a-b)\min\{M,T_2^{\frac{1}{\beta}}\}}{T_2}$$

$$\approx a\int_0^{T_1} \mathcal{K}_M(T_2+t)\,\mathrm{d}t + b + \frac{(a-b)\min\{M,T_2^{\frac{1}{\beta}}\}}{T_2}$$

$$\approx b + \frac{(a-b)\min\{M,T_2^{\frac{1}{\beta}}\}}{T_2}. \qquad \text{(by using Lemma E.10)}$$

Hence the loss is given by

$$\mathcal{E}_K \approx \frac{1}{T^s} + M^{-s\beta} + \frac{\sigma^2}{B}\left(b + (a-b)\frac{\min\{M,T_2^{\frac{1}{\beta}}\}}{T_2}\right).$$

$\square$

**Theorem E.12.** *Assume $b = \frac{a}{K}$, then we have the following data-optimal strategy:*

- *If $s \geqslant 1 - 1/\beta$, we have $\gamma_{\mathrm{opt}} \approx D^{-\frac{1+s\beta-\beta}{1+s\beta}}(\log D)^{-\frac{\beta-1}{1+s\beta}}$, $(D_1)_{\mathrm{opt}}, (D_2)_{\mathrm{opt}} \approx D$ and $\mathcal{E}_{\mathrm{opt}} \approx D^{-\frac{s\beta}{s\beta+1}}(\log D)^{\frac{s\beta-s}{1+s\beta}}$.*

- *If $s < 1 - 1/\beta$, we have $\gamma_{\mathrm{opt}} \approx 1$, $(D_1)_{\mathrm{opt}} \approx D$, $(D_2)_{\mathrm{opt}} \gtrsim D^{\frac{s\beta}{\beta-1}}\log D$ and $\mathcal{E}_{\mathrm{opt}} \approx D^{-s}$.*

*Proof.* Since the total intrinsic time $T \lesssim \gamma D$, we can always take $D_1 \approx D$ to ensure $T \approx \gamma D$. Denote $\tilde{D}_2 := \frac{D_2}{\log K}$, then by Theorem E.11,

$$\mathcal{E}_K \approx M^{-s\beta} + (\gamma D)^{-s} + \frac{\min(M,(\gamma\tilde{D}_2)^{\frac{1}{\beta}})}{\tilde{D}_2}.$$

**Case 1.** When $M^\beta \leqslant \gamma\tilde{D}_2$,

$$\mathcal{E}_K \approx M^{-s\beta} + (\gamma D)^{-s} + \frac{M}{\tilde{D}_2}.$$

We see that in this case $\gamma$ should be as large as possible, since $a \lesssim 1$, we set $\gamma \approx 1$ accordingly.

In this case $M^{-s\beta} + \frac{M}{\tilde{D}_2} \gtrsim \tilde{D}_2^{-\frac{s\beta}{1+s\beta}}$, with equality at $M \approx \tilde{D}_2^{\frac{1}{1+s\beta}}$.

When $s > 1 - \frac{1}{\beta}$, the above equality condition can be achieved as $M^\beta = \tilde{D}_2^{\frac{\beta}{1+s\beta}} < \tilde{D}_2$. Hence we have that

$$M_{\mathrm{opt}} \approx \tilde{D}_2^{\frac{1}{1+s\beta}}, \quad \gamma_{\mathrm{opt}} \approx 1, \quad \mathcal{E}_{\mathrm{opt}} \approx \tilde{D}_2^{-\frac{s\beta}{1+s\beta}}.$$

Therefore $(D_2)_{\mathrm{opt}} \approx D$. Note that $\gamma = \frac{a}{B} \approx 1$ and $a \lesssim 1$, which forces $B \approx 1$, hence $\tilde{D}_2 \approx \frac{D}{\log D}$.

When $s \leqslant 1 - \frac{1}{\beta}$, the quantity $M^{-s\beta} + \frac{M}{\tilde{D}_2}$ is decreasing with respect to $M$, hence the optimal $M$ in this case is $M = (\gamma\tilde{D}_2)^{\frac{1}{\beta}}$, which transfers to case 2.

**Case 2.** When $M^\beta > \gamma\tilde{D}_2$,

$$\mathcal{E}_K \approx M^{-s\beta} + (\gamma D)^{-s} + \gamma^{\frac{1}{\beta}}\frac{1}{\tilde{D}_2^{1-\frac{1}{\beta}}}.$$

Clearly in this case $M_{\mathrm{opt}} = \infty$, and by AM-GM inequality,

$$(\gamma D)^{-s} + \gamma^{\frac{1}{\beta}}\frac{1}{\tilde{D}_2^{1-\frac{1}{\beta}}} \gtrsim D^{-\frac{s}{1+s\beta}}\tilde{D}_2^{-\frac{s\beta-s}{1+s\beta}},$$

with equality at $\gamma \asymp D^{-\frac{s\beta}{1+s\beta}} \tilde{D}_2^{\frac{\beta-1}{1+s\beta}}$.

When $s > 1 - \frac{1}{\beta}$, the equality can be achieved, hence we have that $(D_2)_{\mathrm{opt}} \asymp D$, so $\tilde{D}_2 \asymp \frac{D}{\log K}$,

$$M_{\mathrm{opt}} = \infty, \quad \gamma_{\mathrm{opt}} \asymp D^{-\frac{1+s\beta-\beta}{1+s\beta}}(\log K)^{-\frac{\beta-1}{1+s\beta}}, \quad \mathcal{E}_{\mathrm{opt}} \asymp D^{-\frac{s\beta}{1+s\beta}}(\log K)^{\frac{s\beta-s}{1+s\beta}}.$$

When $s \leqslant 1 - \frac{1}{\beta}$, since $\gamma \lesssim 1$, we must have either $\gamma \asymp 1$ or $\gamma \asymp D^{-\frac{s\beta}{1+s\beta}} \tilde{D}_2^{\frac{\beta-1}{1+s\beta}} \lesssim 1$. To reach the minimum risk, in both cases we require $(\tilde{D}_2)_{\mathrm{opt}} \gtrsim D^{\frac{s\beta}{\beta-1}}$ (this gives $(D_2)_{\mathrm{opt}} \gtrsim D^{\frac{s\beta}{\beta-1}} \log D$), and

$$M_{\mathrm{opt}} = \infty, \quad \gamma_{\mathrm{opt}} \asymp 1, \quad \mathcal{E}_{\mathrm{opt}} \asymp D^{-s}.$$

Similarly, as $a \lesssim 1$, we have $B \lesssim_{\log} D^{1-\frac{\beta}{1+s\beta}}$, which means $K \gtrsim_{\log} D^{\frac{\beta}{1+s\beta}}$, hence $\log K \asymp \log D$, which gives the desired rate.

**Summary.** Combining the two cases together, we see that $M_{\mathrm{opt}} = \infty$ (case 2) always achieves the optimal rate, hence the conclusion follows. $\qquad\square$

**Theorem E.13.** *Assume $b = \frac{a}{K}$, under the compute constraint $C \gg 1$, the optimal strategy for minimizing the final population risk—expressed in terms of the effective maximum learning rate $\gamma$, model size $M$, and data size $D$—is given by:*

- *When $s > 1 - 1/\beta$, the optimal scaling laws are:*

$$\gamma_{\mathrm{opt}} \asymp \left(\frac{C}{\log C}\right)^{-\frac{1+s\beta-\beta}{2+s\beta}}, M_{\mathrm{opt}} \asymp \left(\frac{C}{\log C}\right)^{\frac{1}{2+s\beta}}, D_{\mathrm{opt}} \asymp C^{\frac{1+s\beta}{2+s\beta}}(\log C)^{\frac{1}{2+s\beta}}, (D_1)_{\mathrm{opt}} \asymp D, (D_2)_{\mathrm{opt}} \asymp D,$$

  *which leads to the following optimal final population risk:*

$$\mathcal{E}_{\mathrm{opt}} \asymp C^{-\frac{s\beta}{2+s\beta}}(\log C)^{\frac{s\beta-s}{2+s\beta}}.$$

- *When $s \leqslant 1 - 1/\beta$, the optimal scaling laws are:*

$$\gamma_{\mathrm{opt}} \asymp 1, M_{\mathrm{opt}} \asymp C^{\frac{1}{1+\beta}}, D_{\mathrm{opt}} \asymp C^{\frac{\beta}{1+\beta}}, (D_1)_{\mathrm{opt}} \asymp D, (D_2)_{\mathrm{opt}} \gtrsim D^{\frac{s\beta}{\beta-1}} \log D,$$

  *which leads to the following optimal final population risk:*

$$\mathcal{E}_{\mathrm{opt}} \asymp C^{-\frac{s\beta}{1+\beta}}.$$

*Proof.* Since the total intrinsic time $T \lesssim \gamma D$, we can always take $D_1 \asymp D$ to ensure $T \asymp \gamma D$. Denote $\tilde{D}_2 := \frac{D_2}{\log K}$, the loss can be reformulated as follows.

$$\mathcal{E}_K \asymp M^{-s\beta} + \frac{1}{(\gamma D)^s} + \sigma^2 \frac{\min\{M, (\gamma \tilde{D}_2)^{1/\beta}\}}{\tilde{D}_2}.$$

**Case 1.** $M^\beta < \gamma \tilde{D}_2$ and we have

$$\mathcal{E}_K \asymp M^{-s\beta} + \frac{1}{(\gamma D)^s} + \frac{M}{\tilde{D}_2}.$$

As $\gamma$ only appears in the second term, and $\frac{1}{(\gamma D)^s}$ is monotone decreasing with $\gamma$, we have that when $\mathcal{E}_K$ is minimized, it must hold that

$$M = (\gamma \tilde{D}_2)^{1/\beta}.$$

When $s > 1 - \frac{1}{\beta}$, we then consider a weighted AM-GM inequality, we have

$$M^{-s\beta} = \frac{M}{\tilde{D}_2}.$$

Combining with $M = (\gamma \tilde{D}_2)^{1/\beta}$, we have

$$\gamma_{\mathrm{opt}} \asymp \tilde{D}_2^{-\frac{1+\beta(s-1)}{1+s\beta}}, \quad M_{\mathrm{opt}} \asymp \tilde{D}_2^{\frac{1}{1+s\beta}}$$

and
$$\mathcal{E}_{\mathrm{opt}} \asymp \tilde{D}_2^{\,s - \frac{s\beta}{1+s\beta}} D^{-s}.$$

Notice that
$$C \asymp \tilde{D}_2^{\frac{1}{1+s\beta}} D \geqslant \tilde{D}^{\frac{2+s\beta}{1+s\beta}} \implies \mathcal{E} \gtrsim C^{-\frac{s\beta}{2+s\beta}}.$$

Note that this implies $D^{\frac{2+s\beta}{1+s\beta}} \gtrsim C \gtrsim D \implies \log D \asymp \log C$, and by similar reasons $\log K \asymp \log D$ (the max learning rate $B\gamma \lesssim 1$).

Hence when $\mathcal{E}$ is optimized, we have $\tilde{D}_2 \asymp D / \log C$ and
$$\gamma_{\mathrm{opt}} \asymp \left(\frac{C}{\log C}\right)^{-\frac{1+\beta(s-1)}{2+s\beta}}, \quad M_{\mathrm{opt}} \asymp \left(\frac{C}{\log C}\right)^{\frac{1}{2+s\beta}}, \quad D_{\mathrm{opt}} \asymp C^{\frac{1+s\beta}{2+s\beta}} (\log C)^{\frac{1}{2+s\beta}},$$

and
$$\mathcal{E}_{\mathrm{opt}}(C) \asymp \left(\frac{C}{\log C}\right)^{-\frac{s\beta}{2+s\beta}} (\log C)^s.$$

When $s \leqslant 1 - \frac{1}{\beta}$, since $a \lesssim 1$, we set $\gamma_{\mathrm{opt}} \asymp 1$ accordingly, and proceed as follows:
$$M_{\mathrm{opt}} \asymp \tilde{D}_2^{\frac{1}{\beta}}$$

and
$$\mathcal{E}_{\mathrm{opt}} \asymp D^{-s}.$$

Notice that
$$C \asymp \tilde{D}_2^{\frac{1}{\beta}} D \gtrsim \tilde{D}_2^{\frac{1+\beta}{\beta}} \implies \mathcal{E} \gtrsim C^{-\frac{s\beta}{1+\beta}}.$$

Hence when $\mathcal{E}$ is optimized, we have $\tilde{D}_2 \asymp D / \log C$ and
$$\gamma_{\mathrm{opt}} \asymp 1, M_{\mathrm{opt}} \asymp C^{\frac{1}{1+\beta}}, D_{\mathrm{opt}} \asymp C^{\frac{\beta}{1+\beta}},$$

and
$$\mathcal{E}_{\mathrm{opt}} \asymp C^{-\frac{s\beta}{1+\beta}}.$$

**Case 2.** $M^\beta \geqslant \gamma \tilde{D}_2$ and we have
$$\mathcal{E}_K \asymp M^{-s\beta} + \frac{1}{(\gamma D)^s} + \frac{(\gamma \tilde{D}_2)^{\frac{1}{\beta}}}{\tilde{D}_2}.$$

By AM-GM inequality,
$$(\gamma D)^{-s} + \gamma^{\frac{1}{\beta}} \frac{1}{\tilde{D}_2^{1-\frac{1}{\beta}}} \gtrsim D^{-\frac{s}{1+s\beta}} \tilde{D}_2^{-\frac{s\beta-s}{1+s\beta}},$$

with equality at $\gamma \asymp D^{-\frac{s\beta}{1+s\beta}} \tilde{D}_2^{\frac{\beta-1}{1+s\beta}}$.

When $s > 1 - \frac{1}{\beta}$, the equality can be achieved, hence $(D_2)_{\mathrm{opt}} \asymp D$, and the loss can be written as follows.
$$\mathcal{E}_K \asymp M^{-s\beta} + D^{-\frac{s}{1+s\beta}} \tilde{D}_2^{-\frac{s\beta-s}{1+s\beta}}$$

Combining with $C = MD$, we have the optimal scaling laws as follows:
$$\gamma_{\mathrm{opt}} \asymp C^{-\frac{1+s\beta-\beta}{2+s\beta}} (\log C)^{-\frac{\beta-1}{1+s\beta}}, M_{\mathrm{opt}} \asymp C^{\frac{1}{2+s\beta}} (\log C)^{-\frac{1-1/\beta}{2+s\beta}}, D_{\mathrm{opt}} \asymp C^{\frac{1+s\beta}{2+s\beta}} (\log C)^{\frac{1-1/\beta}{2+s\beta}},$$

which leads to the following optimal final population risk:
$$\mathcal{E}_{\mathrm{opt}} \asymp C^{-\frac{s\beta}{2+s\beta}} (\log C)^{\frac{s\beta-s}{2+s\beta}}.$$

When $s \leqslant 1 - \frac{1}{\beta}$, since $\gamma \lesssim 1$, we must have either $\gamma \asymp 1$ or $\gamma \asymp D^{-\frac{s\beta}{1+s\beta}} \tilde{D}_2^{\frac{\beta-1}{1+s\beta}} \lesssim 1$. To reach the minimum risk, in both cases we require $(\tilde{D}_2)_{\mathrm{opt}} \gtrsim D^{\frac{s\beta}{\beta-1}}$ (this gives $(D_2)_{\mathrm{opt}} \gtrsim D^{\frac{s\beta}{\beta-1}} \log D$), and
$$\gamma_{\mathrm{opt}} \asymp 1, \quad \mathcal{E}_K \asymp M^{-s\beta} + D^{-s}.$$

Combining with $C = MD$, we have the optimal scaling laws as follows:
$$\gamma_{\mathrm{opt}} \asymp 1, M_{\mathrm{opt}} \asymp C^{\frac{1}{1+\beta}}, D_{\mathrm{opt}} \asymp C^{\frac{\beta}{1+\beta}},$$

which leads to the following optimal final population risk:
$$\mathcal{E}_{\mathrm{opt}} \asymp C^{-\frac{s\beta}{1+\beta}}.$$

**Summary.** Combining the results of each case, we get the desired optimal scaling strategy stated in the theorem. □

## F  Auxiliary Lemmas

**Lemma F.1.** *For any PSD matrix* $\mathbf{A}$ *and a random gaussian vector* $\mathbf{x} \sim \mathcal{N}(0, \mathbf{H})$,

$$\mathrm{tr}(\mathbf{HA})\mathbf{H} \preceq \mathbb{E}\left[\mathbf{x}\mathbf{x}^\top \mathbf{A}\mathbf{x}\mathbf{x}^\top - \mathbf{HAH}\right] = \mathrm{tr}(\mathbf{HA})\mathbf{H} + \mathbf{HAH} \preceq 2\mathrm{tr}(\mathbf{HA})\mathbf{H}$$

*Proof.* Assume $\mathbf{A} = (A_{ij})_{i,j=1,\dots,M}$. The $(i,j)$-th entry of $\mathbf{x}\mathbf{x}^\top \mathbf{A}\mathbf{x}\mathbf{x}^\top$ is

$$\sum_{k,l} \mathbf{x}_i \mathbf{x}_k A_{kl} \mathbf{x}_l \mathbf{x}_j.$$

If $i \neq j$,

$$\mathbb{E}\left[\sum_{k,l} \mathbf{x}_i \mathbf{x}_k A_{kl} \mathbf{x}_l \mathbf{x}_j\right] = 2\mathbb{E}\left[A_{ij}\mathbf{x}_i^2 \mathbf{x}_j^2\right] = 2A_{ij}\lambda_i\lambda_j = 2\mathbf{HAH}(i,j).$$

If $i = j$

$$\mathbb{E}\left[\sum_{k,l} \mathbf{x}_i \mathbf{x}_k A_{kl} \mathbf{x}_l \mathbf{x}_j\right] = \mathbb{E}\left[\sum_{k=1}^{M} A_{kk}\mathbf{x}_i^2 \mathbf{x}_k^2\right] = \sum_{k \neq i} A_{kk}\lambda_i\lambda_k + 3A_{ii}\lambda_i^2 = 2\mathbf{HAH}(i,i) + \mathrm{tr}(\mathbf{HA})\mathbf{H}.$$

By the trace inequality we have

$$\mathbf{HA} \preceq \mathrm{tr}(\mathbf{HA}).$$

Multiplying $\mathbf{H}$ at both sides,

$$\mathbf{HAH} \preceq \mathrm{tr}(\mathbf{HA})\mathbf{H}.$$

Combining the results, we have

$$\mathbb{E}[\mathbf{x}\mathbf{x}^\top \mathbf{A}\mathbf{x}\mathbf{x}^\top] = \mathrm{tr}(\mathbf{HA})\mathbf{H} + 2\mathbf{HAH} \preceq 2\mathrm{tr}(\mathbf{HA})\mathbf{H} + \mathbf{HAH}.$$

□

**Lemma F.2.** *Let* $\mathbf{P} \preceq \mathbf{Q}$ *be two PSD matrices. Then for any PSD matrix* $\mathbf{U}$*, we have*

$$\mathrm{tr}(\sqrt{\mathbf{P}}\mathbf{U}\sqrt{\mathbf{P}}) \leqslant \mathrm{tr}(\sqrt{\mathbf{Q}}\mathbf{U}\sqrt{\mathbf{Q}}).$$

*Proof.* It is clear that $\mathrm{tr}(\sqrt{\mathbf{P}}\mathbf{U}\sqrt{\mathbf{P}}) = \mathrm{tr}(\mathbf{UP})$ and

$$\mathrm{tr}(\mathbf{UQ}) - \mathrm{tr}(\mathbf{UP}) = \mathrm{tr}(\mathbf{U}(\mathbf{Q} - \mathbf{P})) \geqslant 0,$$

since $\mathbf{U}$ and $\mathbf{Q} - \mathbf{P}$ are both PSD matrices. □

