# OpenReview forum: "Functional Scaling Laws in Kernel Regression: Loss Dynamics and Learning Rate Schedules"
_NeurIPS.cc/2025/Conference — NeurIPS 2025 spotlight_

### Official Review · Reviewer_Ss9R · 2025-06-07

**Clarity:** 3
**Significance:** 3
**Originality:** 3
**Rating:** 5
**Confidence:** 2

**Summary:**

This paper proposes functional scaling laws, and extension of existing LLM scaling laws, to account for learning rate schedules. This novel scaling law additionally allows analysis of loss during the intermediate-steps during training, whereas existing scaling laws only account for the final-step loss. The authors leverage the SDE view of SGD to provide theoretical analysis based on teacher-student kernel regression setting for three different learning rate schedules.

**Questions:**

- Is it possible to consider generalization bounds as in [1]? Additionally it would be nice to add discussion on this.


[1] Finzi, Marc, et al. "Compute-Optimal LLMs Provably Generalize Better with Scale." https://arxiv.org/abs/2504.15208.

**Ethical Concerns:**

["NO or VERY MINOR ethics concerns only"]

**Final Justification:**

Scaling laws are outside of my expertise, but the author's application of SDE analyses to this field is thorough and novel. Having read the other reviews and replies, I'm happy to keep my positive score.

**Limitations:**

yes

**Quality:**

4

**Strengths And Weaknesses:**

The field of scaling laws is outside the scope of my expertise, so I primarily focused on the theory part of the paper.

**Strengths**:
- The paper is well-written and straightforward to follow.
- The theoretical results are well-justified: the proofs are convincing at a high level, even though I did not check every detail.
- Experimental validation matches the theory.
- Discussion and limitations of assumptions are well discussed.


**Weaknesses**:
- Empirical results are on relatively small models, but this limitation is discussed and the paper's contribution is mainly theoretical.

---

> ### Author Rebuttal · Authors · 2025-07-31
>
> We sincerely thank the reviewer for the positive support and valuable feedback! We greatly appreciate the insightful review, and the recognition of highlighting the significance of our contribution and the strength of our theory and experiments, as well as the clarity of our writing. We are very glad to address the questions and suggestions raised by the reviewer, which we believe will help further refine our work. Below are our responses to the questions and suggestions raised by the reviewer.
>
> ---
>
> >**[Q]** Is it possible to consider generalization bounds as in [1]? Additionally it would be nice to add discussion on this.
>
> **[A]** We thank the reviewer for the valuable comment. Our work focuses on the online (one-pass) SGD setting, where each data point is used only once. This reflects the standard practice in modern LLM pre-training, where massive datasets are typically passed through only once due to their scale. In this setting, traditional notions of in-distribution generalization—such as the gap between training and validation loss—are less relevant. Instead, generalization is often assessed in terms of out-of-distribution performance on downstream tasks. As a result, there is effectively no distinction between training and validation loss during pre-training, and traditional in-distribution generalization bounds do not directly apply. We hope this clarification addresses the reviewer’s concern.
>
> ---
> **Reference**
>
> [1] Compute-Optimal LLMs Provably Generalize Better with Scale.

---

> > ### Comment · Reviewer_Ss9R · 2025-08-01
> >
> > I thank the reviewers for their response. I will keep my score.

---

> > > ### Author Response · Authors · 2025-08-01
> > >
> > > Thank you very much for your kind review, and we are glad that you enjoyed our paper!

---

### Official Review · Reviewer_HjUT · 2025-06-20

**Clarity:** 3
**Significance:** 2
**Originality:** 3
**Rating:** 4
**Confidence:** 4

**Summary:**

This paper analyzes the scaling behaviour of the risk for a teacher-student regression problem when using different learning-rate schedules.
The paper shows that the theoretical model describes well the actual dynamics of SGD in the same teacher-student setting, but also for LLM pretraining (where the assumptions of the theory do not hold). As an application, the paper derives a optimal schedule from fitting their theoretical law, and shows that this schedule also works well in practice.

**Questions:**

* You claim (line 93) that you provide the first theoretical explanation for different scaling behaviours across LR schedules. I am not sure whether this is accurate: the scaling with data can be seen basically as the convergence rate of SGD. Studying convergence rates for LR schedules has been done previously in the optimization community (even though it is not termed a scaling law). For example, this paper (https://arxiv.org/pdf/2501.18965) shows an improved rate for WSD vs. constant schedule, which essentially also implies a different scaling in data budget. Of course the specific setting of your paper allows for much more insights with respect to model capacity, batch sizes etc, thus the contributions of your submission are still very much useful.

* Regarding the sentence:

> As a result, different LRSs can induce distinct scaling behaviors, with the scaling exponent itself potentially depending on the schedule.

Is there actually empirical evidence for this? The only reference I know on this is https://arxiv.org/abs/2406.19146, where it is shown that the scaling law for constant schedule is almost the same as for cosine schedule (under suitable tuning of other hyperparameters).

* Can you analyse the WSD schedule with a linear cooldown (or other forms that exponential in general)? If not, what are the limiting factors in the analysis? This is relevant as the prevalent cooldown shape in practice seems to be linear or 1-sqrt (see the paper by Hägele et al, 2024).

* Caption of Figure 2 says:

> In all panels, dashed curves represent the theoretical predictions from Section 5. (a) Solid curves show the risk predicted via numerical simulation using the FSL.

This was confusing, as I thought that the FSL is the theoretical prediction, and the solid curves come from actual SGD runs. What am I missing here?

* The FSL theory applies to SGD, but is then used to fit Adam runs in the LLM setup. Could you comment on this limitation?

* You are fitting the values of $s$ and $\beta$ in the LLM experiment. Do the scaling coefficients obtained from the fitted values agree with the scaling laws found for LLM pretraining in practice (e.g. the Chinchilla law)?

* The emphasis in the abstract on batch size schedule is very strong, but the main text does not provide many insights into optimal (?) batch size schedules, or empirical validation of the theory regarding the batch size.

* line 245 says that LLM practice is to allocate more compute needs to data. But in contrast, the Chinchilla scaling law concludes that the powerlaw coefficient in data and model is the same?

Minor:
- line 98: "up to a constant factor": multiplicative or additive factor?
- Equation (1): I think there should be a factor 1/2 in front of the squared norm to be consistent
- line 225: how is compute $C$ defined here?
- line 248: typo "brevity"

**Ethical Concerns:**

["NO or VERY MINOR ethics concerns only"]

**Final Justification:**

Based on my initial review, concerns regarding practicality and whether the current experiments reveal the full picture remain. The authors were also transparent about this in the discussion period. Therefore my borderline score remains.

**Limitations:**

No potential negative societal impacts.

**Paper Formatting Concerns:**

-

**Quality:**

3

**Strengths And Weaknesses:**

The structure and presentation of the paper are very clear, and the main objective can be easily grasped (even though the mathematical details are technical).

For the experimental validation, some additional efforts are needed:

* From the LLM experiments it seems that the FSL fit is predictive across schedules (fit on 811, then predict cosine and WSD). This is great, but something that is missing here is the role of the peak learning rate. Usually, for different schedules the optimal peak LR is different. In your experiments, did you tune the peak LR for all schedules? Can the FSL also predict the loss curves across peak learning rate? This discussion is missing, but for practical applications it is highly relevant, as the (peak) learning rate will most likely be the most important hyperparameter to tune.

* The optimal LRS for the LLM experiment has an unfair advantage in the sense that the other schedules do not decay to zero, but a 10% of the peak LR. However, it is known (see your references 7,21) that cooldown to zero gives a benefit, and in fact for WSD (see [21]) this is what was proposed from the beginning. Hence, in order to properly assess whether the optimal schedule brings a benefit, the baseline should be cosine and WSD schedules that decay to zero. It is however a nice finding, that the theoretically optimal schedule decays to zero (but again this has been previously found in other settings, see for example https://arxiv.org/pdf/2310.07831).

* Related to the above, the LLM experiments use the same peak LR for all schedules. It has been shown empirically and theoretically that different schedules will have different optimal peak LRs, hence the missing tuning of the peak LR might confound your findings.

**Ablating on the role of the peak and final learning rate, in particular their effect on the experiments provided, is the main revision that I suggest to improve this submission.**

* A minor weakness regarding clarity: some important information, on which constants are exactly fitted (in Sec 6.2), and which runs form the basis of the fit (in Sec. 6.1) are only provided in the appendix. However, this information would be insightful when reading the main text.

* Another minor point: the connection between the discrete schedule and peak LR, and the continuous LRS $\phi$ does not become fully clear and needs further explaining. For example, in line 173, the maximal argument for $\varphi$ seems to be $\tau$, the number of training steps, but then in line 169 it seems that the argument of $\varphi$ is the step counter times $h$. So on which time scale is $\varphi$ defined and how does that scale related to the (integer) SGD iteration counter?

---

> ### Author Rebuttal · Authors · 2025-07-31
>
> We sincerely thank the reviewer for the positive support and valuable feedback! We greatly appreciate the insightful review, recognizing the significance of our contribution, the solidity of our theory, and the clarity of our writing. We are delighted to address the questions and suggestions raised by the reviewer, which we believe will help further refine our work. Below are our responses.
>
> ---
>
> >**[W1]** In your experiments, did you tune the peak LR for all schedules? Can the FSL also predict the loss curves across peak learning rate?
>
> **[E1]** Due to the high computational cost and limited resources, we could not perform extensive tuning in our LLM experiments. The primary goal of these experiments was to demonstrate FSL's potential in predicting pre-training loss curves. Developing a practical version of FSL remains a key focus of our ongoing work.
>
> In SGD experiments on linear regression, we found that FSL predictions generalize well across different peak learning rates. With hyperparameters $M = 256$, $s = 1.5$, $\beta = 3$, and $10^4$ training steps, we predicted risk curves for the WSD and cosine schedules with a peak learning rate of $10^{-2}$, using coefficients fitted on WSD with a peak learning rate of $10^{-3}$. The resulting RMSE for these predictions was $1.52 \times 10^{-4}$ and $2.31 \times 10^{-4}$, respectively. For comparison, the RMSE with coefficients fitted at $10^{-2}$ was $1.45 \times 10^{-4}$ and $2.34 \times 10^{-4}$. These results show that FSL generalizes well across peak learning rates.
>
> $ $
> >**[W2]** Hence, in order to properly assess whether the optimal schedule brings a benefit, the baseline should be cosine and WSD schedules that decay to zero.
>
> **[E2]** We clarify that we have already verified the effect of the minimal learning rate in additional experiments, as shown in Appendix, Figure 10. Our results demonstrate that the optimal schedule provides a benefit, even when compared to WSD which exponentially decays to $10^{-7}$. We will add the discussion of minimal learning rates in the revised version.
>
> $ $
> >**[W3]** It has been shown empirically and theoretically that different schedules will have different optimal peak LRs, hence the missing tuning of the peak LR might confound your findings.
>
> **[E3]** In our SGD experiments, we did tune the peak learning rates for different LRSs, as shown in Figure 2(b) (right). Therefore it is a fair comparison with peak learning rates into consideration, and it shows that good schedules result in a lower final risk.
>
> However, for the LLM experiments, tuning the peak learning rate for each LRS would be too costly. As our work primarily focuses on theoretical analysis, we did not have sufficient computational resources to conduct these additional experiments.
>
> $ $
> >**[W4]** A minor weakness regarding clarity:...
>
> **[E4]** We clarify that this work primarily focuses on theoretical analysis, and implementation details were placed in the appendix due to space constraints. In the revised version, we will incorporate the relevant information into Section 6 to improve clarity and accessibility.
>
> $ $
> >**[W5]** Another minor point:....
>
> **[E5]** There is a typo in line 173 causing the confusion. The $\tau$ stands for the **continuous** training steps, and the true training step of SGD should be $\tau / h$. $\varphi$ is defined on the continuous training steps, which is $h$ times the discrete steps as defined in line 169.
>
> $ $
> >**[Q1]** You claim (line 93) that you provide the first theoretical explanation for different scaling behaviours across LR schedules.
>
> **[A1]** We clarify that [1] shows that the WSD schedule yields a logarithmic factor improvement over constant LR, but only in terms of worst-case upper bounds on the convergence rate for non-smooth convex problems. Since the comparison is between two upper bounds, it does not provide a theoretical justification for the benefit of WSD.
>
> We also highlight the distinction between scaling laws and “convergence rates” in optimization. The latter typically defines a family of problems (e.g., non-smooth convex problems) and derives an upper bound on the convergence rate that holds uniformly across the family. A matching lower bound may be established to demonstrate minimax optimality. However, such worst-case bounds do not necessarily reflect the actual convergence behavior for a specific problem. In contrast, scaling laws are derived for a fixed problem instance, enabling a more accurate characterization of convergence behavior.
>
> $ $
> >**[Q2]** The only reference I know on this is https://arxiv.org/abs/2406.19146, …
>
> **[A2]** Indeed, in their work [11] they did not compare the scaling exponent of constant and cosine schedule directly, and their training dataset is different from Chinchilla, thus the comparison is unreasonable.
>
> $ $
> >**[Q3]** Can you analyse the WSD schedule with a linear cooldown (or other forms than exponential in general)?
>
> **[A3]**  We can analyze the linear cooldown for the WSD schedule with a fixed ratio of linear cooldown (e.g., a 20% cooldown ratio). If the minimal learning rate is set to zero, we have the following:
>
> * If $\beta \geq 2$:  $\mathcal{E}_K \asymp M^{-s\beta} + T^{-s} + \sigma^2 \frac{1}{T^{\frac{1}{2}}}.
> $
>
> * If $\beta \in (1,2)$:   $\mathcal{E}_K \asymp M^{-s\beta} + T^{-s} + \sigma^2 \frac{\min\left(1, (M^{-\beta} T)^{ \frac{1}{2} - \frac{1}{\beta}}\right)}{T^{\frac{1}{\beta}}}.
> $
>
> From these expressions, we can proceed to derive both the data-optimal scaling and the compute-optimal scaling.
>
> $ $
> >**[Q4]** Caption of Figure 2 ...
>
> **[A4]** We apologize for the confusion and will clarify the caption. The derivation of the “theoretical predictions” $D^{-\alpha}$ consists of two steps: (1) SDE modeling of SGD, and (2) deriving $D^{-\alpha}$ from FSL. In Figure 2(a), the solid line shows the prediction from simulating FSL, validating the second step. In Figure 2(b), the solid line represents the prediction from simulating SGD, confirming the accuracy of the SDE modeling.
>
> $ $
> >**[Q5]** The FSL theory applies to SGD, but is then used to fit Adam runs in the LLM setup. Could you comment on this limitation?
>
> **[A5]** We clarify this discrepancy from three perspectives:
>
> First, while our FSL theory is based on SGD—common in prior studies on scaling laws [2–4]—we view this as a foundational step. Our contribution builds on this by incorporating the influence of learning rate schedules, potentially offering insights into how adaptive methods like Adam improve scaling efficiency.
>
> Second, the fact that our FSL, derived under SGD assumptions, still fits Adam-based LLM runs reasonably well is a surprising and intriguing empirical observation. It suggests that despite Adam’s adaptive nature, its behavior may resemble that of SGD in some respects. This aligns with recent findings: (1) SGD variants can match Adam's performance on training Transformer models [5–7,10], though scale is still limited; (2) most LLM parameters can be trained effectively using SGD [8].
>
> Third, the main practical limitation is modeling the interaction between learning rate and batch size. For SGD, the learning rate is typically scaled linearly with batch size, while for Adam, square-root scaling is recommended, informed by the signSGD interpretation of Adam. Extending our FSL framework to accurately predict batch size scaling for adaptive optimizers like Adam is complex and remains a key area for future work.
>
> We will incorporate these into the discussion section of the revision of our paper.
>
> $ $
> >**[Q6]** You are fitting the values of $s$ and $\beta$ …
>
> **[A6]** We clarify that in our LLM experiments, we focused solely on the learning rate schedule and did not fit the scaling with respect to model size. Our FSL framework, derived from kernel regression, suggests a model-size scaling of $M^{-s\beta}$, indicating that the scaling exponent for model size depends on data size scaling. However, we expect this relationship may not hold in practice for LLM tasks.
>
> Nonetheless, our theoretical results in Section 5 (lines 237, 262, 305) consistently show that, across all three learning rate schedules, the compute-optimal strategy prioritizes increasing data size over model size. Specifically, the data-size scaling exponent slightly exceeds the model size exponent, aligning qualitatively with the Chinchilla law (data: 0.54, model: 0.46).
>
> $ $
> >**[Q7]** The emphasis in the abstract on batch size schedule is very strong…
>
> **[A7]** Due to the space limit, we refer to the response **E2** for **W2** of Reviewer SA1x.
>
> $ $
> >**[Q8]** line 245 says …
>
> **[A8]** We clarify that the Chinchilla law [8] asserts that the power coefficient in data is slightly higher than that in the model, which is consistent with our theoretical results. Specifically, in the Chinchilla paper, the fitted scaling law is (equation (10) in [8])
> $$
> L(M, D) = L_0 + \frac{A}{M^{0.34}} + \frac{B}{D^{0.28}}.
> $$
>
> $ $
> >**[Q9]** Minor typos.
>
> **[A9]** We will correct them in the revised version of our paper. The constant factor in line 98 is multiplicative.
>
> ---
> **Reference**
>
> [1] The Surprising Agreement Between Convex Optimization Theory and Learning-Rate Scheduling for Large Model Training.
>
> [2] Scaling Laws in Linear Regression: Compute, Parameters, and Data.
>
> [3] 4+3 Phases of Compute-Optimal Neural Scaling Laws.
>
> [4] HOW FEATURE LEARNING CAN IMPROVE NEURAL SCALING LAWS.
>
> [5] Noise is not the main factor behind the gap between sgd and adam on transformers, but sign descent might be.
>
> [6] No more Adam: Learning rate scaling at initialization is all you need.
>
> [7] Adam-mini: Use Fewer Learning Rates To Gain More.
>
> [8] Deconstructing what makes a good optimizer for autoregressive language models.
>
> [9] Training compute-optimal large language models.
>
> [10] Small Batch Size Training for Language Models: When Vanilla SGD Works, and Why Gradient Accumulation Is Wasteful.
>
> [11] Resolving Discrepancies in Compute-Optimal Scaling of Language Models.

---

> > ### Comment · Reviewer_HjUT · 2025-08-05
> > **Response to rebuttal**
> >
> > Dear authors,
> >
> > thank you for your detailled answers to each of my questions. Here are some points that I think are worth further discussion:
> >
> > * as pointed out in my review and also by Reviewer 93yY, the paper currently lacks detailled discussion of related work that also shed light on the behaviour of LR schedules and their scaling behaviour. The theoretical framework of each of these works has its natural limitations (as you pointed out), as well as has the framwork of your paper. Discussing these different setups, and how they lead to the same or different conclusions should be better highlighted in my opinion. On the other hand, claiming the "first theoretical explanation" seems an overstatement and does not lead to an additional understanding of your contributions in the context of previous work.
> >
> > * On the LLM experiments:
> >
> > > However, for the LLM experiments, tuning the peak learning rate for each LRS would be too costly. As our work primarily focuses on theoretical analysis, we did not have sufficient computational resources to conduct these additional experiments.
> >
> > It is now widely accepted that seemingly superior optimization techniques are often just due to undertuned baselines, especially when the (peak) learning rate is not properly tuned. Given this, and the fact that your LLM experiments do not tune the LR for each schedule individually, I would strongly advise to make this as clear as possible in the paper. Any (dis)advantage in your runs might be just a spurious effect of this missing tuning step. As pointed out in my initial review, this issue strongly reduces the practical implications of the paper for me; however, the theoretical contributions are of course not affected by this.

---

> > > ### Author Response · Authors · 2025-08-05
> > > **Thanks for the  Thoughtful Follow-up Comments**
> > >
> > > Dear Reviewer,
> > >
> > > Thank you for your thoughtful follow-up comments. We truly appreciate the opportunity to further clarify our work.
> > >
> > > - **Regarding the claim of "first theoretical explanation":** We fully agree that such a statement is inappropriate in scientific writing. In the revised version, we will remove this claim **entirely** and instead focus on clearly articulating the differences in setup, assumptions, conclusions, and limitations among existing explanations. We recognize that such absolute language does not contribute meaningfully to the discussion and will ensure our writing remains factual and precise.
> > >
> > > - **Regarding the experimental setup.** We will clarify in the revision that all experiments were conducted using the same peak learning rate. We also acknowledge and will discuss the limitations of our current experimental design.
> > >
> > > - **On the practicality of the FSL framework.** This paper primarily focuses on the theoretical aspects of FSL. The LLM experiments are intended to illustrate the potential practical relevance of the framework, rather than to propose a fully practical solution. Developing a truly practical FSL involves several additional challenges, such as fitting from short runs to predict long-run behavior, or adapting to varying peak learning rates. We fully agree that the current version of FSL presented in this paper is not yet “practical,” and we will revise the text to reflect this more clearly.
> > >
> > > Thank you again for your valuable feedback.
> > >
> > > Best regards,\
> > > The Authors

---

### Official Review · Reviewer_SA1x · 2025-07-03

**Clarity:** 3
**Significance:** 3
**Originality:** 3
**Rating:** 5
**Confidence:** 3

**Summary:**

Scaling law analyses usually only consider the final loss after training and overlook the effect of learning rate schedule on the training trajectory. This work derives a functional scaling law from a teacher-student kernel regression setting, where the impact of learning rate and batch size schedules is given via a convolution-type noise term. For the constant, exponential decay, and WSD schedules, the authors compute their scaling exponents under data-limited and compute-limited regimes, where the theory aligns with empirical observations in LLM pretraining.

**Questions:**

* Do the authors foresee any trends with the optimal FSL LR schedule with model scale? The optimal schedule for the 100M and 1B models appear similar.
* What is the reason for using the 8-1-1 LRS for the fitting of the FSL as opposed to the other schedulers?
* The FSL incorporates batch-size scheduling, but there doesn’t seem to be mentioning of batch size scheduling later in the paper. Given that batch size scheduling is common for large-scale model pretraining (eg. Llama 3 increases the batch size twice during training), how could the FSL fit an optimal LR schedule in light of varying batch sizes during training?

**Ethical Concerns:**

["NO or VERY MINOR ethics concerns only"]

**Final Justification:**

The authors have done additional experiments looking at the predictive power of their FSL framework during the rebuttal period. Although their theory would have to be extended for adaptive optimizers, I believe the result is of important interest and will lead to follow-up work.

**Limitations:**

Yes

**Quality:**

4

**Strengths And Weaknesses:**

Strengths:
* The paper is generally well-written with the main result and subsequent applications to specific schedules presented clearly.
* Studying the interaction between learning rate schedules and scaling laws is an important problem.
* This work provides a theoretical derivation of a scaling law that is aware of the learning-rate schedule, where previous work similarly tackling this problem were largely heuristic-based.
* Empirical results in the teacher-student kernel regression setting show strong agreement with theory and the FSL-optimal schedule appears competitive for language model pretraining.

Weaknesses:
* The FSL-optimal LR requires fitting a ‘practical’ version of the FSL to an existing loss curve and is not predictive. This has to be done for a specific model scale, duration and maximal learning rate; the method would be more compelling if a trend emerged across model scale that allowed for extrapolation instead of requiring a full training run.
* The practical FSL used for fitting places some assumptions, particularly assuming a fixed batch size; although the authors mention their FSL accounting for batch size schedule as well, this isn’t explored further.
* There still is a gap in understanding what is the optimal LRS is for a given training setting– the optimal LRS identified by the practical FSL all seem to be of a similar nature but a more precise characterization depending on model architecture, batch size, dataset size, etc. would be valuable.

I did not read the theoretical proofs in detail. Despite what I’ve written above, I do believe the results presented are a novel step in this direction and the general framework could be of interest to the broader community.

---

> ### Author Rebuttal · Authors · 2025-07-31
>
> We sincerely thank the reviewer for the encouraging and insightful feedback, and for highlighting the strength of our theoretical contributions and the clarity of our writing. We are very glad to address the questions and suggestions raised by the reviewer, which we believe will help further refine our work. Below are our responses to the questions and suggestions raised by the reviewer.
>
> ---
> >**[W1]** The FSL-optimal LR requires fitting a ‘practical’ version of the FSL to an existing loss curve and is not predictive. This has to be done for a specific model scale, duration and maximal learning rate; the method would be more compelling if a trend emerged across model scale that allowed for extrapolation instead of requiring a full training run.
>
> **[E1]** We thank the reviewer for these insightful suggestions and fully agree that improving the predictive power of FSL across model scales,  durations, and maximal learning rates is essential for making it practical.
>
> In response, we trained a 100M-parameter model and find that the fitted FSL  generalizes well across different durations. Specifically, we fit a FSL using a 20B-token training run and successfully predict loss curves for 80B tokens, achieving a consistently low RMSE of 0.0120 across durations.
>
> To extend FSL across model scales, we follow the dependency modeling approach of [2]. On the positive side, we observe some degree of generalization. However, due to the lack of a principled design, the fitted FSL tends to fail when extrapolating to model scales far from those used in fitting—at least in our current experiments.
>
> In summary, addressing these problems remains challenging and will require a quite amount of effort. We will add a discussion in the revised version to highlight these directions for future research toward making FSL practical.
>
> $\quad$
> >**[W2]** The practical FSL used for fitting places some assumptions, particularly assuming a fixed batch size; although the authors mention their FSL accounting for batch size schedule as well, this isn’t explored further.
>
> **[E2]** We thank the reviewer for raising this point and would like to clarify our rationale for focusing on a fixed batch size in this paper.
>
> First, the study of learning rate schedules (LRSs) already involves substantial technical development and occupies significant space.
>
> Second,  the FSL framework is derived for SGD but practical LLM training typically relies on adaptive optimizers, where the interaction between learning rate and batch size can differ substantially. Understanding how insights from the FSL framework translate to such settings would require considerable additional theoretical and empirical analysis. For these reasons, we believe that an in-depth study of batch size schedules is best addressed in a separate work. In the revision, we will clarify this point in a discussion section.
>
> That said, we have already conducted some preliminary exploration of batch size scheduling. When the learning rate is fixed, our analysis reveals that the optimal batch size schedule should be: $B_{k} \asymp  (K-k)^{2-1/\beta}$ in the data-optimal regime for the learning problem we considered. This indicates that the batch size should gradually increase over the course of training. To validate this insight, we trained a LLaMA-0.5B model on the C4 dataset for 10B tokens using a constant learning rate of 1e−3, and compared different batch size schedules:
> | setting | final loss |
> | --- | --- |
> | all tokens using 512 batch size | 2.9 |
> | all tokens using 1024 batch size | 2.86 |
> | 32% tokens using 512, 68% tokens using 1024 batch size | 2.83 |
>
> These results support the theoretical insight that increasing batch size over time can improve performance.
>
> $\quad$
> >**[W3]** There still is a gap in understanding what is the optimal LRS is for a given training setting– the optimal LRS identified by the practical FSL all seem to be of a similar nature but a more precise characterization depending on model architecture, batch size, dataset size, etc. would be valuable.
>
> **[E3]** We thank the reviewer for the insightful comments. We conducted additional experimental analysis and found that the fitted FSL-optimal LRS consistently follow a WSD pattern with power decay of the form:
> $$
> f(k - K_{stable}) = \left( \frac{K_{total} - k}{K_{total} - K_{stable}} \right)^\gamma,
> $$
> We further explored how the optimal exponent $\gamma$ depends on two factors:
>
> - **Training tokens**: For a 100M-LLaMA model, we fit the optimal $\gamma$ for different training tokens:
> | Dataset Size | 10B  | 20B  | 40B  | 80B   |
> |-------------|------|------|------|-------|
> | $\gamma$    | 1.91 | 1.59 | 1.86 | 2.15  |
>
> - **Model size and architecture:** We trained a 1B-LLaMA and 1B-MoE model for 20B tokens. The optimal $\gamma$ are $1.40$ and $1.28$, respectively.
>
> Although the optimal $\gamma$ varies with tokens, model size, and architectures, it is consistently greater than 1, suggesting that convex decay outperforms concave decay.
>
> A comprehensive analysis of these trends is an important direction for future research.
>
> $\quad$
> >**[Q1]** Do the authors foresee any trends with the optimal FSL LR schedule with model scale? The optimal schedule for the 100M and 1B models appear similar.
>
> **[A1]** See [E3] for experimental results on the shape of optimal LRS. According to our preliminary experiments, we conjecture that optimal LRS takes the form of power-decay, and the exponent $\gamma$ is smaller for larger model sizes: for 100M-LLaMA models, the optimal $\gamma$ is around $2$, while for the 1B LLaMA and MoE, the optimal $\gamma$ is around $1.5$. The detailed analysis is beyond the scope of our paper, and we consider this an important future direction.
>
> $\quad$
> >**[Q2]** What is the reason for using the 8-1-1 LRS for the fitting of the FSL as opposed to the other schedulers?
>
> **[A2]** We thank the reviewer for the valuable comment. The 8-1-1 LRS is used due to its simplicity, being a multi-stage piecewise constant schedule. However, we acknowledge that other schedules can also be used for fitting with similar results. For example, we conducted fitting experiments on a 100M model trained with 20B tokens using WSD and cosine schedules, then predicted the loss curve of these 3 schedules (8-1-1, WSD, cosine). The average RMSE of these predictions was 0.04615 and 0.04916 for coefficients fitted with WSD and cosine, respectively, while the prediction RMSE for coefficients fitted with the 8-1-1 schedule was 0.04617.
>
> $\quad$
> >**[Q3]** doesn’t seem to be mentioning of batch size scheduling later in the paper. Given that batch size scheduling is common for large-scale model pretraining (eg. Llama 3 increases the batch size twice during training), how could the FSL fit an optimal LR schedule in light of varying batch sizes during training?
>
> **[A3]** First, we refer to our response **[E3]** for weakness **[W3]**. Second, as for identifying an optimal LR schedule under varying batch sizes, we agree this is an important direction for future work. However, the current FSL framework is derived for SGD, whereas practical LLM training typically uses Adam, where the interaction between learning rate and batch size can differ significantly. Extending FSL to this setting would require substantial additional theoretical and empirical development, which we believe falls beyond the scope of the current paper and is best pursued in future research.
>
> ---
>
> **Reference**
>
> [1] A Multi-Power Law for Loss Curve Prediction Across Learning Rate Schedules.
>
> [2] Scaling Law with Learning Rate Anneal.

---

> > ### Comment · Reviewer_SA1x · 2025-08-04
> >
> > I thank the authors for their response and for the additional results. They have addressed my concerns and am happy to raise my score to 5, accept.

---

> > > ### Author Response · Authors · 2025-08-04
> > >
> > > Thank you very much for your kind review, and we are glad that you enjoyed our paper!

---

### Official Review · Reviewer_93yY · 2025-07-05

**Clarity:** 3
**Significance:** 3
**Originality:** 2
**Rating:** 4
**Confidence:** 4

**Summary:**

This paper considers how learning rate and batch size influence the loss trajectory in a teacher-student kernel regression setup. The setup considers learning teacher function $f^{\*}(x) = \langle \phi(x), \theta^{\*} \rangle$ where $\phi(x)$ is an $N$ dimensional feature map and $x \in R^d$ with noise $\sigma^2$, with a student model $f(x; v) =  \langle W \phi(x), v \rangle $, where W is either a random $M$ dimension projection or the top $M$ feature of the feature map $\phi$. M is the concept of model size here. The paper then introduce the concept of model capacity by defining $\phi_j = \lambda_j^{1/2} e_j$ where $e_j$ is an orthonormal function basis and $\lambda_j$ follow $j^{-\beta}$ scaling. With lower $\beta$, the feature decays slower and hence is a 'richer' model. The authors then assume $\theta_j$ follows $j^{-1/2}\lambda_j^{(s - 1)/2}$ for different $j$ so that the function become $j^{-(s\beta + 1)} e_j$ and $s$ here characterize the relative difficulty.

After establishing this setup, the authors consider a continuous time approximation, using SDE to simulate the discrete optimization process. Given the SDE, it is then possible to give a sharp characterization of the loss scaling of the model sizes and data sizes, with a form similar to $1/M^{s\beta} + 1/t^s + Noise$. They then consider how different learning rate corresponds to different continuous time $t$ (which is an integration of the learning rate) and noise level (which is a convolution of the learning rate). Finally, they use the function form to fit the training loss curve of kernel regression and standard pretraining.

**Questions:**

Please refer to the weakness section for my main questions. Some of the minor questions include:

1. How will the current theory predict the batch size to scale with training duration?

2. A minor suggestion is that 8-1-1 learning rate schedule is not defined explicitly in the empirical section and is confusing to understand at the beginning.

**Ethical Concerns:**

["NO or VERY MINOR ethics concerns only"]

**Final Justification:**

This paper build an interesting theoretical model that is reflective of neural network landscape and respond promptly to compare with the baseline in other works. Although the empirical advantage over previous method is slight, the theoretical novelty should be enough for publication in NeurIPS..

**Limitations:**

yes.

**Paper Formatting Concerns:**

I didn't notice any.

**Quality:**

3

**Strengths And Weaknesses:**

## Strength

1. The theoretical setup is well-presented and very easy to understand. The concept of model capacity and relative difficulty is well motivated. The final result is also of high theoretical interest.

2. The theoretical analysis is supported by strong empirical evidence, showing the relevance of the analysis presented in this paper.

## Weakness

This paper lacks a more complete discussion and comparison of previous works. While previous works didn't have an end-to-end theoretical formulation like this paper (which is indeed a unique contribution), the final scaling form presented in this paper is very similar to previous ones, which is worth further discussion. I will list some of the previous works and their similarity below:

1. The idea of using SDE to approximate the discrete optimization process is studied in deep learning literature and the concept of intrinsic time is also introduced in previous works. A very incomplete list includes [1, 2, 3]. Notably this work also considers the interplay between batch size and learning rate (as in this paper).

2. Even in the concept of learning rate scheduling and how it influences loss, [4,5] has already mentioned the decomposition of the integration of learning rate and the noise level (which is determined by a convolution of the learning rate).

3. Finally, it would be worth considering comparing the scaling law formation in this paper with previous works. There should be two comparisons (1) explicitly comparing the formulations and (2) comparing the fitting/prediction quality in real-world experiments.

[1] Stochastic Modified Equations and Dynamics of Stochastic Gradient Algorithms I: Mathematical Foundations
[2] On the SDEs and Scaling Rules for Adaptive Gradient Algorithms
[3] On the Validity of Modeling SGD with Stochastic Differential Equations (SDEs)
[4] Understanding Warmup-Stable-Decay Learning Rates: A River Valley Loss Landscape Perspective
[5] A Multi-Power Law for Loss Curve Prediction Across Learning Rate Schedules

Also, it is worth noticing that while the FSL framework allows varying batch size, it is not studied in depth in this paper. It would improve the clarity of the paper if this is clarified in the abstract and introduction.

---

> ### Author Rebuttal · Authors · 2025-07-31
>
> We sincerely thank the reviewer for the positive support and valuable feedback! We greatly appreciate the insightful review, and the recognition of highlighting the significance of our contribution and the strength of our theory and experiments, as well as the clarity of our writing. We are very glad to address the questions and suggestions raised by the reviewer, which we believe will help further refine our work. Below are our responses to the questions and suggestions raised by the reviewer.
>
> ---
> >**[W1]** This paper lacks a more complete discussion and comparison of previous works.
>
> **[E1]** We thank the reviewer for pointing out the relevant works. We will ensure that the discussion of these works is highlighted in the revised version of the paper.
>
> $\quad$
> >**[W1-1]** The idea of using SDE to approximate the discrete optimization process is studied in deep learning literature and the concept of intrinsic time is also introduced in previous works.
>
> **[E1-1]**
> We thank the reviewer for the helpful comment and would like to clarify that both the SDE formulation and the concept of intrinsic time have indeed been introduced in prior works. Our contribution does not lie in introducing these notions themselves—we cite many relevant references in Section 3, though we have missed some important ones and will work to address that.
>
> Our contribution, rather, lies in showing that both the SDE formulation and, in particular, the notion of intrinsic time are crucial for deriving precise scaling laws that hold under general learning rate schedules. In the revision, we will add a paragraph to clarify these points and better highlight our unique contribution within the SDE-based framework for analyzing stochastic optimization in deep learning.
>
> $\quad$
> >**[W1-2]** Even in the concept of learning rate scheduling and how it influences loss, [4,5] has already mentioned the decomposition of the integration of learning rate and the noise level (which is determined by a convolution of the learning rate).
>
> **[E1-2]** We agree that both [4] and [5] have previously leveraged the notion of intrinsic time (i.e., the integral of the learning rate) to analyze the impact of learning rate schedule (LRS).
>
> - [4] provides a heuristic explanation for the sharp loss reduction observed under warmup–stable–decay (WSD) schedules. In contrast, we establish that WSD-like schedules indeed outperform pure decay schedules by deriving precise scaling laws.
>
> - The theoretical part of [5] analyzes the effect of LRS in a quadratic optimization setting and presents a risk decomposition into an intrinsic-time scaling term and a noise term. Compared to it, we consider a more structured learning problem characterized by two exponents, $s$ and $\beta$, which allows  to characterize the LRS’s influence **quantitatively**. Moreover, we derive precise scaling laws for three LRSs, enabling a direct and quantitative comparison across them.
>
> We will include a more detailed comparison with [4] and [5] in the revision to better clarify the contributions of our work over them.
>
> $\quad$
> >**[W1-3]** Finally, it would be worth considering comparing the scaling law formation in this paper with previous works.
>
> **[E1-3]** We thank the reviewer for this insightful suggestion. We will include a comparison of our FSL formulation with previous works in the revised version of our paper.
>
> **Comparison with Chinchilla-style scaling law [6][7].** Chinchilla-style scaling laws take the form of the final loss $\mathcal{L} = \mathcal{L}_0 + c_1 M^{-a_1} + c_2 D^{-a_2}$, where $\mathcal{L}_0$ represents the irreducible loss, and $c_1$, $c_2$, $a_1$, and $a_2$ are parameters that depend on the specific problem. In contrast, our FSL framework incorporates the effect of LRS through an additional noise term. Moreover, while Chinchilla-style scaling laws are limited to estimating the final loss of a given LRS (different LRS may have different exponents), our FSL model can predict the entire training process loss of any LRS.
>
> **Comparison with multi-power law [5]**. Reference [5] developed an empirical law that incorporates a carefully designed loss reduction term to predict the training loss under general LRSs. In contrast, our work provides a theoretical explanation with a different decomposition: in their law, the loss reduction term is negative (subtracted from the first term), while our noise term is positive (added to the first term). In terms of fitting ability, our FSL slightly outperforms the multi-power law, as shown by the following experimental results:
> We fit the laws on the 8-1-1 training curve and predict on cosine and WSD schedules. The results are shown in the table.
> |       | Cosine   |        | WSD     |        |
> |-------|----------|--------|---------|--------|
> |       | MPL      | Ours   | MPL     | Ours   |
> | $R^2$ | 0.9811   | **0.9825** | 0.9819  | 0.9819 |
> | RMSE ($L^2$ distance) | 0.0528   | **0.0512** | 0.0509  | 0.0509 |
> | MAE ($L^1$ distance) | 0.0276   | **0.0242** | 0.0242  | **0.0240** |
> | WorstE ($L^\infty$ distance) | 0.5886   | **0.5814** | 0.5764  | **0.5691** |
>
> $\quad$
> >**[W2]** Also, it is worth noticing that while the FSL framework allows varying batch size, it is not studied in depth in this paper.
>
> **[E2]** We thank the reviewer for raising this point and would like to clarify our rationale.
>
> First, the study of the influence of learning rate schedules (LRSs) already involves substantial technique development and takes up significant space.
>
> Second, while the FSL framework in principle allows for varying batch sizes, it is derived for SGD. However, practical LLM training uses adaptive optimizers such as Adam, where the interaction between learning rate and batch size can differ substantially. Understanding how the insights from  the FSL framework  translate to such settings would require substantial additional theoretical and empirical analysis.
>
> For these reasons, an in-depth analysis of batch size schedules is better suited for a separate work. We will clarify this point in the revised Conclusion section.
>
> $\quad$
> >**[Q1]** How will the current theory predict the batch size to scale with training duration?
>
> **[A1]** The final-step risk depends on both the learning rate schedule (LRS) and the batch size. Therefore, a comprehensive analysis requires substantial effort and is beyond the scope of this paper. However, under constant learning rate, i.e., it does not scale with training tokens, a careful calculation suggests that the batch size should scale as
>
> $$
> \qquad\qquad\qquad\qquad\qquad\qquad\qquad\qquad\qquad\qquad B \asymp D^{\frac{s}{s+1}}.
> $$
>
> This indicates that **larger batch sizes should be used when training for more tokens**. This theoretical prediction is consistent with empirical Step Law of batch size reported in [8], which also suggests that $B\asymp D^{\gamma}$ for some $\gamma>0$.
>
> $\quad$
> >**[Q2]** A minor suggestion is that 8-1-1 learning rate schedule is not defined explicitly in the empirical section and is confusing to understand at the beginning.
>
> **[A2]** We thank the reviewer for the helpful suggestion. The definition of the 8-1-1 schedule is currently provided in line 605 of the Appendix. We will move it to the main text in the revised version to improve clarity.
>
> ---
> **Reference**
>
> [1] Stochastic Modified Equations and Dynamics of Stochastic Gradient Algorithms I: Mathematical Foundations.
>
> [2] On the SDEs and Scaling Rules for Adaptive Gradient Algorithms.
>
> [3] On the Validity of Modeling SGD with Stochastic Differential Equations (SDEs).
>
> [4] Understanding Warmup-Stable-Decay Learning Rates: A River Valley Loss Landscape Perspective.
>
> [5] A Multi-Power Law for Loss Curve Prediction Across Learning Rate Schedules.
>
> [6] Scaling laws for neural language models.
>
> [7] Training compute-optimal large language models.
>
> [8] Predictable Scale: Part I, Step Law – Optimal Hyperparameter Scaling Law in Large Language Model Pre-training.

---

> > ### Comment · Reviewer_93yY · 2025-08-01
> >
> > I thank the author for the detailed rebuttal and will keep my positive rating.

---

> > > ### Author Response · Authors · 2025-08-01
> > >
> > > Thank you very much for your kind review, and we are glad that you enjoyed our paper!

---

### Decision · Program_Chairs · 2025-09-17

**Decision:**

Accept (spotlight)

**Comment:**

(a) This paper develops a theory to predict the scaling behavior for different learning rate schedules, with applications to LLMs.

(b) Strenghts: currently, the design of learning rate schedulers is more of an art than a science, making this paper a valueable contribution to the community to improve our understanding of scaling behavior. Theory is complemented with strong empirical evidence. All reviewers find the paper easy to read.

(c) Weaknesses: some comparisons with recent works were missing, but were provided during rebuttal. There were some concerns about the hyperparameter choices for the experiments, but seeing as this is a theoretical paper mostly, these concerns are minor.

(d) Accept. The paper is important and on a timely topic, likely with high impact.

(e) Some of the asked ablations were carried out, and several questions were clarified.